# Synaptotagmin-1-dependent phasic axonal dopamine release is dispensable for basic motor behaviors in mice

Benoît Delignat-Lavaud[1,2,3], Jana Kano[1,2,3,8], Charles Ducrot[1,2,3,8], Ian Massé[4], Sriparna Mukherjee[1,2,3,5], Nicolas Giguère [1,2,3], Luc Moquin[6], Catherine Lévesque[7], Samuel Burke[1,2,3], Raphaëlle Denis [1,2,3], Marie-Josée Bourque[1,2,3,5], Alex Tchung [1,2,3], Pedro Rosa-Neto [6], Daniel Lévesque [7], Louis De Beaumont[4] & Louis-Éric Trudeau [1,2,3,5] ✉

In Parkinson's disease (PD), motor dysfunctions only become apparent after extensive loss of DA innervation. This resilience has been hypothesized to be due to the ability of many motor behaviors to be sustained through a diffuse basal tone of DA; but experimental evidence for this is limited. Here we show that conditional deletion of the calcium sensor synaptotagmin-1 (Syt1) in DA neurons (Syt1 cKO[DA] mice) abrogates most activity-dependent axonal DA release in the striatum and mesencephalon, leaving somatodendritic (STD) DA release intact. Strikingly, Syt1 cKO[DA] mice showed intact performance in multiple unconditioned DA-dependent motor tasks and even in a task evaluating conditioned motivation for food. Considering that basal extracellular DA levels in the striatum were unchanged, our findings suggest that activity-dependent DA release is dispensable for such tasks and that they can be sustained by a basal tone of extracellular DA. Taken together, our findings reveal the striking resilience of DA-dependent motor functions in the context of a near-abolition of phasic DA release, shedding new light on why extensive loss of DA innervation is required to reveal motor dysfunctions in PD.

The neuromodulator dopamine (DA) plays a key role in motor control, motivated behaviors, and cognition[1,2]. Blockade of DA receptors severely impairs most motor behaviors[3] and a severe loss of DA in the striatum and other brain regions leads to the characteristic motor dysfunctions of Parkinson's disease (PD). It is striking that in PD, motor dysfunctions only appear when the striatum is severely denervated and that DA replacement therapy with L-DOPA is able to restore motor functions by boosting DA production in the remaining, sparse DA axonal fibers[4]. Such observations are puzzling and ill-understood and

suggest the possibility that a minimal basal tone of DA, coupled with adaptations to the DA system to boost its sensitivity, are sufficient to maintain motor functions[5].

Compatible with such observations, DA is thought to act in the brain as a neuromodulator involved in a form of "volume transmission" and not as a point-to-point fast neurotransmitter[6–9]. DA release occurs not only from release sites in the vast axonal arbor of DA neurons, but also from the neurons' somatodendritic (STD) compartment[10]. Knowledge about the molecular machinery underlying exocytotic DA

[1]Department of Pharmacology and Physiology, Faculty of Medicine, Université de Montréal, Montreal, QC, Canada. [2]Department of Neurosciences, Faculty of Medicine, Université de Montréal, Montreal, QC, Canada. [3]SNC and CIRCA Research Groups, Université de Montréal, Montreal, QC, Canada. [4]Hôpital du Sacré-Cœur-de-Montréal, CIUSSS NIM, Université de Montréal, Montreal, QC, Canada. [5]Aligning Science Across Parkinson's (ASAP) Collaborative Research Network, Chevy Chase, MD 20815, USA. [6]Centre intégré universitaire de santé et de services sociaux (CIUSSS) de l'Ouest-de-l'Île-de-Montréal; Department of Neurology and Neurosurgery, Psychiatry and Pharmacology and Therapeutics, McGill University, Montreal, QC, Canada. [7]Faculty of Pharmacy, Université de Montréal, Montreal, QC, Canada. [8]These authors contributed equally: Jana Kano, Charles Ducrot. ✉e-mail: louis-eric.trudeau@umontreal.ca

release from terminals[11–14] or dendrites[10,11,15–18] is presently fragmentary. Discoveries on such mechanisms are likely to lead to a better understanding of the functions and connectivity of all classes of modulatory neurons.

Vesicular exocytosis requires the concerted action of SNARE proteins and calcium sensors from the synaptotagmin family (Syt)[19]. Of the 17 Syt isoforms identified so far, only Syt1, 2, 3, 5, 6, 7, 9, and 10 have been reported to bind calcium and drive vesicular fusion[20]. Syt1, 2, and 9 were confirmed as calcium sensors for fast synaptic neurotransmitter release[21]. Syt1 was first shown to play a key role in axonal DA release in primary DA neurons[11] in which it is present at both synaptic and non-synaptic release sites[9], but most likely absent from dendrites[11]. Recent work has confirmed and extended this finding in the intact brain by showing that Syt1 is essential for evoked DA release[22]. Syt4 and Syt7 were recently shown to control STD DA release, suggesting a distinct molecular machinery compared to axonal DA release[11,23]. Very recent functional evidence also supports the possibility that Syt1 is also involved in DA release occurring in the ventral midbrain[24,25].

Here, we defined the consequence of loss of Syt1 in DA neurons by evaluating the impact of this deletion on DA-dependent behaviors and by characterizing the selective roles of Syt1 in axonal and STD DA release. We find that basic unconditioned motor functions, and conditioned motivation for food are intact in mice lacking Syt1 in DA neurons (Syt1 cKO$^{DA}$). Combined with our observations that loss of Syt1 leads to extensive loss of axonal DA release in the striatum, to partial loss of DA release in the mesencephalon and to unaltered extracellular DA levels in the striatum and mesencephalon, our findings suggest that basal unconditioned motor functions in rodents only require the maintenance of a basal tone of extracellular DA largely independent of Syt1-dependent phasic release. We hypothesize that this situation is very similar to pre-symptomatic stages of PD and sheds light on the surprising resilience of motor functions after the extensive loss of evoked DA release.

## Results

### Syt1 is the main calcium sensor for fast axonal dopamine release

In line with its high expression and localization in the axonal varicosities of DA neurons, Syt1 is a key regulator of activity-dependent DA release[9,11,22]. We generated conditional deletion of Syt1 in DA neurons (Syt1 cKO$^{DA}$), by crossing Syt1$^{lox/lox}$ mice with DAT$^{IRescre}$ mice (Fig. 1A). Validating earlier results, we found using FSCV ex vivo in brain slices of Syt1$^{+/+}$, Syt1$^{+/-}$, and Syt1$^{-/-}$ mice an extensive reduction of phasic DA release induced by single electrical pulses in both the dorsal striatum (Fig. 1B: 1.42 ± 0.15 μM in Syt1$^{+/+}$; $n$ = 9 mice (5 males/4 females) vs 1.123 ± 0.091 μM in Syt1$^{+/-}$; $n$ = 8 mice (4 M/4 F) and 0.074 ± 0.012 μM in Syt1$^{-/-}$; $n$ = 8 mice (2 M/6 F); one-way ANOVA with Tukey, $F_{(2,22)}$ = 43.02, $P$ = 0,139 for Syt1$^{+/-}$ and $P$ < 0.0001 for Syt1$^{-/-}$) and ventral striatum (Fig. 1C: 1.252 ± 0.08 μM in Syt1$^{+/+}$; $n$ = 9 mice (5 M/ 4 F) vs 0.999 ± 0.03 μM in Syt1$^{+/-}$; $n$ = 8 mice (4 M/4 F) and 0.144 ± 0.019 μM in Syt1$^{-/-}$; $n$ = 8 mice (2 M/6 F); one-way ANOVA with Tukey, $F_{(2,22)}$ = 109, $P$ = 0.0098 for Syt1$^{+/-}$ and $P$ < 0.0001 for Syt1$^{-/-}$) of Syt1$^{-/-}$ mice. In Syt1$^{-/-}$ mice, the residual signal was low in amplitude but still represented DA, as confirmed by the cyclic voltammogram (Supplementary Fig. S1A, B). Stimulation with 10 Hz pulse trains, although prolonging DA release kinetics (results not shown), did not enhance peak DA overflow in the dorsal or ventral striatum of Syt1$^{-/-}$ mice (Supplementary Fig. S1C). Mice in which DA neurons expressed a single allele of Syt1 also showed a small decrease in DA release, which reached significance in the ventral striatum (0.99 ± 0.03 μM, $n$ = 8 Syt1$^{+/-}$ mice (4 M/4 F); one-way ANOVA with Tukey, $F_{(2,22)}$ = 109, $P$ < 0.0001) (Fig. 1C). This observation argues for the existence of only a modest safety factor in the amounts of Syt1 required for normal function of DA neuron terminals. We conclude that Syt1 acts as the main calcium sensor for fast activity-dependent DA release in the striatum, but that other calcium sensors are also likely to play a complementary role, which may be more extensive in the ventral (nucleus accumbens) compared to dorsal striatum.

### Syt1 deletion also reduces DA release in the ventral mesencephalon

As DA release in the cell body region of DA neurons is also calcium-dependent and since synaptotagmin isoforms and active zone proteins have been suggested to have a role in STD DA release[11,23,26], we also examined activity-dependent DA overflow in the ventral mesencephalon (Fig. 1D–E). We first measured DA release in the VTA of Syt1$^{-/-}$ mice using an optimal FSCV paradigm, with pulse-train stimulation (30 pulses at 10 Hz) and aCSF containing the DAT blocker nomifensine and the D2 antagonist sulpiride, allowing the detection of STD DA release without the influence of DA uptake and D2 autoreceptors activation[23]. We observed a robust ≈68% decrease of activity-dependent DA overflow in the VTA of Syt1$^{-/-}$ mice (0.09 ± 0.01 μM, $n$ = 8 (2 M/6 F), vs 0.28 ± 0.03 μM in $n$ = 9 Syt1$^{+/+}$ mice (5 M/4 F); one-way ANOVA with Tukey, $F_{(2,22)}$ = 20.07, $P$ < 0.0001) (Fig. 1D). This decrease was also significant in heterozygote mice (0.20 ± 0.02 μM, $n$ = 8 (4 M/4 F); one-way ANOVA with Tukey, $F_{(2,22)}$ = 20.07, $P$ = 0.0310). Similar recordings performed in the SNc also revealed a robust reduction (≈65%) of evoked DA overflow (0.09 ± 0.01 μM, $n$ = 5 Syt1$^{-/-}$ mice (4 M/1 F) vs 0.25 ± 0.04 μM, $n$ = 6 Syt1$^{+/+}$ male mice; one-way ANOVA with Tukey, $F_{(2,13)}$ = 8.86, $P$ = 0.0053) (Fig. 1E).

Because there is yet no quantitative evidence for the presence of Syt1 in the STD compartment of DA neurons[11] or any other neurons, this finding was unexpected. Interestingly, although the VTA region is well known for containing the cell body and dendrites of DA neurons, it has also been previously suggested to contain a small contingent of local dopaminergic axon collaterals with potential axonal release sites[10]. The available anatomical data is however limited[27,28]. On the other hand, the SNc is believed to be devoid of axonal DA release sites[29–31], but here again, the supporting data is limited.

### Somatodendritic optogenetic stimulation reveals unaltered STD DA release in the absence of Syt1

In view of the surprising reduction of evoked DA overflow in the ventral mesencephalon of Syt1$^{-/-}$ mice, we devised a strategy to trigger STD DA release more selectively. For this, we combined FSCV with selective optogenetic stimulation of the STD compartment of neurons by expressing a STD-targeted version of channelrhodopsin (ChR2-Kv) (Fig. 2). Kv2.1 channels were previously reported to be restricted to the somatic and proximal dendritic membrane and absent from distal dendritic membrane, axons, and nerve terminals of cortical and hippocampal neurons[32–34]. A 65 amino acid motif of the Kv2.1 voltage-gated potassium channel fused with the carboxy terminus of ChR2-EYFP[35] was inserted in a Cre-dependent AAV vector (AAV2/5-hsyn-DIO-ChR2-eYFP-Kv). A standard cell-wide hChR2 (AAV2/5-hsyn-DIO-ChR2-eYFP), previously used to trigger DA release in the striatum[36] and mesencephalon[23] was used as a control.

Validation of the construct in primary DA neurons from DAT$^{IRES}$Cre mice showed the expected STD expression of the ChR2-Kv construct, with no expression in TH$^+$/MAP2$^-$ processes and varicosities (Supplementary Fig. S2). Expression of the construct in vivo in DAT$^{IRES}$Cre mice was efficient and restricted to the ventral mesencephalon (Fig. 2A). Confocal imaging at ×60 revealed a membrane localization of ChR2-Kv at the soma of DA neurons (Fig. 2B). In the striatum, no eYFP signal was detected in the dorsal striatum but sparse signal was detectable in the ventral striatum, suggesting limited expression in a subset of VTA DA neuron axons in addition to the predominant STD expression. In line with this limited axonal expression of ChR2-Kv, DA release triggered by pulse-train optogenetic stimulation of ChR2-Kv in the dorsal striatum of Syt1$^{+/+}$ mice was about tenfold lower compared to release evoked with conventional ChR2 (0.13 ± 0.02 μM, $n$ = 7 mice (5 M/2 F) vs 1.22 ± 0.16 μM, $n$ = 5 mice

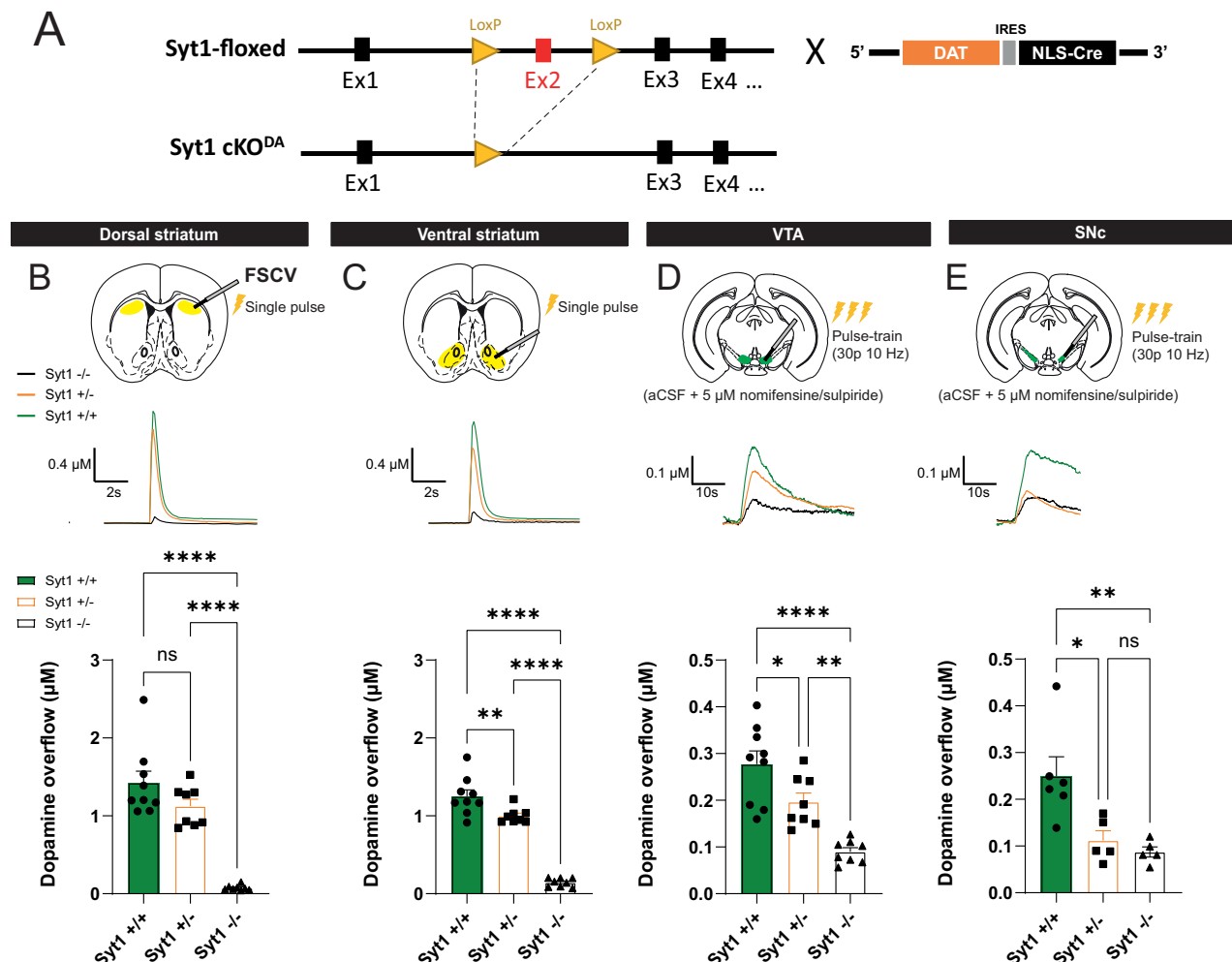

**Fig. 1 | Syt1 is the main calcium sensor for fast axonal dopamine release.**
**A** Generation of conditional knockout of Syt1 in DA neurons by crossing Syt1-floxed mice (Syt1$^{lox/lox}$) with DAT$^{IREScre}$ mice. **B** Fast-scan cyclic voltammetry recording of Syt1 cKO$^{DA}$ mice in the dorsal striatum. Representative traces (top) and quantification of peak amplitude (bottom) obtained with single-pulse electrical stimulation (1 ms, 400 μA) in Syt1$^{+/+}$ ($n = 18$ slices/9 mice), Syt1$^{+/-}$ ($n = 16/8$) and Syt1$^{-/-}$ mice ($n = 16/8$). **C** Same, but in the ventral striatum (NAc core and shell, $n = 18$ slices/9 mice in Syt1$^{+/+}$, $n = 16/8$ in Syt1$^{+/-}$ and $n = 16/8$ in Syt1$^{-/-}$). **D** Representative traces (top)

and quantification of peak amplitude (bottom) obtained in the VTA ($n = 16$ slices/9 mice in Syt1$^{+/+}$, $n = 14/8$ in Syt1$^{+/-}$ and $n = 16/8$ in Syt1$^{-/-}$) with aCSF containing nomifensine (DAT blocker) and sulpiride (D2 antagonist) (both at 5 μM), and pulse-train stimulation (30 pulses of 1 ms at 10 Hz, 400 μA). **E** Same for the SNc ($n = 11$ slices/6 mice in Syt1$^{+/+}$, $n = 10/5$ in Syt1$^{+/-}$ and $n = 9/5$ in Syt1$^{-/-}$). Error bars represent ± SEM and the statistical analysis was carried out by one-way ANOVAs followed by Tukey tests (ns, non-significant; *$P < 0.05$; **$P < 0.01$; ***$P < 0.001$; ****$P < 0.0001$). Source data are provided as a Source Data file.

(3 M/2 F); two-way ANOVA with Tukey, F$_{(2, 36)}$ = 62.84, $P < 0.0001$), and twofold lower in the ventral striatum (0.79 ± 0.12 μM, $n = 7$ mice (5 M/2 F) vs 1.58 ± 0.29 μM, $n = 5$ mice (3 M/2 F); two-way ANOVA with Tukey, F$_{(2, 36)}$ = 16.49, $P = 0.0003$) (Fig. 2C, D).

Intriguingly, DA overflow evoked with ChR2-Kv in the dorsal striatum of Syt1$^{-/-}$ mice (0.13 ± 0.05 μM) was not significantly different from the release obtained in Syt1$^{+/+}$ mice (0.13 ± 0.02 μM) (two-way ANOVA with Tukey, F$_{(2, 36)}$ = 62.84, $P > 0.9999$). In the ventral striatum, evoked DA overflow in Syt1$^{-/-}$ mice was more than twofold lower compared to Syt1$^{+/+}$ mice (0.33 ± 0.07 μM, $n = 9$ mice (6 M/3 F) vs 0.79 ± 0.12 μM, $n = 7$ mice (5 M/2 F), two-way ANOVA with Tukey, F$_{(2, 36)}$ = 16.49, $P = 0.0256$) (Fig. 2C, D). The sodium channel blocker TTX (1 μM) abolished this small axonal DA release in all mice, thus demonstrating the requirement of action potentials for this response (Fig. 2H).

In the VTA (Fig. 2E), the release of DA induced by optogenetic train stimulation with hChR2 was easily detectable and of an amplitude similar to previously reported (203 ± 13 nM, $n = 5$ mice (3 M/2 F)[23]. Release evoked by optogenetic stimulation with ChR2-Kv in Syt1$^{+/+}$ mice was significantly lower (132 ± 16 nM, $n = 7$ mice (5 M/2 F); two-way

ANOVA with Šidák, F$_{(2, 36)}$ = 30.23, $P = 0.0005$) compared to hChR2 (203 ± 13 nM, $n = 5$ mice (3 M/2 F) (Fig. 2E) and the signal detected in Syt1$^{-/-}$ mice was lower compared to Syt1$^{+/+}$ mice (57 ± 7 nM, $n = 9$ mice (6 M/3 F); two-way ANOVA with Šidák, F$_{(2, 36)}$ = 30.23, $P < 0.0001$). To exclude any contribution from local axonal DA release, additional FSCV recordings were obtained in the presence of TTX (1 μM). Although STD DA release is normally activity-dependent and blocked by TTX, we hypothesized that direct membrane depolarization induced by ChR2-Kv activation might trigger sufficient calcium influx to trigger DA release from the STD compartment. Under these conditions, STD DA release was indeed evoked by optogenetic stimulation of ChR2-Kv in Syt1$^{+/+}$ mice (44 ± 8 nM, $n = 7$ mice (5 M/2 F)) and the amount of release was not significantly reduced in Syt1$^{-/-}$ mice (28 ± 7 nM, $n = 9$ (6 M/3 F)). Together these observations suggest that when STD DA release is triggered using an approach that minimizes any axonal activation, the signal that is detected is unaltered by loss of Syt1. The results also suggest that some of the DA release evoked by train stimulation in the ventral mesencephalon includes an axonal component, with electrical and optical stimulation with hChR2 therefore triggering a mixture of axonal and STD DA release.

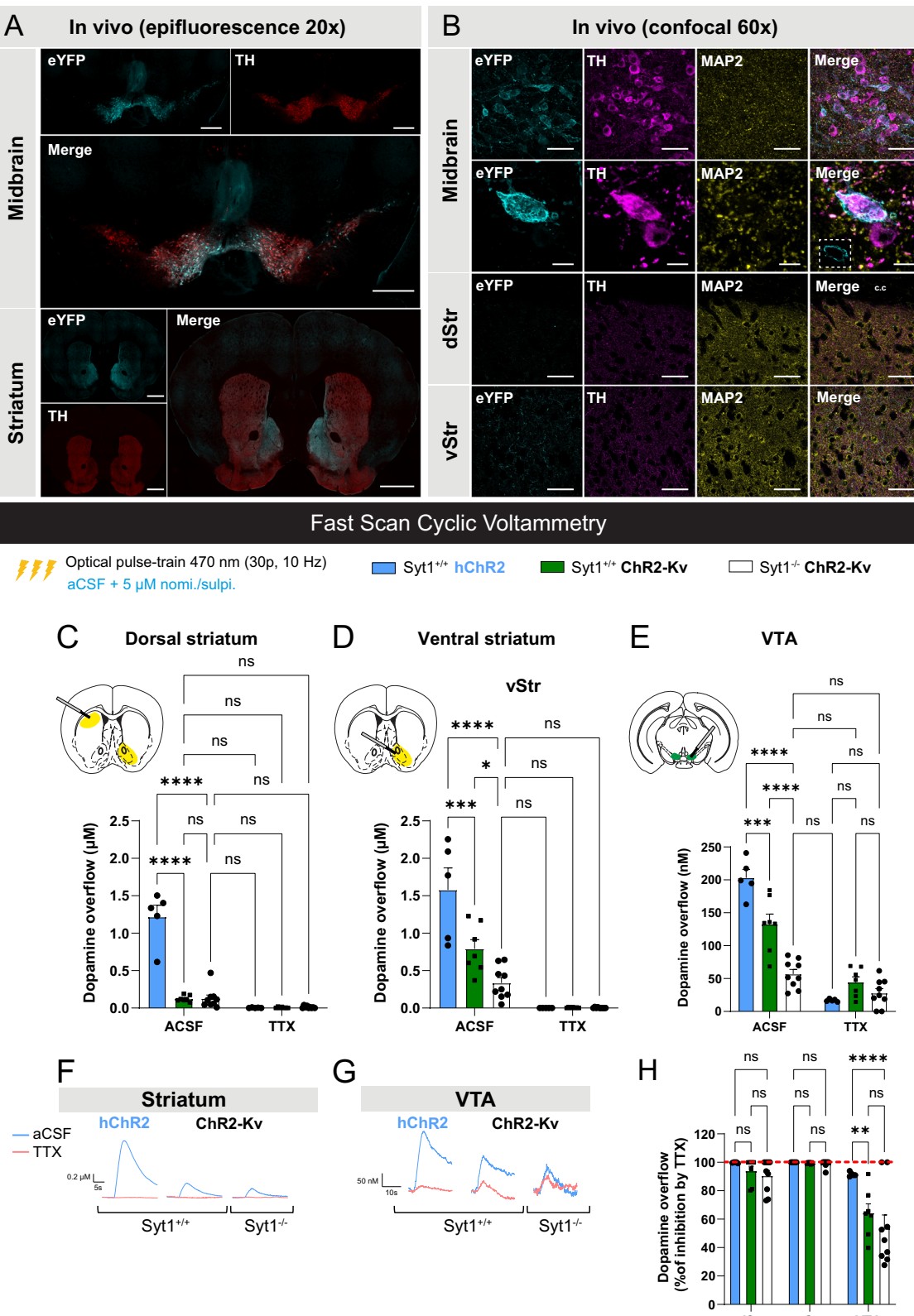

## Syt1 cKO^DA mice do not exhibit substantial motor defects

DA is known to play a key role in multiple forms of movement[37,38]. Notably, loss of striatal DA innervation in PD is responsible for the cardinal motor features observed in the disease[39]. Blockade of DA receptors induces catalepsy and a range of other motor deficits in rodents and other species[3,40]. Based on our results showing a dramatic impairment of phasic axonal DA release in Syt1^−/− mice, we

hypothesized that this would lead to major motor deficits in Syt1 cKO^DA mice.

First, we evaluated motor coordination using the rotarod (Fig. 3A) and the pole test (Fig. 3B). Surprisingly, Syt1^−/− and Syt1^+/− showed no deficits and even exhibited better performance on the rotarod task than wild-type littermates, with a significantly higher latency to fall from the rod (124 ± 12 s for 8 Syt1^+/+ mice (5 M/3 F) vs 188 ± 20 s for 8

**Fig. 2 | Somatodendritic optogenetics reveals unaltered STD DA release in the absence of Syt1. A** Immunohistochemistry in brain slices of adult Syt1 cKO[DA] mice infected with AAV2/5-hsyn-DIO-ChR2-eYFP-Kv ($n = 4$ mice, one representative set of images is shown), showing expression of ChR2-Kv (eYFP) in the whole ventral mesencephalon (scale bar = 500 μm) and striatum (scale bar = 1 mm) using epi-fluorescence microscopy at ×20. **B** Expression of ChR2-Kv in the same mice evaluated by confocal microscopy at ×60 shows infected DA neurons (TH) in the midbrain (top panel, scale bar = 50 μm). Optical zoom and z-stack on an infected DA neuron showing the membrane distribution of the eYFP signal (bottom panel, scale bar = 10 μm, insert in the merge image shows the eYFP signal at a single focal plane). Evaluation of the eYFP signal in the striatum shows no axonal processes in the dorsal sector but a small contingent of positive fibers in the ventral sector of the striatum (scale bar = 50 μm). **C–E** Fast-scan cyclic voltammetry recordings with average [DA]o peaks obtained in the dorsal striatum (**C**), ventral striatum (**D**) and VTA (**E**) slices of infected Syt1[+/+] and Syt1[−/−] mice with AAV2/5-hsyn-DIO-ChR2-eYFP-Kv (ChR2-Kv, $n = 14$ slices/7 mice in Syt1[+/+], $n = 17$ slices/9 mice in Syt1[−/−]) or AAV5-EF1a-DIO-hChR2(H134R)-eYFP (control hChR2, $n = 10$ slices/5 mice). Representative traces for the striatum (dorsal + ventral) and VTA are shown in (**F**) and (**G**). DA release was optically triggered in each region using pulse-train stimulation (30 pulses of 470 nm blue light at 10 Hz) in ACSF containing 5 μM of nomifensine and sulpiride. **H** TTX (1 μM) effect on average [DA]o peaks in the striatum and the VTA (% of inhibition) ($n = 14$ slices/7 mice in Syt1[+/+], $n = 17$ slices/9 mice in Syt1[−/−]). Error bars represent ± SEM and the statistical analysis was carried out by two-way ANOVA followed by Šidák (**E**) and Tuckey (**C**, **D**, **H**) tests (ns, non-significant; *$P < 0.05$; **$P < 0.01$; ***$P < 0.001$; ****$P < 0.0001$). Source data are provided as a Source Data file.

Syt1[+/−] mice (4 M/4 F); one-way ANOVA with Dunnett test, $F_{(2, 21)} = 4.51$, $P = 0.0460$ and 197 ± 23 s for 8 Syt1[−/−] mice (2 M/6 F), one-way ANOVA with Dunnett test, $F_{(2, 21)} = 4.51$, $P = 0.0220$). In the pole test, no statistical difference was observed between genotypes for the time required for the animals to orient themselves facing in a downward direction (t-turn). For the time required to climb down the pole, no difference was detected between Syt1[+/+] and Syt1[−/−] mice, although Syt1[+/−] mice performed the task faster than Syt1[+/+] mice (8 ± 0.7 s for Syt1[+/−] vs 11 ± 0.5 s for Syt1[+/+], $n = 8$ (5 M/3 F); one-way ANOVA with Dunnett test, $F_{(2, 21)} = 4.706$, $P = 0.0414$). Front paw grip strength was evaluated in the grip test (Fig. 3C). No statistical differences were observed regarding the force developed by the mice on the grid.

We next measured spontaneous locomotion and locomotion induced by the psychostimulant drugs cocaine (20 mg/kg) and amphetamine (5 mg/kg). We measured the traveled distances using an open field for 20 min, followed by another 40 min after drug or vehicle injection (0.9% saline). No difference was detected between Syt1[+/+] ($n = 8$ (4 M/4 F)) and Syt1[−/−] ($n = 8$ (3 M/5 F)) mice on basal locomotion after saline injection. A mixed-effects ANOVA model with the Geisser–Greenhouse correction revealed a significant effect of time ($F_{(5.949, 123.8)} = 29.21$, $P < 0.0001$), but not of genotype ($F_{(2, 21)} = 1.745$, $P = 0.1991$), nor an interaction between time x genotype ($F_{(22, 229)} = 1.285$, $P = 0.1825$) (Fig. 3D), with only a non-significant trend for higher traveled distance within the 5 min post injection in Syt1[+/−] mice (Fig. 3L, 91% of baseline ± 16%, $n = 8$ (4 M/4 F) Syt1[+/−] mice vs 56 ± 13%, $n = 8$ (4 M/4 F) Syt1[+/+] mice; one-way ANOVA with Dunnett test, $F_{(2, 21)} = 3.068$, $P = 0.1211$). The global decrease in locomotion over time after saline injection reflects the normal habituation to the open field.

The locomotor response to the DA transporter blocker cocaine was comparable between genotypes. A mixed-effects ANOVA model with the Geisser–Greenhouse correction revealed only a significant effect of time ($F_{(1.975, 36.63)} = 27.28$, $P < 0.0001$), but not of genotype ($F_{(2, 19)} = 0.1283$, $P = 0.8803$) nor an interaction between time × genotype ($F_{(22, 204)} = 0.6858$, $P = 0.8509$) (Fig. 3E). The peak magnitude of the increase in locomotion was also not different between groups (Fig. 3G). However, locomotion induced by the DA releaser amphetamine was sharply elevated in Syt1[−/−] mice. A mixed-effects ANOVA model with the Geisser–Greenhouse correction revealed a significant effect of time ($F_{(2.79, 57.28)} = 50.76$, $P < 0.0001$), of genotype $F_{(2, 21)} = 6.902$, $P = 0.005$) and an interaction between time × genotype $F_{(22, 226)} = 3.218$, $P < 0.0001$). The difference between groups became significant 10 min after the injection (+714 ± 103% in Syt1[−/−] mice, $n = 8$ (2 M/6 F) vs +252 ± 29% in Syt1[+/+], $n = 8$ (4 M/4 F); mixed-effect ANOVA model with Dunnett test, $F_{(22, 226)} = 3.218$, $P = 0.0046$), and was maximal 20 min post injection (+1045 ± 126% vs +523 ± 58%; mixed-effect ANOVA model with Dunnett test, $F_{(22, 226)} = 3.218$, $P = 0.0069$) (Fig. 3F). Cumulative distance for the 40 min under treatment confirmed this effect with an average traveled distance for Syt1[−/−] mice of +777 ± 67% vs +472 ± 53% for Syt1[+/+] (one-way ANOVA with Tukey, $F_{(2, 20)} = 7.441$,

$P = 0.0072$) and +481 ± 73% for Syt1[+/−] mice, $n = 8$ (4 M/4 F) (one-way ANOVA with Tukey, $F_{(2, 20)} = 7.441$, $P = 0.0117$) (Fig. 3G).

The observed increase in amphetamine-induced locomotion could result from increased amphetamine-induced DA secretion or from striatal DA receptor sensitization. We next tested this hypothesis by administering the selective D1 receptor agonist SCH23390 (50 μg/kg), the D2 receptor agonist quinpirole (0.2 mg/kg) or the D2 receptor antagonist raclopride (1 mg/kg). At the selected doses, all drugs caused an abrupt decrease in locomotion, with no significant differences between genotypes when considering the complete drug treatment period. A mixed-effects ANOVA with Geisser–Greenhouse corrections revealed only a significant effect of time ($P < 0.0001$, with $F_{(6.457, 165.5)} = 80{,}09$ for SCH23390, $F_{(4.397, 105.9)} = 272.4$ for quinpirole and $F_{(2.782, 80.69)} = 378{,}3$ for raclopride), but not of the genotype (respectively, $F_{(2, 26)} = 0.865$, $P = 0.433$; $F_{(2, 25)} = 2.672$, $P = 0.089$ and $F_{(2, 319)} = 2.474$, $P = 0.086$) nor any interaction between time x genotype for SCH23390 and raclopride (respectively, $F_{(22, 282)} = 1.374$, $P = 0.126$ and $F_{(22, 319)} = 1.411$, $P = 0.106$), except for quinpirole treatment ($F_{(22, 265)} = 1.753$, $P = 0.022$) (Fig. 3H–K). However, comparing the mean traveled distance within the first 5 min following drug injection, a time likely to correspond to partial receptor occupancy (Fig. 3L), Syt1[−/−] mice showed a lower traveled distance in response to the D2 agonist quinpirole compared to Syt1[+/+] mice (respectively, 21 ± 3%, $n = 10$ (3 M/7 F), vs 47 ± 8%, $n = 8$ (2 M/6 F); Brown–Forsythe ANOVA with $F_{(2,13.48)} = 5.266$, $P = 0.0204$ and a Dunnett T3 post hoc test, $P = 0.035$). The opposite pattern was detected in response to the D2 antagonist raclopride, with significantly less inhibition of locomotion in the Syt1[−/−] mice compared to Syt1[+/+] mice (14 ± 3% in Syt1[−/−], $n = 10$ males vs 3 ± 1% in Syt1[+/+], $n = 10$ (4 M/6 F); one-way ANOVA with Dunnett test, $F_{(2, 27)} = 5.343$, $P = 0.0056$). During this period, no significant differences were once again observed among genotypes in their response to the D1 agonist SCH23390. These results suggest that D2 receptor neuroadaptations occur after genetic deletion of Syt1 in DA neurons, but this has only a limited impact on DA-mediated motor behaviors.

**Phasic DA release is unnecessary for motivation to work for food**
Next, we tested if the abrogation of phasic release of DA in Syt1 cKO mice would impair learning in an operant food-rewarded nose-poking task. We tested Syt1[+/+] ($n = 8$, 3 M/7 F) and Syt1[−/−] ($n = 10$, 5 M/5 F) mice of both sexes with the open-source nose-poking feeding devices "FED3"[41]. First, the devices were placed in the home cage of Syt1 animals and a fixed ratio 1 (FR1) paradigm was used, in which mice learned to nose-poke on the active port for receiving a single food pellet. We quantified how nose-poking for pellets varied over the circadian cycle, by running mice on the FR1 task for 3 consecutive days. As expected, pellet consumption and active pokes were higher during the dark cycle, which is the active period for mice. A mixed-effects ANOVA model with the Geisser–Greenhouse correction revealed no significant global effect of genotype for pellet consumption ($F_{(1, 16)} = 0.1489$, $P = 0.7047$) nor for nose-poking ($F_{(1, 16)} = 0.1468$, $P = 0{,}7067$). However, a significant effect of time (respectively $F_{(7.988,}$

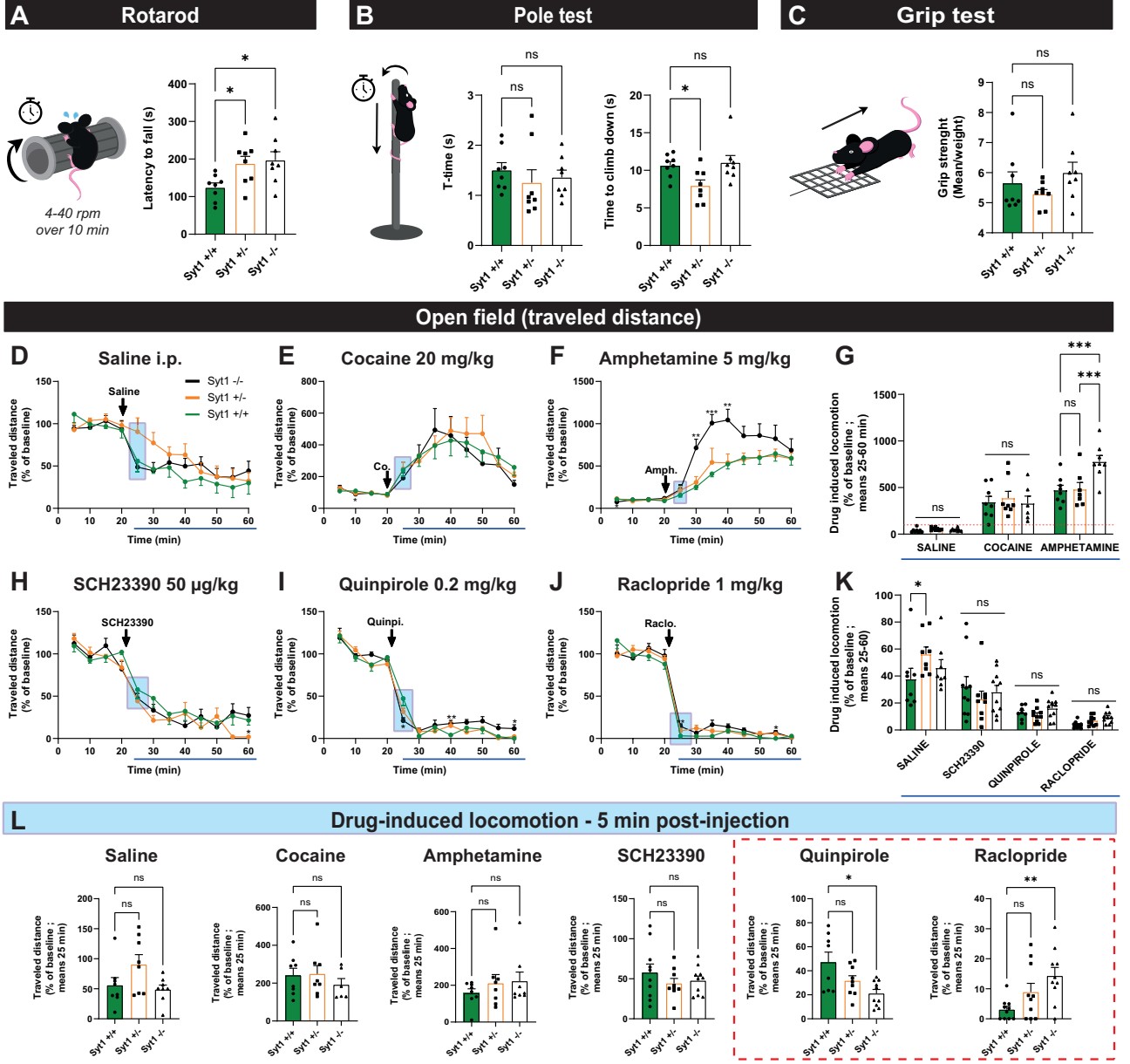

**Fig. 3 | Syt1 cKO^DA mice do not exhibit any substantial motor defects. A** Latency to fall from the device during the rotarod test for Syt1^{+/+}, Syt1^{+/-}, and Syt1^{-/-} mice (n = 8, mean of three attempts). **B** Time for Syt1^{+/+}, Syt1^{+/-}, and Syt1^{-/-} mice (n = 8) to turn (t-turn) and climb down the vertical pole (mean of three attempts/2 days). **C** Forelimb grip strength (mean of three trials/weight) developed by Syt1^{+/+}, Syt1^{+/-} and Syt1^{-/-} mice (n = 8), using a force sensor connected to a grid. Statistical analysis for (**A**–**C**) were carried out by one-way ANOVAs followed by Dunnett tests. **D**–**L** Locomotion of Syt1 cKO^DA mice measured as traveled distance (% of a 20 min baseline) under saline treatment (n = 8 mice) (**D**), cocaine at 20 mg/kg (n = 8 Syt1^{+/+}/Syt1^{+/-} and 6 Syt1^{-/-}) (**E**), amphetamine at 5 mg/kg (n = 8 mice) (**F**), the D1 agonist SCH23390 at 50 µg/kg (n = 10 Syt1^{+/+}/Syt1^{-/-} and 9 Syt1^{+/-}) (**H**), the D2 agonist quinpirole at 0.2 mg/kg (n = 10 Syt1^{-/-}/Sy^{+/-} and 8 Syt1^{+/+}) (**I**) and the D2 antagonist raclopride at 1 mg/kg (n = 10 mice) (**J**). Statistical analyses for (**D**–**J**) were carried out by two-way ANOVAs followed by Dunnett tests. **G, K** Average drug-induced locomotion (% of baseline) during a 40 min recording period (mean between 25 and 60 min) are represented for cocaine/amphetamine treatment (**G**) and for SCH23390/quinpirole/raclopride treatments (**K**). Blue rectangles indicate the average traveled distance for each mouse at the 5 min time point after receiving an i.p. injection of each tested drugs (**L**). Statistical analysis was carried out by one-way ANOVAs followed by Dunnett tests (**A**–**C**, **L**), mixed-effects ANOVA model with the Geisser–Greenhouse correction (**D**–**F**, **H**–**J**) and two-way ANOVAs followed by Tuckey tests (**G**, **K**). Error bars represent ± SEM (ns, non-significant; *P < 0.05; **P < 0.01; ***P < 0.001; ****P < 0.0001). Source data are provided as a Source Data file.

$_{126.0}) = 17.85$ and $F_{(7.985, 125.9)} = 17.91$, $P < 0.0001$) and an interaction between time x genotype was observed (respectively $F_{(35, 552)} = 2.611$ and $F_{(35, 552)} = 2.631$, $P < 0.0001$) (Fig. 4B, C and Supplementary Fig. S3B, S3C). Interestingly, we noted a ~4 h window of enhanced nose-poking rates starting before the onset of the dark cycle (Fig. 4B, C). This nocturnal peak of activity was found to be significantly increased at ZT16 in Syt1^{-/-} mice, only during the first night, with $8.5 ± 4$ pellets consumed for $8.4 ± 4$ active pokes in Syt1^{+/+} mice (n = 8) versus $26.9 ± 3$ for both pellets and active pokes in Syt1^{-/-} mice (n = 10)

(two-way ANOVA with Šidák, $F_{(35, 568)} = 2.49$, $P < 0.0001$). The performance of the mice under the FR1 task was evaluated by measuring the poke efficiency (% of total pokes on the active port) over the 3 days on FR1. We found a significant effect of time (two-way ANOVA, $F_{(35, 502)} = 7.059$, $P < 0.0001$) but no significant difference between genotypes (two-way ANOVA, $F_{(1, 502)} = 0.5806$, $P = 0.4464$) or interaction between time × genotype (two-way ANOVA, $F_{(35, 502)} = 1.323$, $P = 0.1061$) (Fig. 4D), suggesting that overall, all mice learned how to use the active port to obtain their food at the same rate.

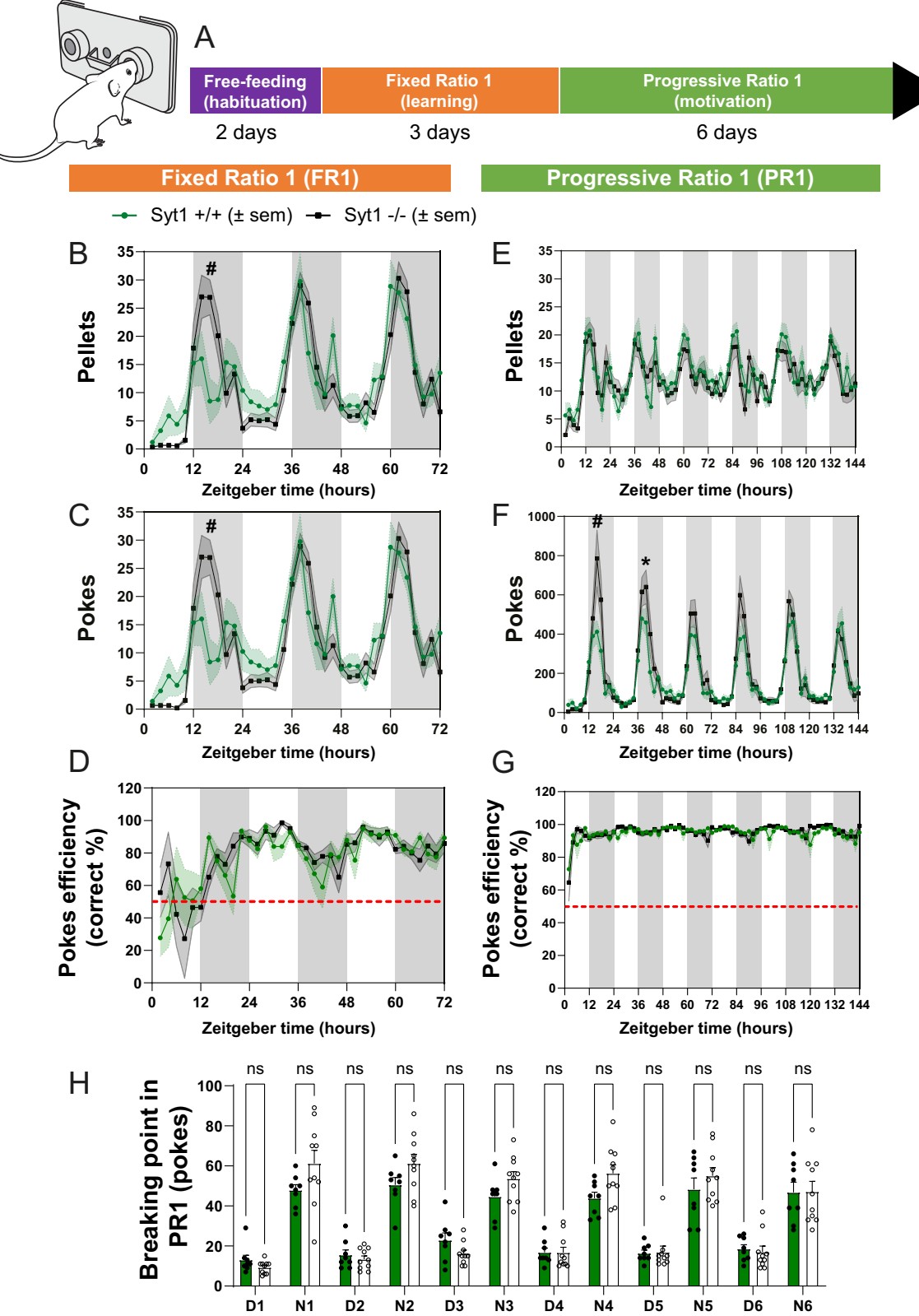

Next, we examined motivation for food. To determine this, we measured their willingness to work in a closed-economy food-rewarded operant task, which has been used to quantify the economic demand for food[42,43], as well as changes in economic demand due to manipulation of the DA system[44,45]. After completion of the FR1 task, the protocol used in the FED3 device was changed for a 6-day repeating progressive ratio (PR1) task, in which the nose-poking

requirement began on FR1 and increased by 1 poke each time a pellet was earned. When mice refrained from poking on either active or inactive port for 30 min, the ratio was reset to FR1. Once again, the nose-poking rates of the mice were higher during the dark phase of their daily cycle. A mixed-effects ANOVA model with the Geisser–Greenhouse correction revealed no global significant effect of genotype either for pellet consumption ($F_{(1, 16)} = 0.639$, $P = 0.436$) or

**Fig. 4 | Syt1 is dispensable for motivation to work for food. A** Schematic representation of operant food-rewarded nose-poking protocol in Syt1 cKO$^{DA}$ mice. **B, C** Time course of the number of pellets earned (**B**) and pokes made (**C**) by Syt1$^{+/+}$ (green lines, $n = 8$ mice) and Syt1$^{-/-}$ (black lines, $n = 10$) mice using FED3 feeding devices over 3 days in a fixed ratio (FR) paradigm (1 pellet for 1 poke). **D** Efficiency of the pokes (% of correct port entries) in the FR1 condition. **E, F** Time course of the number of pellets earned by Syt1$^{+/+}$ and Syt1$^{-/-}$ mice (**E**) and pokes (**F**) over 6 days in a progressive ratio (PR) 1 paradigm (the nose-poking requirement began on FR1 and increased by 1 poke each time a pellet was earned. The requirement was reset to FR1 if no poking was performed on either the active or inactive port for 30 min). **G** Efficiency of the pokes (% of correct port entries) in the

PR1 condition. **H** Maximal breaking points (poke thresholds) achieved by Syt1$^{+/+}$ (green, $n = 8$) and Syt1$^{-/-}$ (black, $n = 10$) during each day (D) and night (N) session of the progressive ratio protocol. For each experiment, mice were placed in a 12 h/12 h light/dark cycle and the Zeitgeber time corresponds to the onset of the devices. Statistical analyses were carried out by mixed-effects ANOVA model with the Geisser–Greenhouse correction (**A–F**) and two-way ANOVAs followed by Šidák's test (**G**). Error shading bands in (**A–F**) (green for Syt1$^{+/+}$, light gray for Syt1$^{-/-}$) and error bars in (**G**) represent ± SEM (ns, non-significant; *$P < 0.05$; #$P < 0.0001$). The schematic diagram of the FED3 nose-poking device is modified from https://github.com/KravitzLabDevices/FED3/blob/main/photos/mouse_feeder.svg. Source data are provided as a Source Data file.

nose-poking ($F_{(1, 16)} = 0.715$, $P = 0.4103$). However, a significant effect of time (respectively, $F_{(71, 1134)} = 8.977$ and $F_{(71, 1134)} = 26.00$, $P < 0.0001$) and an interaction between time × genotype was detected for poking activity ($F_{(71, 1134)} = 1.589$, $P = 0.002$) (Fig. 4E, F and Supplementary Fig. S3D, S3E) This result shows that motivation for food varies over the circadian cycle such that mice are willing to work harder during the dark cycle vs. the light cycle, in line with previous results[41]. Interestingly, although in PR1, the pattern of response was not significantly different between Syt1$^{+/+}$ and Syt1$^{-/-}$ animals in terms of pellets earned (Fig. 4E), Syt1$^{-/-}$ showed a significant increase of nose-poking activity during the first night at 4 h and 6 h after the onset of the dark cycle, with $785 \pm 145$ pokes vs $412 \pm 58$ for Syt1$^{+/+}$ mice at 4 h (two-way ANOVA with Šidák, $F_{(71, 1134)} = 1.589$, $P < 0.0001$) (Fig. 4F) and $574 \pm 118$ pokes vs $314 \pm 78$ at 6 h (two-way ANOVA with Šidák, $F_{(71, 1134)} = 1.589$, $P = 0.0106$), respectively. The performance of the mice under the PR1 task was evaluated by measuring poke efficiency (% of total pokes on the active port) over the 6 days on PR1. We found a significant effect of time (two-way ANOVA, $F_{(71, 1123)} = 5.558$, $P < 0.0001$) but no significant difference between genotypes (two-way ANOVA, $F_{(1, 1123)} = 2.577$, $P = 0.1087$), nor interaction between time × genotype (two-way ANOVA, $F_{(71, 1123)} = 1.095$, $P = 0.281$), revealing that all mice showed high performance on this task, with about ~95% of correct pokes (Fig. 4G). Arguing against changes in motivation for food, daily breaking points were not different between Syt1$^{+/+}$ and Syt1$^{-/-}$ mice in this experiment, although they were expectedly higher during the dark cycle (Fig. 4H). Overall, these results suggest that loss of Syt1 in DA neurons and the associated loss of phasic DA release do not impair basic motivation to work for food.

### Basal extracellular DA levels and total tissue DA are not altered in Syt1 cKO$^{DA}$ mice

Taken together, our behavioral results suggest that despite a >90% impairment of phasic DA release in the striatum and a substantial decrease in DA release in the mesencephalon, Syt1 cKO$^{DA}$ mice do not exhibit any obvious defects in basic unconditioned DA-dependent motor tasks as well as in a conditioned food motivation task. Our observations lead us to hypothesize that such behaviors do not require phasic, activity-dependent DA release and can be maintained by basal resting levels of extracellular DA, previously shown to be unimpacted in Syt1 cKO$^{DA}$ mice[22]. To validate this, we used microdialysis in anesthetized mice implanted with two cannulas to measure extracellular DA and metabolites in the dorsal striatum and in the ventral mesencephalon. Following this sampling, the striatum of each animal was also microdissected, and the total striatal content of DA and metabolites were quantified by HPLC (Fig. 5A). Our results reveal that striatal tissue contents of DA, DOPAC, serotonin (5-HT) and norepinephrine (NE) were not significantly different in Syt1$^{-/-}$ mice ($n = 12$, 9 M/3 F)) compared to Syt1$^{+/+}$ mice ($n = 10$, 4 M/6 F) (unpaired $t$ test, respectively $F_{(9, 11)} = 1.083$, $P = 0.946$; $F_{(9, 11)} = 2.966$, $P = 0.857$; $F_{(9, 11)} = 1.122$, $P = 0.124$ and $F_{(9, 11)} = 2.049$, $P = 0.572$) (Fig. 5B). In addition, the DA/DOPAC ratio was found to be unchanged in Syt1$^{-/-}$ mice, suggesting an intact DA turnover (unpaired $t$ test, $F_{(11, 9)} = 1.001$, $P = 0.7971$). Microdialysis similarly revealed that even if activity-dependent DA release

was mostly blocked in Syt1$^{-/-}$ mice, basal extracellular DA levels of these mice were not reduced in comparison to Syt1$^{+/+}$ mice in the striatum (unpaired $t$ test, $F_{(11, 9)} = 2.326$, $P = 0.215$) and mesencephalon (unpaired $t$ test, $F_{(11, 8)} = 2.507$, $P = 0.157$) (Fig. 5C). Similarly, extracellular levels of DOPAC (unpaired $t$ test, $F_{(11, 10)} = 1.355$, $P = 0.416$ for striatum, $F_{(9, 11)} = 2.768$, $P = 0.684$ for midbrain), and NE (Welch's $t$ test, $F_{(11, 9)} = 4.158$, $P = 0.268$ for striatum, unpaired $t$ test $F_{(11, 9)} = 1.223$, $P = 0.798$ for midbrain) were unchanged. However, we found a significant increase of the extracellular levels of 5-HT in the striatum ($0.14 \pm 0.032$ ng/mL in Syt1$^{-/-}$ vs $0.057 \pm 0.006$ ng/mL in Syt1$^{+/+}$, Welch's $t$ test, $F_{(11, 7)} = 34.72$, $P = 0.025$), and although a similar tendency was noted in the mesencephalon, the effect did not reach significance ($0.981 \pm 0.009$ in Syt1$^{-/-}$ vs $0.500 \pm 0.031$ ng/mL in Syt1$^{+/+}$, $F_{(11,8)} = 4.377$, $P = 0.055$ for midbrain).

Together, these results indicate that the pool of releasable DA is not decreased by loss of Syt1 and that these mice can maintain intact and sufficient levels of extracellular DA to maintain a basal tone of D1 and D2 receptor activation.

### Increased D2 autoreceptor and DAT function in Syt1 cKO$^{DA}$ mice

Our behavioral results with D2 receptor ligands suggest that adaptations are likely to occur in the DA system of Syt1 cKO mice, in a manner that is similar to adaptations occurring at early stages of DA neuron degeneration in PD[46,47]. In particular, D2 receptors can be hypothesized to undergo an increased density and/or affinity.

As a first step to examine underlying adaptations to the DA system in Syt1 cKO$^{DA}$ mice, we used FSCV to probe the functionality of D2 autoreceptors and of the membrane DA transporter (DAT) in the dorsal and ventral striatum. We measured the impact on DA release of the D2 receptor agonist quinpirole ($1 \mu M$). In line with previous work showing that D2 autoreceptors negatively regulate DA release[48,49], DA overflow evoked by single electrical pulses was decreased by ≈85% in the dorsal striatum ($1.38 \pm 0.18 \mu M$ pre-treatment vs $0.21 \pm 0.07 \mu M$ after treatment, $n = 9$ (5 M/4 F); Welch's $t$ test, $F_{(8, 8)} = 7.4$, $P = 0.0001$) and by ≈84% in the ventral striatum ($1.22 \pm 0.1 \mu M$ pre-treatment vs $0.196 \pm 0.06 \mu M$ after treatment, $n = 9$ (4 M/4 F); unpaired $t$ test, $F_{(8, 8)} = 2.79$, $P < 0.0001$) after a 15 min perfusion of quinpirole (Fig. 6A, B). In Syt1$^{-/-}$ mice, quinpirole caused a complete abolition of detectable DA release in both the dorsal striatum ($0.06 \pm 0.006 \mu M$ pre-treatment, vs $0.00 \pm 0.00 \mu M$ after treatment, $n = 7$ (1 M/6 F); Welch's $t$ test, $F_{(6, 6)} = 117.3$, $P < 0.0001$) and ventral striatum ($0.13 \pm 0.02 \mu M$ pre-treatment vs $0.00 \pm 0.00 \mu M$ after treatment, $n = 8$ (1 M/7 F); Welch's $t$ test, $F_{(7,7)} = 144.4$, $P = 0.0014$). This observation further confirms that the remaining signal detected by FSCV in Syt1$^{-/-}$ mice is indeed DA. The results are also compatible with our observation of an increased relative effect of D2 autoreceptor stimulation on locomotion (Fig. 3) and supports the possibility of an increased sensitivity of D2 receptors in Syt1$^{-/-}$ mice.

We next evaluated the impact of the D2 autoreceptor antagonist sulpiride, previously shown to enhance DA release triggered by pulse trains[50]. DA release was evoked by pulse trains (30 pulses at 10 Hz) to allow a basal level of D2 autoreceptor activation to occur. DA release evoked in this way in the striatum of Syt1$^{+/+}$ mice was robustly

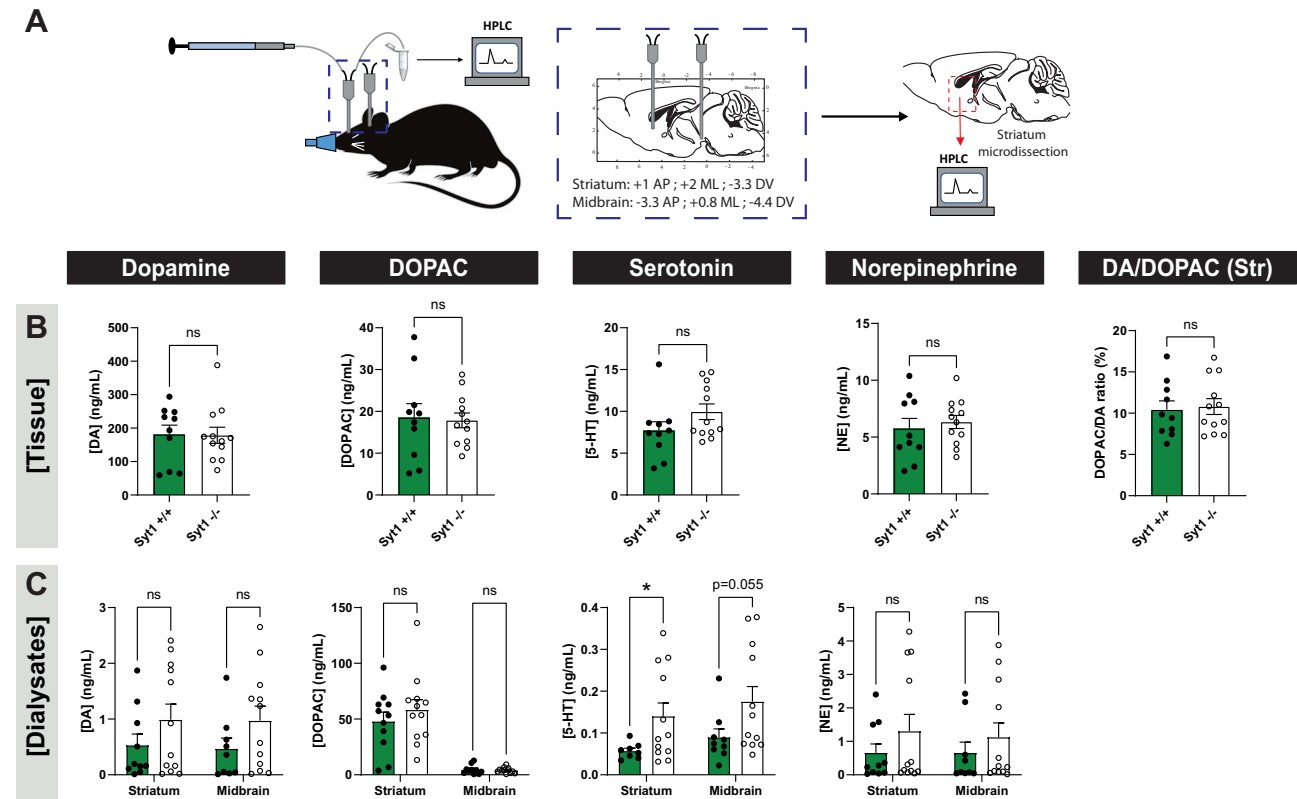

**Fig. 5 | Basal extracellular DA levels and total tissue DA are not altered in Syt1 cKO$^{DA}$ mice. A** Schematic representation of the intracerebral microdialysis protocol performed on anesthetized Syt1 cKO$^{DA}$ and control mice. **B** Quantification of total DA, DOPAC, serotonin and norepinephrine striatal content (ng/mg of total proteins) and DA/DOPAC ratio ($n = 10$ Syt1$^{+/+}$ and 12 Syt1$^{-/-}$). Statistical analysis was carried out by two-tailed unpaired $t$ test. **C** Extracellular quantification (ng/mL) by microdialysis of the same molecules in the dorsal striatum ($n$ (Syt1$^{+/+}$/ Syt1$^{-/-}$ mice) = 10/12 for DA, 11/12 for DOPAC, 8/12 for 5-HT and 10/12 for NE) and in the mesencephalon ($n$ (Syt1$^{+/+}$/ Syt1$^{-/-}$ mice) = 9/12 for DA, 10/12 for DOPAC, 9/12 for 5-HT and 9/12 for NE) of Syt1 cKO$^{DA}$ mice, with an average of three dialysates for each animal. Statistical analysis was carried out by two-way ANOVAs followed by Šidák's tests. Error bars represent ± SEM (ns, non-significant; *$P < 0.05$; **$P < 0.01$; ***$P < 0.001$; ****$P < 0.0001$). Source data are provided as a Source Data file.

increased by sulpiride (5 μM) in both the dorsal striatum (149%) (1.34 ± 0.10 μM pre-treatment vs 2.0 ± 0.19 μM after treatment, $n = 8$ (4 M/4 F); unpaired $t$ test, $F_{(7, 7)} = 3.54$, $P = 0.0079$) and ventral striatum (167%) (1.13 ± 0.08 μM pre-treatment vs 1.89 ± 0.21 μM after treatment, $n = 8$ (4 M/4 F); Welch's $t$ test, $F_{(7, 7)} = 6.9$, $P = 0.0085$) (Fig. 6C, D). Despite a dramatic impairment of DA release evoked by train stimulation in Syt1$^{-/-}$ mice (Supplementary Fig. S1C), the relative effect of sulpiride on evoked DA release was similar in magnitude compared to the wild-type animals in the dorsal striatum, with a +143% increase (0.07 ± 0.01 μM pre-treatment vs 0.09 ± 0.01 μM after treatment, $n = 11$ (4 M/7 F); unpaired $t$ test, $F_{(10, 10)} = 1.8$, $P = 0.0466$). In the ventral striatum, the relative effect of sulpiride on train-evoked DA overflow was higher in Syt1$^{-/-}$ compared to Syt1$^{+/+}$ mice, with a +269% increase (0.13 ± 0.02 μM pre-treatment vs 0.34 ± 0.08 μM after treatment, $n = 11$ (4 M/7 F); Welch's $t$ test, $F_{(10, 10)} = 14.9$, $P = 0.0318$). Together with our results with quinpirole, these findings again argue that D2 autoreceptor function is increased in Syt1$^{-/-}$ mice, especially in the ventral striatum.

Finally, we evaluated the functionality of the DAT by quantifying the impact of the DAT blocker nomifensine (5 μM) on DA overflow evoked by train stimulation, used here to have conditions allowing basal DAT function to be optimally revealed. A near saturating concentration of nomifensine was used to examine the net contribution of DAT activity on phasic DA release[51] As expected, such treatment greatly enhanced DA overflow in wild-type animals, with a +291% increase in the dorsal striatum (1.79 ± 0.19 μM pre-treatment vs 5.15 ± 0.74 μM after treatment, $n = 4$ (1 M/3 F); unpaired $t$ test, $F_{(3, 3)} = 14.35$, $P = 0.0045$) and a +221% increase in the ventral

striatum (1.51 ± 0.2 μM pre-treatment vs 3.34 ± 0.52 μM after treatment, $n = 4$ (1 M/3 F); unpaired $t$ test, $F_{(3, 3)} = 6.62$, $P = 0.0173$) (Fig. 6E, F). The relative effect of nomifensine in Syt1$^{-/-}$ mice was significantly higher than in Syt1$^{+/+}$ mice, with a +706% increase in the dorsal striatum (0.11 ± 0.03 μM pre-treatment vs 0.74 ± 0.22 μM after treatment, $n = 6$ (3 M/3 F); Welch's $t$ test, $F_{(5, 5)} = 64.75$, $P = 0.0367$) and a +588% in the ventral striatum (0.16 ± 0.03 μM pre-treatment vs 0.87 ± 0.13 μM after treatment, $n = 6$ (3 M/3 F); Welch's $t$ test, $F_{(5, 5)} = 24.1$, $P = 0.0023$) (Fig. 6E, F). These results demonstrate that DAT function is intact in Syt1$^{-/-}$ mice. Interestingly, they further suggest that nomifensine may act in some way to rescue part of the impaired activity-dependent DA release in these mice. Although speculative, this could be through its electrogenic activity that directly modulates membrane potential[52].

### Higher D2 receptor density and lower D2 affinity in Syt1 cKO$^{DA}$ mice

Because the maintenance of motor behaviors in Syt1$^{-/-}$ mice could be due in part to adaptations of DA receptors in the striatum, we next used autoradiography to quantify D1 and D2 binding using the selective D1 and D2 receptor radioligands [³H]-SCH23390 and [³H]-raclopride, respectively. No change of Bmax and Kd for D1 binding was observed in the whole striatum ($n = 4$ females Syt1$^{+/+}$ mice and $n = 4$ Syt1$^{-/-}$ (3 M/1 F) mice; unpaired $t$ test, $F_{(3, 3)} = 3.53$, $P = 0.9961$ for Bmax, and $F_{(3, 3)} = 2.80$, $P = 0.466$ for Kd) nor any analyzed striatal region (dorso-lateral, ventrolateral, dorso-medial and ventro-medial), indicating that neither D1 receptor density or affinity for [³H]-SCH23390 was changed in Syt1 cKO$^{DA}$ mice (Fig. 7A, B). This lack of

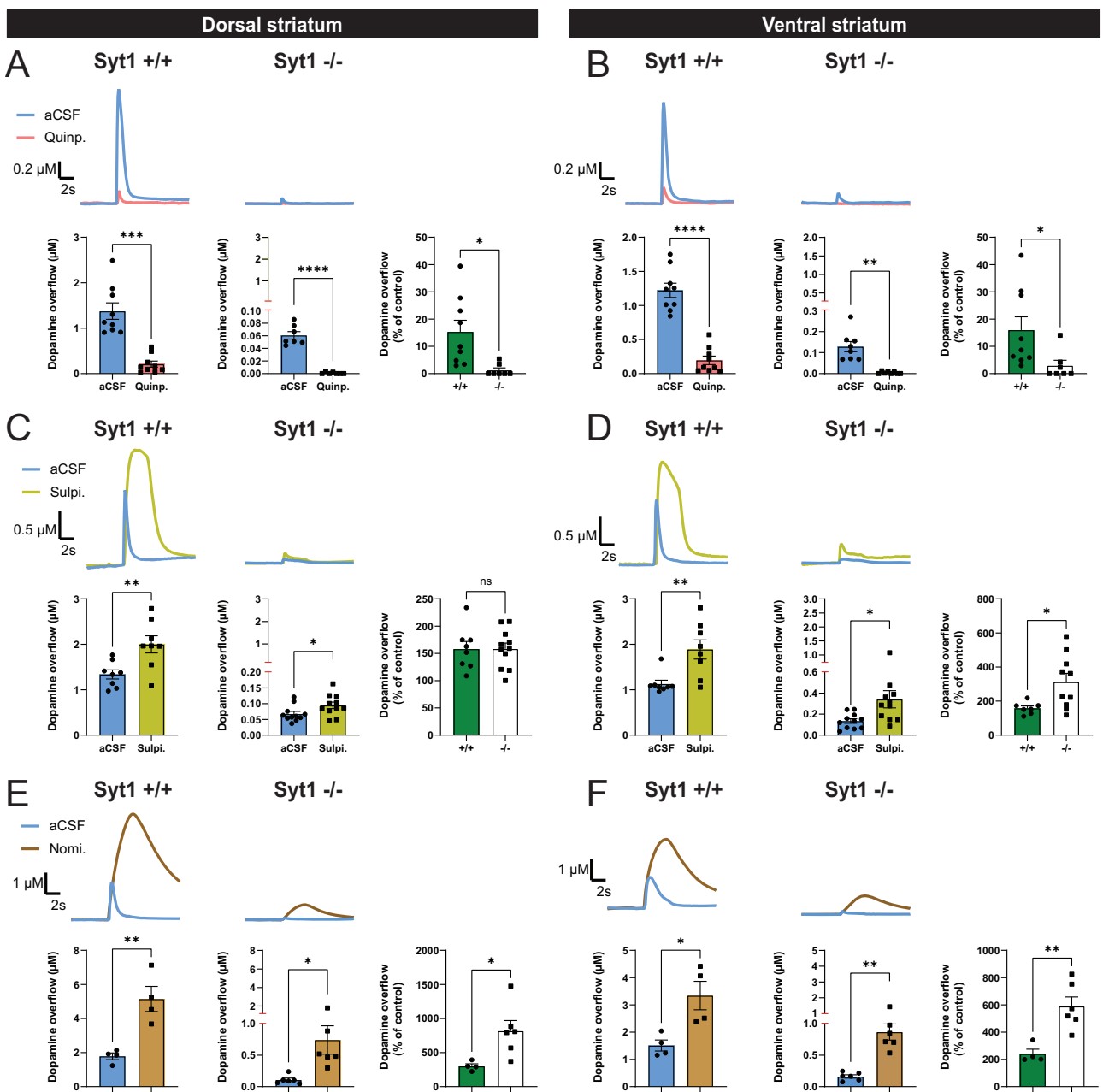

**Fig. 6 | Increased D2 autoreceptor function in Syt1 cKO^DA mice. A** FSCV representative traces (top) and quantification of peak DA overflow amplitude (bottom) obtained in Syt1^{+/+} (n = 9) and Syt1^{−/−} (n = 7) mice with single-pulse electrical stimulation and aCSF containing 1 μM of the D2 agonist quinpirole. **B** Same, but in the ventral striatum (n = 9 Syt1^{+/+} and 8 Syt1^{−/−}). **C** FSCV representative traces (top) and quantification of peak DA overflow amplitude (bottom) obtained in Syt1^{+/+} (n = 8) and Syt1^{−/−} (n = 11) mice with pulse-train stimulation and aCSF containing 5 μM of the D2 antagonist sulpiride. **D** Same, but in the ventral striatum (n = 8 Syt1^{+/+} and 11

Syt1^{−/−}). **E** FSCV representative traces (top) and quantification of peak amplitude (bottom) obtained in Syt1^{+/+} (n = 4) and Syt1^{−/−} (n = 6) mice with pulse-train stimulation and aCSF containing 5 μM of the DAT blocker nomifensine. **F** Same, but in the ventral striatum (n = 4 Syt1^{+/+} and 6 Syt1^{−/−}). For each tested drug, the effect on DA release (% of control) for both genotypes is indicated in the right panel. Statistical analysis was carried out by two-way ANOVAs followed by Šidák's corrections. Error bars represent ± SEM (ns, non-significant; *P < 0.05; **P < 0.01; ***P < 0.001; ****P < 0.0001). Source data are provided as a Source Data file.

change in D1 receptor binding in Syt1 cKO^DA mice is in line with the lack of change in the behavioral response to the D1 receptor antagonist SCH23390 in Syt1^{−/−} mice. However, a significant increase in D2 receptor Bmax and Kd (reduced affinity) was detected in the whole striatum of Syt1^{−/−} mice (n = 5 Syt1^{+/+} (3 M/2 F) and four males Syt1^{−/−}; unpaired t test, $F_{(3, 4)} = 4.65$, P = 0.0002 for Bmax and $F_{(3, 4)} = 6.65$, P = 0.0025 for Kd) (Fig. 7C, D). Finally, we estimated the binding potential of D1 and D2 receptors, which is defined by the Bmax/Kd ratio. We found that binding potential of both D1 and D2 receptors was not significantly different between Syt1^{+/+} and Syt1^{−/−}

mice (unpaired t test, $F_{(3, 3)} = 2.89$, P = 0.2065 for D1 and $F_{(4, 3)} = 1.95$, P = 0.3369 for D2) (Fig. 7E). This suggests that D2 receptor overall binding potential was maintained in Syt1^{−/−} mice. This raises the possibility of a molecular rearrangement, for example implicating desensitization of some of the receptors, helping to maintain homeostasis[53,54]. These results provide further support for the hypothesis that in the absence of phasic DA release, adaptations to DA receptors combined with maintained basal extracellular DA levels are sufficient to support many types of DA-dependent behaviors.

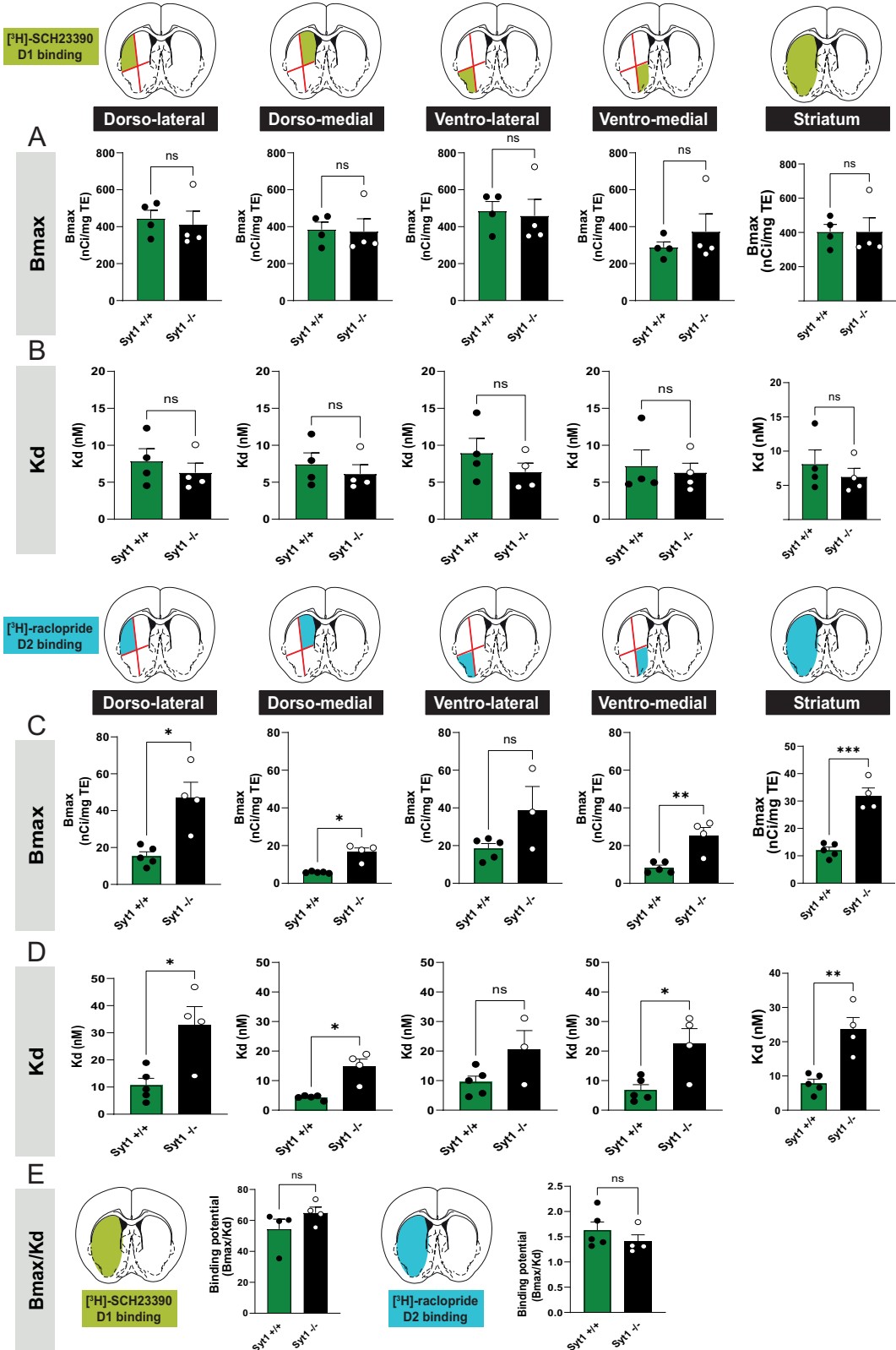

**Fig. 7 | Higher D2 receptor density and lower D2 affinity in Syt1 cKO$^{DA}$ mice.**
**A**, **B** Density of the dopamine D1 receptors (Bmax) (**A**) and affinity (Kd) for the radioligand [$^3$H]-SCH23390 (**B**) in Syt1$^{+/+}$ ($n = 4$) and Syt1$^{-/-}$ ($n = 4$) mice. **C**, **D** Density of the dopamine D2 receptors (Bmax) (**C**) and affinity (Kd) for the radioligand [$^3$H]-raclopride (**D**) in Syt1$^{+/+}$ ($n = 5$) and Syt1$^{-/-}$ mice ($n = 4$). Each binding was analyzed with a series of four serial coronal sections (bregma +1.34 mm to +0.38 mm) and by dividing each section into four quadrants (dorso-lateral, ventrolateral, dorso-

medial, and ventro-medial). **E** Binding potential of D1 and D2 receptors defined as the Bmax/Kd ratio ($n = 4$ Syt1$^{+/+}$/4 Syt1$^{-/-}$ for D1 and 5 Syt1$^{+/+}$/4 Syt1$^{-/-}$ for D2). The data followed a log-normal distribution and two-tailed unpaired $t$ tests were done after the data were converted to a log scale. Error bars represent ± SEM (ns, non-significant; *$P < 0.05$; **$P < 0.01$; ***$P < 0.001$; ****$P < 0.0001$). Source data are provided as a Source Data file.

## Adaptations of DA synthesis and packaging in Syt1 cKO$^{DA}$ mice

Compensatory changes at early stages of DA neuron degeneration in PD include not only receptor adaptations, but also enhanced DA synthesis[46,47] and increased axonal sprouting. The latter has been linked to reduced activation of D2 autoreceptors[55-60]. We therefore hypothesized that reduced activity-dependent DA release in Syt1$^{-/-}$ mice might lead to an increased density of DA neuron terminal markers in the striatum. We next used immunohistochemistry and quantitative confocal microscopy to measure striatal levels of the DA synthesis enzyme TH, the vesicular DA transporter VMAT2 and the DAT. Serotonin (5-HT) innervation was also examined because previous work showed that this system can undergo compensatory sprouting in response to reduced DA levels[61].

We measured signal surface and intensity for these markers in a series of five different striatal slices ranging from bregma +0.98 to bregma −1.06 mm, with a total of 22 different areas for each hemisphere (Supplementary Fig. S4A). We found a global increase of TH signal surface in Syt1$^{-/-}$ mice ($n = 10$, 3 M/7 F) compared to Syt1$^{+/+}$ animals ($n = 10$, 4 M/6 F), both in the dorsal striatum ($135 \pm 6\%$ of control $n = 20$ hemispheres from ten mice; two-way ANOVA with Šidák, $F_{(1, 76)} = 51.84$, $P = 0.0001$) and ventral striatum ($148 \pm 6\%$ of control, $n = 20/10$ mice; two-way ANOVA with Šidák, $F_{(1, 76)} = 51.84$, $P < 0.0001$) (Fig. 8A, B). A global increase of TH signal intensity was also observed, in the dorsal ($117 \pm 3\%$ of control, $n = 20/10$ mice; two-way ANOVA with Šidák, $F_{(1, 76)} = 43.72$, $P = 0.0005$) and ventral striatum ($125 \pm 4\%$ of control, $n = 20/10$ mice; two-way ANOVA with Šidák, $F_{(1, 76)} = 43.72$, $P < 0.0001$) (Fig. 8C). A similar increase in the surface of VMAT2 immunoreactivity ($145 \pm 10\%$ of control, $n = 20/10$ mice; two-way ANOVA with Šidák, $F_{(1, 76)} = 25.74$, $P = 0.0043$) and in the intensity of VMAT2 signal ($123 \pm 5\%$ of control, $n = 20/10$ mice; two-way ANOVA with Šidák, $F_{(1, 76)} = 35.40$, $P = 0.0011$) was observed in the dorsal striatum and in the ventral striatum (signal surface; $157 \pm 10\%$ of control, $n = 20/10$ mice; two-way ANOVA with Šidák, $F_{(1, 76)} = 25.74$, $P = 0.0003$, signal intensity; $130 \pm 5\%$ of control, $n = 20/10$ mice; two-way ANOVA with Šidák, $F_{(1, 76)} = 35.40$, $P < 0.0001$). Overall, these results suggest that in Syt1$^{-/-}$ mice, the total density of dopaminergic axonal processes is increased, possibly explaining the selective increase in amphetamine-induced but not cocaine-induced locomotion in these mice.

Opposite changes in striatal DAT immunostaining were observed in Syt1 cKO$^{DA}$ mice, with a global decrease of signal surface in the dorsal striatum ($62 \pm 6\%$ of control, $n = 20/10$ Syt1$^{-/-}$ and 22/11 Syt1$^{+/+}$; two-way ANOVA with Šidák, $F_{(1, 80)} = 16.24$, $P = 0.0094$) and in the ventral striatum ($64 \pm 7\%$ of control, $n = 20/10$ Syt1$^{-/-}$ and 22/11 Syt1$^{+/+}$; two-way ANOVA with Šidák, $F_{(1, 80)} = 16.24$, $P = 0.0131$) (Fig. 8A–C). The DAT signal intensity appeared unchanged in the ventral striatum but was significantly decreased in the dorsal striatum ($89 \pm 2\%$ of control, $n = 20/10$ Syt1$^{-/-}$ and 22/11 Syt1$^{+/+}$; two-way ANOVA with Šidák, $F_{(1, 80)} = 9.764$, $P = 0.0280$).

These results are compatible with the establishment of compensatory mechanisms acting together to maintain extracellular DA levels in the striatum Syt1 cKO$^{DA}$ mice. These compensatory changes in TH and VMAT2 occurred in the absence of any change in DA neuron number in the VTA, SNc or retrorubral field (RRF) of Syt1$^{-/-}$ mice, as confirmed by stereological counting of TH-positive neurons in these structures (Fig. 8D–E).

Finally, while 5-HT immunoreactive signal intensity in the striatum remained unchanged after Syt1 deletion, a small decrease of the 5-HT signal surface was visible in the dorsal striatum ($72 \pm 8\%$ of control, $n = 20/10$ Syt1$^{-/-}$ and 22/11 Syt1$^{+/+}$; two-way ANOVA with Šidák, $F_{(1, 80)} = 13.58$, $P = 0.0399$) and ventral striatum ($65 \pm 7\%$ of control, $n = 20/10$ Syt1$^{-/-}$ and 22/11 Syt1$^{+/+}$; two-way ANOVA with Šidák, $F_{(1, 80)} = 13.58$, $P = 0.0114$) (Fig. 8A–C). This result suggests a lack of serotoninergic compensatory axonal sprouting in the striatum of Syt1 cKO$^{DA}$ mice.

We also evaluated if these adaptations observed for striatal TH, VMAT2 and DAT also occurred in the SN and VTA of Syt1 cKO$^{DA}$ mice (Supplementary Fig. S5). Using the same analysis in both midbrain regions, we found no change of DAT signal surface (two-way ANOVA with Šidák, $F_{(1, 44)} = 0.7354$, $P = 0.99$ for SNc and $P = 0.47$ for VTA, $n = 12$ hemispheres from 6 Syt1$^{-/-}$ mice (3 M/3 F) and $n = 12$ hemispheres from 6 Syt1$^{+/+}$ mice (1 M/5 F)), but a significant decrease of TH signal surface in the VTA (two-way ANOVA with Šidák, $F_{(1, 44)} = 12$, $P = 0.0156$) and increase of VMAT2 signal surface in the SNc (two-way ANOVA with Šidák, $F_{(1, 44)} = 7.53$, $P = 0.0111$) (Supplementary Fig. S5B). Such changes cannot be attributed to change in DA neurons number in both regions (Fig. 8E). Moreover, no changes of signal intensity (Supplementary Fig. S5C) were detected for all markers, suggesting that the extent of the adaptations seen in the axonal compartment was more limited in the mesencephalon.

Arguing that compensatory adaptations occur gradually in Syt1$^{-/-}$ mice during the postnatal period as a result of abrogation of activity-dependent DA release, we observed no significant changes in the general morphological development of primary postnatal mesencephalic DA neurons isolated from Syt1$^{-/-}$ P0-P3 pups and grown in vitro for 2-weeks (Supplementary Fig. S6). As expected, Syt1 immunoreactivity was absent from the axonal varicosities of these neurons (Supplementary Fig. S6A) ($n = 29$ fields for Syt1$^{-/-}$ and 35 for Syt1$^{+/+}$). A subset of these TH-positive axonal varicosities was also positive for VMAT2, a proportion that was not different in Syt1$^{-/-}$ mice compared to Syt1$^{+/+}$ mice (Supplementary Fig. S6B, S6G, $n = 50$ fields for Syt1$^{-/-}$ and 55 for Syt1$^{+/+}$). Most of these varicosities were nonsynaptic in structure, as revealed by the observation that only a very small subset was in contact with a MAP2-positive dendritic segment independently of mouse genotype (Supplementary Fig. S6C, S6F, $n = 47$ fields for Syt1$^{-/-}$ and 64 for Syt1$^{+/+}$), in line with recent results[9]. The intensity of TH immunoreactivity in these release sites and of VMAT2 in the cell body of DA neurons was also unchanged in Syt1$^{-/-}$ mice compared to Syt1$^{+/+}$ mice (Supplementary Fig. S6H, S6J) (TH; $n = 138$ fields for Syt1$^{-/-}$ and 166 fields for Syt1$^{+/+}$, VMAT2; $n = 17$ Syt1$^{-/-}$ fields and 22 Syt1$^{+/+}$ fields). A small increase in VMAT2 signal in the axonal release sites of Syt1$^{-/-}$ mice was however observed, similar to in vivo observations ($6002 \pm 732$ pixels/field, $n = 43$ for Syt1$^{-/-}$ fields vs $3765 \pm 590$ pixels/field, $n = 45$ for Syt1$^{+/+}$ fields; unpaired $t$ test, $F_{(42,44)} = 1.47$, $P = 0.0191$).

Survival of DA neurons in these cultures over 14 days was not significantly different between genotypes (Supplementary Fig. S6K, $n = 10$ coverslips Syt1$^{+/+}$ and 12 Syt1$^{-/-}$). Finally, the number of axonal branches and their length were unchanged (Supplementary Fig. S6L–S6O). These results argue that, in vitro, the basic development of DA neurons is not impacted by Syt1 deletion.

Finally, we also unveiled compensatory adaptations in adult Syt1 cKO$^{DA}$ mice by quantification of mRNA levels in total microdissected ventral mesencephalon (Supplementary Fig. S7A, $n = 5$ Syt1$^{+/+}$ (1 M/4 F), 5 Syt1$^{+/-}$ (5 M) and 5 Syt1$^{-/-}$ (4 M/1 F) mice). We observed a significant upregulation of *Slc6a3* (DAT) mRNA (1.31-fold, one-way ANOVA with Dunnett, $F_{(2, 12)} = 3.779$, $P = 0.0324$) and of *Syt4* mRNA (1.23-fold, one-way ANOVA with Dunnett, $F_{(2, 12)} = 6.97$, $P = 0.0197$) in Syt1$^{+/-}$ mice and an upregulation of *Syt4* mRNA (1.26-fold, one-way ANOVA with Dunnett, $F_{(2, 12)} = 6.97$, $P = 0.0095$) and *th* mRNA (1.39-fold, one-way ANOVA with Dunnett, $F_{(2, 12)} = 4.771$, $P = 0.0179$) in Syt1$^{-/-}$ mice. It should be noted that these changes of expression could be underestimated as our analysis was performed on whole microdissected ventral midbrains, therefore including non-dopaminergic neurons. This dilution effect is visible for instance in the Syt1 mRNA quantification, which is only reduced by $\approx 17\%$ in Syt1$^{+/-}$ mice ($P = 0.0603$) and by $\approx 49\%$ in Syt1$^{-/-}$ mice ($P < 0.0001$). Adaptations also occurred at protein levels, as measured by Western Blot in total microdissected striatum (Supplementary Fig. S7B–S7F, $n = 10$ Syt1$^{+/+}$ (4 M/6 F), 8 Syt1$^{+/-}$ (5 M/3 F) and 10 Syt1$^{-/-}$ (7 M/3 F) mice), with a

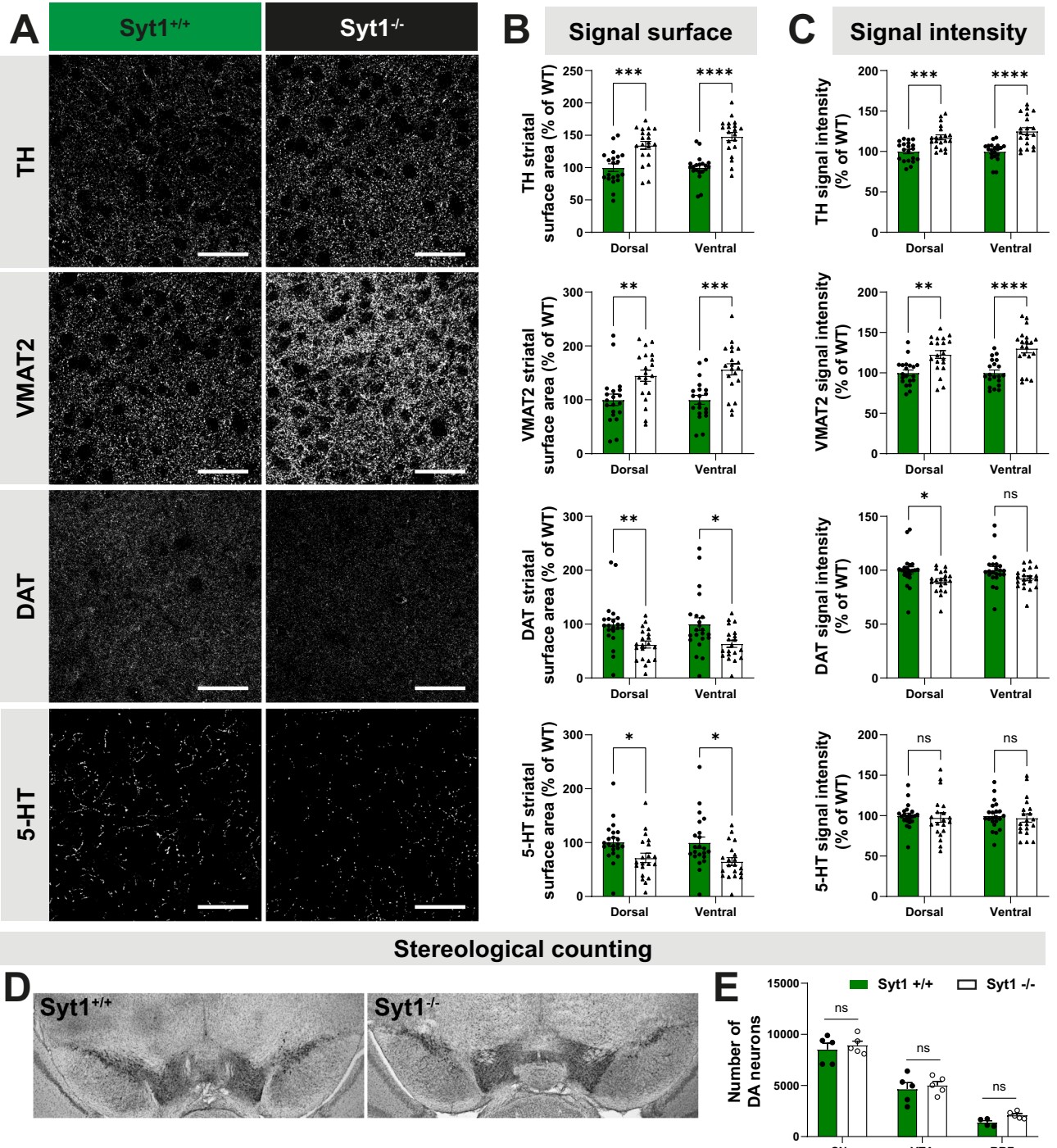

**Fig. 8 | Adaptations of DA innervation in the striatum of Syt1 cKO^DA mice.**
**A** Immunochemistry of striatal slices from 10 to 12-week-old Syt1^+/+ and Syt1^−/− mice (60x confocal) using (from top to bottom): tyrosine hydroxylase (TH), vesicular monoamine transporter 2 (VMAT2), dopamine transporter (DAT) and serotonin (5-HT) immunostainings. Scale bar = 50 μm. **B** Quantification of each signal surface (% of WT) in the dorsal and ventral striatum ($n = 20$ hemispheres/10 mice for both genotypes). **C** Same with signal intensity ($n = 20$ hemispheres/10 mice for both genotypes). **D** TH-immunoreactive cells in the midbrain of Syt1^+/+ and Syt1^−/− mice, identified by the brown DAB staining. **E** Stereological counts of DA neurons in the SNc ($n = 5$ Syt1^+/+/5 Syt1^−/−), VTA ($n = 5/5$), and RRF ($n = 4/5$). Unbiased stereological counting was performed by a blinded observer to estimate the number of DA neurons. Statistical analysis was carried out by two-way ANOVAs followed by Šidák's corrections. Error bars represent ± SEM (ns, non-significant; *$P < 0.05$; **$P < 0.01$; ***$P < 0.001$; ****$P < 0.0001$). Source data are provided as a Source Data file.

significant increase of the VMAT2 in Syt1^+/− mice (1.2-fold; one-way ANOVA with Brown–Forsythe correction and Dunnett T3 post hoc test, $F_{(2,\ 20.13)} = 13.36$, $P = 0.0243$) and in Syt1^−/− mice (1.38, $P = 0.0003$). No change in relative protein levels was observed for TH (one-way ANOVA, $F_{(2,\ 25)} = 0.2591$, $P = 0.774$) and DAT (one-way ANOVA, $F_{(2,\ 25)} = 2.381$, $P = 0.1131$).

## Acute deletion of Syt1 in adult mice impairs DA release but prevents adaptations

Since Syt1 cKO^DA developed strong adaptations, which likely happened during postnatal development, we knocked out Syt1 in adult mice by injecting 6–7 weeks-old Syt1^lox/lox mice with an AAV vector allowing expression of the Cre recombinase in DA neurons (AAV9-TH-cre-myc-

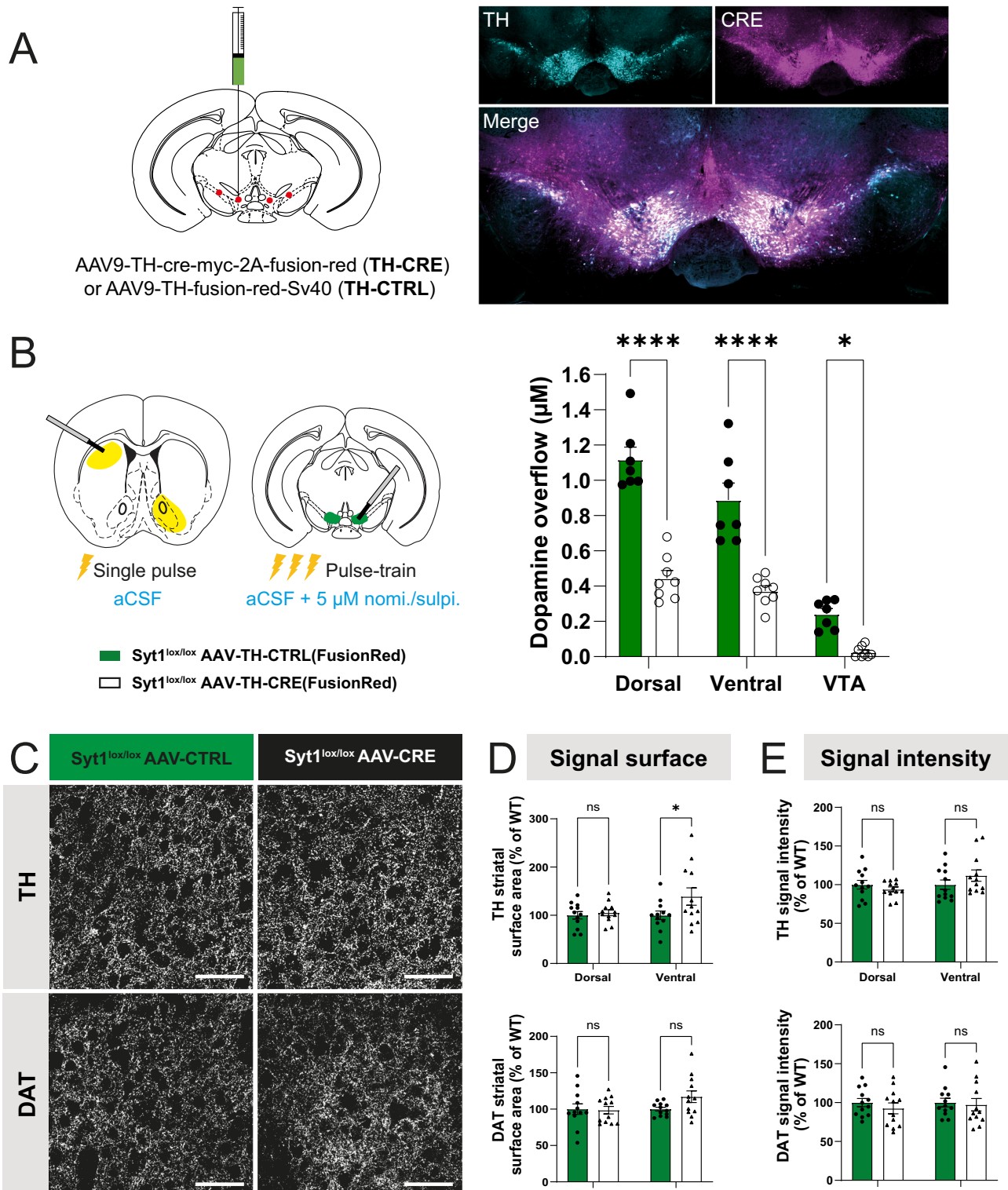

**Fig. 9 | Acute deletion of Syt1 in adult mice impairs DA release but prevents adaptations. A** Stereotaxic injections of AAV9-TH-Cre and AAV9-TH-FusionRed (control) viruses in 6–7-week-old Syt1^lox/lox mice effectively infects the whole ventral mesencephalon (epifluorescence imaging of TH (cyan) and FusionRed (RFP) (Cre, magenta) immunostainings were performed in four mice per condition and one representative image of the infection is represented). **B** FSCV recordings (DA overflow in μM) in the dorsal and ventral striatum (single electrical pulse 1 ms, 400 μA, normal aCSF) and VTA (pulse-train 30p, 1 ms, 400 μA, 10 Hz in presence of 5 μM of nomifensine/sulpiride) from Syt1^lox/lox mice injected with AAV9-TH-Cre

(*n* = 8) and AAV9-TH-Control viruses (*n* = 7). **C** Immunohistochemistry of striatal slices from 9 to 10-week-old Syt1^lox/lox injected with AAV9-TH-Cre (black) and AAV9-TH-control (green) illustrating TH and DAT immunostaining (×60 confocal). Scale bar = 50 μm. **D** Quantification of each signal surface (% of WT) in the dorsal and ventral striatum of injected mice (*n* = 12 hemispheres/6 mice for each condition). **E** Same with signal intensity (*n* = 12 hemispheres/6 mice for each condition). Statistical analysis was carried out by two-way ANOVAs followed by Šidák's corrections. Error bars represent ± SEM (ns, non-significant; *P < 0.05; **P < 0.01; ***P < 0.001; ****P < 0.0001). Source data are provided as a Source Data file.

2A-fusion-red). Three weeks after stereotaxic virus injection in the mesencephalon of these mice, strong expression of the fusion-red reporter was detected in a majority of DA neurons (Fig. 9A). The reporter was also found in a subset of TH-negative neurons at the sites of infection, revealing the preferential but imperfect selectivity of the construct used. FSCV recordings in the striatum and mesencephalon of these mice revealed a robust decrease of DA release evoked by single electrical pulses compared to controls, both in the dorsal ($0.44 \pm 0.04\,\mu M$, $n = 8$ (3 M/5 F) vs $1.12 \pm 0.07\,\mu M$, $n = 7$ (2 M/5 F); two-way ANOVA with Šidák, $F_{(1, 39)} = 120.7$, $P < 0.0001$) and ventral striatum ($0.37 \pm 0.03\,\mu M$, $n = 8$ (3 M/5 F) vs $0.89 \pm 0.1\,\mu M$, $n = 7$ (2 M/5 F); two-way ANOVA with Šidák, $F_{(1, 39)} = 120.7$, $P < 0.0001$). DA release in the VTA of Cre virus-injected mice showed a strong decrease in comparison to control mice injected with the control AAV (CTRL: $0.24 \pm 0.03\,\mu M$, $n = 7$ (2 M/5 F), Cre AAV: $0.026 \pm 0.01\,\mu M$, $n = 8$ (3 M/5 F); two-way ANOVA with Šidák, $F_{(1, 39)} = 120.7$, $P = 0.0171$). This result validates the key role of Syt1 as the main calcium sensor controlling phasic DA release in the adult brain.

Immunohistochemical analysis of TH and DAT in these Syt1 adult KO mice ($n = 12$ hemispheres/6 mice (3 M/3 F) in CTRL and $n = 12/6$ (6 F) in Cre AAV) showed that both global TH and DAT surface and signal intensity were unchanged (two-way ANOVA, $F_{(1, 44)} = 3.919$, $P = 0,0540$ and $F_{(1, 44)} = 0.2478$, $P = 0.6211$ for TH surface and intensity, and $F_{(1, 44)} = 1.645$, $P = 0,2063$ and $F_{(1, 44)} = 0.6035$, $P = 0.4414$ for DAT surface and intensity), except for a slight increase of the TH signal surface in the ventral striatum ($+139 \pm 18\%$ vs control; two-way ANOVA with Šidák, $F_{(1, 44)} = 3.919$, $P = 0.0348$) (Fig. 9C–E). We also evaluated if adaptations could occur at the level of the injection site in the SNc and VTA, but we failed to detect any significant change of TH, VMAT2 and DAT signal surface and intensity (Supplementary Fig. S8). Overall, these results reveal that adult deletion of Syt1 leads to a strong reduction of activity-dependent DA release, accompanied by blunted neuroanatomical adaptations in comparison with conditional Syt1 cKO[DA] mice.

## Discussion

In the present study, we find that Syt1 is the main calcium sensor involved in fast activity-dependent DA release, confirming previous results obtained first in vitro[11], and more recently in vivo[22]. Our work confirms, but also extends, these previous observations in three ways.

First, while previous in vivo work using amperometry concluded that release evoked by single spikes was mostly abrogated in Syt1[-/-] mice, we find that a small component of evoked DA release is in fact maintained and that there was less release remaining in the dorsal ($\approx 5\%$) compared to the ventral striatum ($\approx 12\%$) This remaining release was abolished using the D2 receptor agonist quinpirole. This observation, together with the analysis of voltammograms, confirms the dopaminergic origin of the signal detected by FSCV in Syt1[-/-] mice and strongly suggests the involvement of another calcium sensor mediating fast, activity-dependent release. Syt7 would be a logical candidate because it was previously demonstrated to be highly expressed in DA neurons and localized in both axonal varicosities and in the STD compartment of these neurons[11]. However, Syt7 constitutive KO mice were recently found to have no impairment of single-pulse evoked DA release in the striatum, instead playing a role in STD DA release[23]. This absence of effect at the terminals echoes previous results in GABAergic neurons where Syt7 deletion failed to reduce fast synchronous release, slow asynchronous release, or short-term synaptic plasticity[62]. However, a role of Syt7 in asynchronous release of neurotransmitter was demonstrated using double knockout of Syt1 and Syt7[63,64]. We can thus speculate that a role of Syt7 in gating evoked DA release would be unveiled by a double KO of Syt1 and 7 in DA neurons. Other synaptotagmin isoforms expressed in DA neurons could also be involved in supporting release in the absence of Syt1. We found that Syt4 mRNA is also upregulated in Syt1[-/-] mice. However, this isoform appears to only

play a role in STD DA release in DA neurons[11,23]. Syt11, suggested to participate in regulating DA vesicle pool replenishment[65], could also potentially be involved. Calcium sensors of the Doc2 family are also potential candidates because they have previously been shown to act as modulators of spontaneous synaptic transmission by a calcium-independent mechanism in cortical synapses[66]. However, expression of Syt11 and Doc2b were found to be unchanged in our RT-qPCR analysis of Syt1 cKO[DA] mice. Irrespective of the specific calcium sensor mediating residual evoked DA release in Syt1 cKO[DA] mice, it is possible that this release could also be mobilized through activation of nicotinic acetylcholine receptors located on DA neuron terminals[67], something that should be examined in future work.

Second, results from the current study extend previous work by showing that basal unconditioned motor functions in mice are unimpaired under conditions where most activity-dependent DA release from VTA and SNc DA neurons is blocked. This observation may be surprising considering that axonal and STD DA release are known to play a critical role in motor control[37,38]. Our findings that extracellular levels of DA detected by microdialysis were unchanged in the Syt1 cKO[DA] mice is compatible with previous work showing only minimal changes in similar Syt1 cKO[DA] mice[22]. Our work suggests that basal levels of DA in the striatum, deriving either from spontaneous exocytosis and/or from the very small remaining phasic or desynchronized release, are sufficient to support these basic behaviors and that a large phasic DA signal is not necessary. Additional measurements of basal extracellular DA levels using the more precise no-net-flux method would be useful to extend the present results.

Moreover, we find that Syt1 cKO[DA] mice, despite a dramatic impairment of evoked DA release, produce viable, fertile offspring's, with normal body weight (data not shown) and no gross physical or behavioral abnormalities. However, we observed multiple behavioral, physiological and anatomical phenotypes pointing toward the establishment of functional adaptations that tune the DA system to the existing conditions by maximizing DA production and packaging as well as striatal DA receptor activation.

In open-field experiments, we found that while cocaine-induced locomotion was unchanged in Syt1[-/-] animals, amphetamine-induced locomotion was robustly enhanced. Cocaine is known to increase extracellular DA by blocking reuptake through the DAT, while amphetamine induces a similar elevation by a more complex mechanism including impaired VMAT2-dependent vesicular storage and reverse transport[68]. These results, together with our observation of unchanged tissue DA levels, indicate that while the total pools of DA are not altered in Syt1 KO animals, there are pre- and postsynaptic adaptations taking place after the loss of Syt1. The observed increase in VMAT2 expression in Syt1 KO mice might explain in part the differential impact of loss of Syt1 on amphetamine and cocaine-induced locomotor activation. Increased mobilization of VMAT2-dependent DA stores might allow a higher rate of amphetamine-induced DAT reverse transport. Although limited by the use of single doses, our behavioral results with DA receptor agonists and our autoradiography results suggest that D2 receptors are most affected, with an increased response to the D2 agonist quinpirole and a reduced response to the D2 antagonist raclopride, together with increased D2 receptor binding in the ventrolateral striatum. The D1 receptor system was unchanged in Syt1[-/-] animals, as no differences emerged for locomotion induced by the D1 agonist SCH23390, or for D1 ligand binding. Regarding striatal DA receptors, the absence of effects at D1 receptors and unchanged binding potential at the D2 receptors suggest that compensatory changes were able to maintain a normal balance between direct and indirect pathways, which is consistent with the mild motor phenotype observed in Syt1[-/-] mice. Further work to distinguish changes in D2 autoreceptors from changes in striatal D2 receptors would help to clarify the interpretation. For this, dose-response experiments would be required.

We also find clear presynaptic adaptations illustrated by an increase of TH and VMAT2 signal surface and intensity and a decrease in DAT immunoreactivity in both the ventral and dorsal striatum. One interpretation of the increases in TH and VMAT2 signal is that there may be an increase in the axonal arborization of DA neurons in Syt1[−/−] mice as well as a decrease in reuptake capacity. We and others have previously provided strong support for the hypothesis that DA, through D2 autoreceptors is a key regulator of axon growth by DA neurons[56,60,69]. In the context of a strong reduction in evoked DA release, it is to be expected that D2 autoreceptors would be chronically under-stimulated leading to removal of a brake on axon growth and thus an enhanced density of terminals, in line with our recent work showing axonal arbor expansion after conditional deletion of the D2 receptor gene in DA neurons[60]. Further experiments will be required to demonstrate such an axonal arbor expansion more directly in Syt1[−/−] mice. The increase in TH immunoreactivity, but without a similar increase in total TH protein as seen in western blot, suggests the possibility that there may be more DA neuron release sites, with a lower relative level of TH in each, a model that would be compatible with the lack of change in total tissue DA levels. The decrease in DAT immunoreactivity suggests reduced expression of DAT in the KO mice. Our qPCR results show no significant change in DAT mRNA levels in the KO mice. Western blot also failed to detect a change in DAT protein, although an increase in VMAT2 levels was confirmed with this technique. Although speculative, it is possible that the antibody used for immunohistochemical detection of DAT preferentially recognized cell surface DAT in fixed tissue and that it is specifically cell surface DAT levels that are decreased. Experiments to specifically quantify cell surface DAT levels would be useful to extend our findings. A reduced level of DAT in DA neuron release sites could in theory lead to reduced rates of DA reuptake and thus contribute to maintaining extracellular DA levels in the Syt1 deficient mice. Our FSCV experiments were not able to resolve this as the very small responses remaining in the KO mice precluded a reliable analysis of estimated reuptake kinetics. Quantifications of DAT function in striatal synaptosome preparations might help to resolve this.

Compensatory adaptations to the 5-HT system could also potentially be involved in maintaining performance in some of the behavioral tasks that were examined. While we did not detect morphological evidence for compensatory axonal sprouting of 5-HT axons in the striatum, we did observe an increase of extracellular 5-HT levels in the striatum of the Syt1 deficient mice.

Arguing that compensatory changes in the DA system of Syt1[−/−] mice result from an early and substantial loss of evoked DA release, we found little if any changes in TH and DAT immunoreactivity in adult mice in which Syt1 was deleted by injecting a TH-Cre virus in the ventral mesencephalon. These results indicate that compensatory events are likely to take place during early pre- and postnatal development of the Syt1 KO mice.

A third notable discovery deriving from the present work is the unexpected impairment of activity-dependent DA release in the ventral mesencephalon. This observation is surprising because on the one hand, DA release in this region is thought to derive mostly from the STD compartment of DA neurons, and on the other hand, Syt1 is typically known as a axonal release site-specific protein. In line with this, previous work in primary DA neurons failed to detect Syt1 immunoreactivity in the cell body and dendrites of DA neurons, with the protein detected in the vast majority of nonsynaptic and synaptic terminals along DA neuron axons[9,11]. One possibility is that part of the DA release detected in the VTA and SNc derives from local DA neuron axon collaterals, which would contain Syt1. However, very little solid anatomical data exists currently to support the existence of a large number of such local collaterals, although minimal quantitative data suggest that the VTA contains axonal collaterals arising from its own axons and from SNc DA neurons[27,28]. To address this question more

directly, we devised a strategy to selectively activate the STD compartment of DA neurons using an optogenetic approach. For this, we expressed a Kv2.1-ChR2-YFP fusion construct in DA neurons that we showed was predominantly expressed in the STD compartment of DA neurons. While the expression of this ChR2 protein was purely STD in primary DA neurons, a small contingent of axons in the ventral striatum appeared to contain the protein. As optically evoked DA release in the ventral striatum of these mice was blocked by suppressing firing with TTX, we performed all experiments in ventral mesencephalic slices in the present of TTX, thus limiting any contribution of axonal processes to evoked DA release. We found that STD DA release was still detectable under such conditions and this STD DA release was not reduced by *syt1* gene deletion. Together, these results provide support for the hypothesis that activity-dependent DA release detected in the VTA, and potentially also in the SNc, implicates a mix of STD and axonal DA release. A dual nature of released DA in the mesencephalon was also previously suggested based on an analysis of the calcium-dependency of DA release in the guinea pig VTA[70], although other work has provided support for a similar calcium-dependency of release in mouse striatum and mesencephalon[23,71–75]. Because the available anatomical data suggest that DA containing axonal varicosities are extremely scarce in the SNc and VTA except in the context of compensatory axonal sprouting associated with partial lesions[76], it would be important to revisit this question in the future with a more thorough anatomical investigation. Such anatomical work would also help to clarify the anatomical substrate of the Syt1-dependent DA release detected in the ventral midbrain in the present study and in two recent publications[24,25]. The original optogenetic approach described here in order to trigger STD DA release more selectively should be very useful in the future to further explore the distinct molecular machinery involved in STD and axonal DA release so as to extend recent work showing a role for Syt4 and Syt7 isoforms in this process[11,23].

Considering that electrically evoked DA release in the VTA of Syt1 cKO[DA] mice was reduced by approximately two-thirds, the near-abolition of DA release in the VTA after the adult KO of Syt1 is a puzzling observation. Considering the suboptimal selectivity of the TH-Cre virus used, one obvious possibility is that this could be an artifact due to abrogation of neurotransmitter release from local non-DA neurons that are also stimulated by the extracellular electrodes and that contribute to triggering STD DA release. An approach to resolve this would be to combine this adult KO strategy with the proposed STD optogenetic approach. The use of an inducible DAT-Cre line would also be another possibility to obtain a more selective adult Syt1 KO.

Our results shed new light on the striking resilience of DA-dependent motor functions at early stages of PD, during which an extensive loss of DA in the striatum gradually occurs with substantial motor dysfunctions only appearing after more than 70% of terminals in the striatum are lost[39]. This resilience has been hypothesized to be due to gradually established compensations in the DA system, including increased DA synthesis in the remaining DA axons and D2 receptor upregulation, which has been observed in human tissue and in multiple PD models[4,5,77]. Our work raises the hypothesis that as loss of DA terminals progresses in PD, even if phasic DA release is greatly impaired, as long as a sufficient tone of basal extracellular is maintained, many basic motor functions will be preserved. This may also help explain why even strategies that increase the basal tone of DA without restoring the normal connectivity of DA neurons, such as the implantation of DA-secreting non-neuronal cells can help to reduce motor symptoms in PD animal models[78]. In this context, Syt1 cKO[DA] mice could represent a model to study compensatory mechanisms and sensitivity in PD.

This model also represents a new tool to study spontaneous release mechanisms and to tackle the role of phasic DA release in the brain. Previous studies have demonstrated that although transgenic

mice that are hyperdopaminergic or hypodopaminergic can learn various tasks, their motivation to engage in these tasks is altered. For example, DA-deficient mice appear to be demotivated and will not engage in some tasks[79], while hyperdopaminergic mice perform some tasks faster and with less difficulty, suggesting increased motivation[80]. These results are in line with the well-established role of DA in reward processes, and with its contribution to motivation for palatable foods ("wanting"), rather than in learning to obtain this food[81]. In the context of a dramatic reduction in phasic DA release after deletion of Syt1, we evaluated learning and motivation of Syt1 cKO[DA] mice in an operant food conditioning task (Fig. 4). Although pellets earned from the FR1 and PR1 tasks were their only food source, body weight remained stable across days (data not shown), demonstrating that mice satisfied their entire caloric needs with these tasks. Interestingly, we found no global difference in terms of performance, suggesting mice learned both tasks with the same efficiency and at the same rate (Fig. 4D, G). In both FR1 and PR1 tasks, we noted a 4–6-h window of enhanced nose-poking rates after the onset of the dark cycle. This period could coincide with a circadian peak in extracellular DA levels in the striatum[82], and a peak in activity levels in mice[41]. Interestingly, Ferris et al., by using microdialysis and FSCV in rats across the circadian cycle, demonstrated that the amount of stimulated [DA]o is lowest midway through the dark phase, a time that is characterized by the highest extracellular DA levels. The inverse relationship is exhibited in the light phase, such that when stimulated [DA]o is highest, extracellular DA is at its lowest point. In the PR1 task, we also noticed that the nose-poke activity during the first night was significantly higher in the Syt1 cKO[DA] mice than in control animals (Fig. 4F). Since the number of pellets obtained was not different between the genotypes, it is possible that the KO mice poke more at the beginning of the protocol and once they realize the ratio periodically resets, they start acting thriftier (i.e, fewer pokes to get the same number of pellets). However, poking efficiency and breaking points were not different between genotypes. Adaptations occurring in Syt1 cKO mice at the DA transporter and D2 receptor could possibly play a role in this enhancement in nose poking at the beginning of the dark phases. Ferris and al. demonstrated that D2 autoreceptor sensitivity is regulated by extracellular DA levels and that the DAT is a critical governor of diurnal variation in extracellular DA tone[82]. Overall, our results indicate that phasic DA release is dispensable for baseline motivation to work for food. This result is compatible with observations made in NMDA cKO[DA] mice in which phasic DA release was also impaired; in these mice many DA-dependent behaviors were unaffected[83]. Notably, normal motivation to work for food in a progressive ratio task was observed[83]. Evaluation of other more complex DA-controlled functions would be an important next step to document further the roles of phasic DA release. Our findings are also likely to be relevant to a better understanding of signaling by other neuromodulators such as serotonin and norepinephrine, which show a connectivity pattern that is similar to that of DA neurons.

Our results are also compatible with previous work obtained in mice in which 90% of SNc dopaminergic neurons were genetically ablated, leading to a consequent depletion of >95% of striatal DA[5]. In these mice, motor behaviors of young-adult or aged mice were also largely unaltered. These mutant mice also exhibited an exaggerated response to L-DOPA, suggesting that preservation of motor functions involves sensitization of striatal DA receptors. This result is in keeping with our amphetamine-induced locomotion and autoradiography results, which also point toward the sensitization of DA receptors in Syt1[−/−] mice. The authors concluded that <5% of striatal DA is sufficient to maintain basic motor functions.

In summary, we conclude that although Syt1 is clearly the main calcium sensor controlling DA release, a small component of phasic DA release requires another calcium sensor. We also conclude that Syt1-dependent phasic DA release is dispensable for many basic unconditioned motor functions that require instead only a basal level of extracellular DA, something that could be provided either by (a) spontaneous DA exocytosis, (b) by the small component of phasic DA release that depends on other calcium sensors, (c) by phasic but desynchronized exocytosis or even (d) by DAT-mediated reverse transport. Further work will be required to test these possibilities.

## Methods
Detailed protocols for this manuscript are available on the protocols.io repository (https://doi.org/10.17504/protocols.io.n92ldpoj8l5b/v1).

### Animal model
The Syt1-floxed mouse line (Syt1[lox/lox]) was obtained from Dr. Schneggenburger who rederivated the Syt1[tm1a(EUCOMM)Wtsi] (EMMA, Monterotondo, Italy; EM06829; RRID: RRID:IMSR_EM:06829) strain with a C57Bl6 background carrying a constitutively expressing FLP gene in order to remove the Frt-flanked lacZ/neo insert[84]. We then crossed the Syt1[lox/lox] mice with B6.SJL-Slc6a3[tm1.1(cre)Bkmm]/J; DATIRES-Cre mice (RRID:IMSR_JAX:006660), driving expression of the cre recombinase under the control of the DAT promoter, resulting in the selective deletion of *Syt1* alleles in DA neurons (Fig. 1A) (B6.Cg-Syt1<tm1a(EUCOMM)Wtsi> Slc6a3<tm1.1(cre)Bkmn >/Trudo, RRID:MGI:7488382). Syt1-floxed/DAT[IRES-cre] mice were bred from heterozygous crosses to generate Syt1 conditional knockout in DA neurons (cKO[DA]), heterozygotes and wild-type animals, referred in the manuscript as Syt1[−/−], Syt1[+/−], and Syt1[+/+] respectively. Genotyping for Syt1 cKO[DA] mice was determined using specific primers to target the *Syt1*-floxed alleles - Syt1[lox] forward: GATTCATGATGTCACTGAATCCTATGC and Syt1[lox] reverse CTGGCAAGTAGCTTAGTGAGTC. Experiments were performed blind with regards to animal genotype. Male and female mice were used in this study and their proportion (M/F) is reported for each experiment. However, the number of mice from each sex did not allow to separately analyze males and females in statistical analyses. All procedures involving animals and their care were conducted in accordance with the Guide to care and use of Experimental Animals of the Canadian Council on Animal Care. The experimental protocols were approved by the animal ethics committees of the Université de Montréal (protocols 21–113 and 22–108). For microdialysis experiment, the study was approved by the animal care committee of the research center of the Hôpital du Sacré-Cœur de Montréal (CIUSSS NIM). Housing was at a constant temperature (21 °C) and humidity (60%), under a fixed 12 h light/dark cycle, with food and water available ad libitum.

### Virus injections
In all, 6–7-week-old cKO[DA] or Syt1[lox/lox] mice were anesthetized with isoflurane (Aerrane; Baxter, Deerfield, IL, USA) and fixed on a stereotaxic frame (Stoelting, Wood Dale, IL, USA). A small hole was drilled in the exposed skull and a 10 μl Hamilton syringe connected to a borosilicate injection micropipette pulled using a P-2000 puller (Sutter Instrument) coupled with a Quintessential Stereotaxic Injector (Stoelting) were used for the injections. For virally induced Syt1 KO, a AAV9-TH-cre-myc-2A-fusion-red virus (2.1E+13 vg/ml) and AAV9-TH-fusion-red-Sv40 (1.99E+13 vg/ml) control virus (kindly provided by Dr. James Surmeier) were injected bilaterally to Syt1[lox/lox] mice at the following injection coordinates [AP (anterior−posterior; ML (medial−lateral); DV (dorsal-ventral), from bregma]: AP −2.8 mm; ML + and − 0.9 mm; DV −4.3 mm and AP −3.2 mm; ML + and − 1.5 mm; DV −4.2 mm (0.5 μL per sites at a rate of 0.25 μL/min, for a total of 2 μl per mice), in order to infect neurons in the entire ventral mesencephalon. For optogenetic experiments, a pAAV-syn-ChR2-eYFP-Kv plasmid was purchased (RRID:Addgene_89256, originally from McLean Bolton), modified to insert a double-floxed inverse open reading frame (DIO) and packaged into an AAV2/5 backbone. The AAV2/5-hsyn-DIO-ChR2-eYFP-Kv (6.2 × 10[12] vg/mL, Neurophotonics Core, Québec, Canada)

virus or a AAV5-EF1a-DIO-hChR2(H134R)-eYFP (4.2 × 10^{12} vg/mL, UNC GTC Vector core, USA, RRID:Addgene_35507) control virus were injected with the same protocol as above to infect the entire mesencephalon of Syt1^{+/+} and Syt1^{−/−} mice. Each virus was diluted (1:3) with sterile saline solution and kept on ice prior to injections. Animals recovered in their home cage and were closely monitored for 3 days. The brains were collected for FCSV and immunohistochemistry 3–4 weeks after injections. For FSCV experiment success of the injections was visually validated each time during the slicing of the brains by visualizing the presence of the eYFP reporter. Although expression of the ChR2 constructs was similar in the different mice, small variations in expression level could have occurred and this represents a limitation of these experiments.

### Brain slices preparation

Acute brain slices from 10–12-week-old male or female cKO^{DA} mice were used for all fast-scan cyclic voltammetry (FSCV) recordings. When possible, matched pairs of Syt1^{+/+} and Syt1^{+/−}/Syt1^{−/−} mice were used on each experimental day. The animals were anesthetized with halothane, decapitated and the brain quickly harvested. Next, the brain was submersed in ice-cold oxygenated artificial cerebrospinal fluid (aCSF) containing (in mM): NaCl (125), KCl (2.5), KH_2PO_4 (0.3), NaHCO_3 (26), glucose (10), CaCl_2 (2.4), MgSO_4 (1.3) and coronal striatal and/or midbrain brain slices of 300 μm thickness were prepared with a Leica VT1000S vibrating blade microtome. Once sliced, the tissue was transferred to oxygenated aCSF at room temperature and allowed to recover for at least 1 h. For recordings, slices were placed in a custom-made recording chamber superfused with aCSF at 1 ml/min and maintained at 32 °C with a TC-324B single-channel heater controller (Warner Instruments, USA). All solutions were adjusted at pH 7.35–7.4, 300 mOsm/kg and saturated with 95% O_2–5% CO_2 at least 30 min prior to each experiment.

### Fast-scan cyclic voltammetry

Electrically or optically evoked DA release was measured by FSCV using a 7 μm diameter carbon-fiber electrode placed into the tissue ~100 μm below the surface. A bipolar electrode (Plastics One, Roanoke, VA, USA) or an optical fiber connected to a 470 nm wavelength LED was placed ~200 μm away from the recording site. Carbon-fibers (Goodfellow Cambridge Limited, UK) of 7 μm in diameter were aspirated into ethanol-cleaned glass capillaries (1.2 mm O.D., 0.68 mm I.D., 4 inches long; World Precision Instruments, FL, USA). The glass capillaries were then pulled using a P-2000 micropipette puller (Sutter Instruments, Novato, USA), dipped into 90 °C epoxy for 30 s (Epo-Tek 301, Epoxy Technology, MA, USA) and cleaned in hot acetone for 3 s. The electrodes were heated at 100 °C for 12 h and 150 °C for 5 days. Electrodes were then polished and filled with potassium acetate at 4 M and potassium chloride at 150 mM. The protruding carbon fibers were cut using a scalpel blade under direct visualization to a length allowing to obtain maximal basal currents of 100 to 180 nA. The electrodes were calibrated with 1 μM DA in aCSF before and after each recorded slice and the mean of the current values obtained were used to determine the amount of released DA. After use, electrodes were cleaned with isopropyl alcohol (Bioshop, France). The potential of the carbon-fiber electrode was scanned at a rate of 300 V/s according to a 10 ms triangular voltage wave (−400 to 1000 mV vs Ag/AgCl) with a 100 ms sampling interval, using a CV 203BU headstage preamplifier (Molecular Devices) and a Axopatch 200B amplifier (Molecular Devices, USA). Data were acquired using a Digidata 1440a analog to digital converter board (Molecular Devices, USA) connected to a computer using Clampex and Clampfit (version 10.7, Molecular Devices, USA, pClamp RRID:SCR_011323). Slices were left to stabilize for 20 min before any electrochemical recordings. After positioning of the bipolar stimulation electrode or the optical probe and carbon-fiber electrodes in the tissue, single pulses (400 μA or 30 mW, 1 ms,) or pulses-train (30

pulses at 10 Hz) were applied to the tissue to trigger DA release. For recordings in the VTA and SNc, it is theoretically possible that FSCV signals were partly contaminated by the presence of 5-HT, as previous studies showed that this biogenic amine is present in these regions and can be detected together with DA[85,86]. However, the shape of our cyclic voltammograms suggests that such contamination was minimal or inexistant.

### Behavioral analyses

For the behavioral experiments, different cohorts of mice were used for experiments involving the administration of pharmacological agents. Only mice used for the rotarod and grip strength tests were from the same cohort.

**Grip strength test.** A grip strength apparatus (BioSeb instruments, BIO-GS3, France) was used to evaluate the paw force developed by Syt1 cKO^{DA} mice. In all, 10–12-week-old male or female mice were held firmly by the tail and slowly lowered until their forepaws grasped the middle of the grid. Subjects were then lowered to a horizontal position and pulled following the axis of the sensor until they released their grasp on the grid. The maximum force exhibited by the subject was recorded. The test was repeated 3 times with 30 min resting period between each trial. The results are presented as the means of the 3 trials divided by the body weight of each recorded mouse.

**Pole test.** The test was conducted with a homemade 48-cm metal rod of 1-cm diameter covered with adhesive tape to facilitate traction, placed in a cage. 10–12-week-old male or female Syt1 cKO^{DA} mice were positioned head-up at the top of the pole, and the time required to turn (t-turn) and climb down completely was recorded. The results are presented as the mean of two sessions, with 3 trial per session for each mouse.

**Rotarod.** Motor coordination was evaluated with a rotarod apparatus (Harvard Apparatus, LE8205, USA). 10–12-week-old male or female Syt1 cKO^{DA} mice were pre-trained on the rotarod for two consecutive days to reach a stable performance and then tested on day three. On the first day, mice were placed on the rotarod, rotating at a constant speed of 4 rpm. They remained on the rotarod until either one min had passed, or up to a maximum of three attempts at placing them on the rod. On the second day, mice were trained on the device with an accelerated rotation of 4 to 40 rpm, over a ten-min period. This training was repeated three times with ~30 min of rest between each trial. Mice were tested on the accelerating rod on day 3 and the latency to fall off the device of each tested animal was recorded. The results are presented as the means of the three trials on day 3.

**Open field.** The locomotor behavior of 11–12-week-old male or female Syt1 cKO^{DA} mice was recorded using an infrared actimeter (Superflex sensor version 4.6, Omnitech) using the Fusion software (v5.6 Superflex Edition, RRID:SCR_017972). A chamber partition was used to measure two mice at a time. Subjects were not given time to acclimate and spent a total of 60 min in the chamber with the following protocol: the first 20 min was used to record basal locomotion, while the following 40 min was used to record locomotion after saline or drug administration (i.p.). Drug treatments correspond to a single dose of cocaine hydrochloride at 20 mg/kg (Medisca, cat# 53-21-4, Canada) or D-amphetamine sulfate at 5 mg/kg (Tocris, 2813, UK) at doses known to increase locomotion[87,88]. SCH23390-HCl (Sigma, D-054, Canada) at 50 μg/kg, quinpirole-HCl at 0.2 mg/kg (Sigma, Q-102, Canada) and raclopride L-tartrate at 1 mg/kg (Sigma, R-121, Canada) were also used at selected high doses known to reduce locomotion in the open field[89–91]. Each drug was diluted into 0.9% sodium chloride saline solution (Halyard, #cat 116). For each drug tested, a different cohort of mice was used. Results are presented as the mean of the traveled distance.

**Operant learning.** Appetitively motivated operant learning was assessed using the open-source Feeding Experimentation Device 3 (FED3)[41]. Individually housed animals were placed in a 12 h/12 h light/dark cycle. First, the feeding devices were placed in the home cages under a free feeding paradigm, which automatically dispenses a new pellet each time the mouse removes the pellet in the well. This magazine training was performed for 2 days prior to changing the device mode for a fixed ratio 1 (FR1) paradigm, in which mice learned to nose-poke on the active port to receive a single pellet. To evaluate the operant learning capacity of the mice, nose-poking for pellets across the circadian cycle and nose-poking efficiency (% of total pokes on the active port) were measured by running mice on the FR1 task for 3 consecutive days. After completion of the FR1 task, the device protocol was changed for a 6-days repeating progressive ratio (PR1) task, in which the nose-poking requirement began on FR1 and increased by 1 poke each time a pellet was earned. When mice refrained from poking on either active or inactive port for 30 min, the ratio was reset to FR1. The nose-poking rate and poke efficiency were once again measured during the PR1 task to evaluate the motivation of mice for food across the circadian cycle.

## Dopamine receptor autoradiography (saturation curves)

In all, 10–12-week-old male or female Syt1 cKO$^{DA}$ mice were anesthetized using pentobarbital NaCl solution (7 mg/mL) injected intraperitoneally and then were perfused intracardially with 20 ml of PBS to remove the blood. The brains were extracted and quickly dipped in isopentane at −30 °C for 1 min. Overall, 12-microns-thick coronal sections of the whole striatum were then cut using a cryostat (Leica Biosystem, Model 3050) and mounted on charged microscope slides (X-tra, Leica, Canada). For binding assays, after a pre-wash step in Tris-HCl buffer (15 min) at room temperature, incubation was performed for 1 h at room temperature in a 50 mM Tris-HCl buffer (pH 7.4), containing 120 mM NaC1, 5 mM KC1, 2 mM CaC1$_2$, 1 mM MgC1$_2$, in the presence of seven increasing concentrations (0.15–10 nM) of [$^3$H]-raclopride (specific activity 82.8 Ci/mmol, Perkin Elmer, #NET975250UC, Canada) for D2 binding, 7 increasing concentrations (0.15–10 nM) of [$^3$H]-SCH23390 (specific activity 83.6 Ci/mmol, Perkin Elmer, #NET930025UC, Canada) for D1 binding to generate saturation curves. To determine the nonspecific binding, adjacent sections were incubated respectively with 5 μM of sulpiride (Sigma, cat#S7771, Canada) for D2 binding and 20 nM mianserin hydrochloride (Sigma, cat#M2525, Canada) with 1 μM SCH-39166 hydrobromide (Tocris, cat #2299, Canada) for D1 binding for each radioligand concentrations used. Incubation was terminated by rinsing sections twice for 5 min in ice-cold Tris-HC1, 50 mM (pH 7.4). Sections were then dipped briefly in cold distilled water and dried overnight. The microscope slides with tissue sections were placed in a light-proof X-ray cassette. Autoradiographs were prepared by placing the slide-mounted tissue sections in contact with a tritium-sensitive film (Biomax MR, Kodack, cat#8715187, Canada) for 6 weeks at room temperature. The films were then developed, and autoradiograms analyzed by densitometry. Autoradiograms were acquired using a grayscale digital camera (Model CFW-1612M, Scion Corporation, MD, USA). The precise topography of D1 and D2 receptors in the striatum was determined from autoradiograms of 4 serial coronal sections for each radioligand concentration corresponding to approximately bregma +1.34 mm to 0.38 mm of the Paxinos and Watson mouse brain atlas[92]. In each section, the striatum was arbitrarily divided into 4 quadrants (dorso-lateral, ventrolateral, dorso-medial and ventro-medial). For each quadrant, the mean signal intensities were measured on ImageJ software (ImageJ 1.53 v software, Wayne Rasband, NIH, RRID:SCR_003070) and nonspecific binding was subtracted from all density readings. The corrected optical gray densities of the different brain areas were converted into nCi/mg of tissue equivalent (nCi/mg TE) using a [$^3$H] radioactive

standard disposed on each film (American Radiolabeled Chemicals Inc., Cat# ART 0123 A). The saturation binding curves were then generated and analyzed on GraphPad software (GraphPad Prism, V9, RRID:SCR_002798) using a nonlinear regression with the equation $Y = Bmax*X/(Kd + X)$ (one-site binding model), where Y represents the specific binding values (nCi/mg of tissue equivalent) and X the radioligand concentration to estimate Bmax, the maximum specific binding and Kd, the dissociation constant at equilibrium. The two last parameters were extracted and represent respectively the receptor density and the radioligands affinity for D1 or D2 receptors.

## Tissue preparation and immunohistochemistry

In all, 10–12-week-old male or female Syt1 cKO$^{DA}$ mice were anesthetized using a pentobarbital NaCl solution (7 mg/mL) injected intraperitoneally and then were perfused intracardially with 20 mL of PBS followed by 30 mL of paraformaldehyde (PFA) 4%. The brains were extracted, placed 48 h in PFA followed by 48 h in a 30% sucrose solution and frozen in isopentane at −30 °C for 1 min. In all, 40 microns-thick coronal sections were then cut using a cryostat (Leica CM1800) and placed in antifreeze solution at −20 °C until used. For slice immunostaining, after a PBS wash, the tissue was permeabilized, nonspecific binding sites were blocked, and slices were incubated overnight with a rabbit anti-tyrosine hydroxylase (TH) (1:1000, Millipore Cat# AB152, RRID:AB_390204), a rat anti-DAT (1:2000, Millipore Cat# MAB369, RRID:AB_2190413), a rabbit anti-VMAT2 (1:2000, kindly provided by Dr. G.W. Miller, Columbia University, USA[93]), a rabbit anti-5-HT (1:2000, ImmunoStar Cat # 20080, RRID:AB_572263), a chicken anti-GFP (1:1000, Aves Labs Cat# GFP-1020, RRID:AB_10000240) and/or a rabbit anti-RFP (1:1000, Rockland Cat# 600-401-379, RRID:AB_2209751) antibody. Primary antibodies were subsequently detected with goat anti-rabbit, rat or chicken Alexa Fluor-488–conjugated (A-11008 RRID:AB_143165, A-11006 RRID:AB_2534074, A-11039 RRID:AB_2534096), 546-conjugated (A-11010 RRID:AB_2534077, A-11081 RRID:AB_2534125, A-11040 RRID:AB_2534097) and/or 647-conjugated secondary antibodies (A-21244 RRID: AB_2535812, A-21247 RRID:AB_141778, A-21449 RRID:AB_2535866) (1:500, 2 h incubation; Invitrogen, Canada). Slices were mounted on charged microscope slides (Superfrost/Plus, Fisher Scientific, Canada) and stored at 4 °C prior to image acquisition.

## Confocal imaging

Images were acquired using an Olympus Fluoview FV1000 point-scanning confocal microscope (Olympus FV1000 Confocal Microscope, RRID:SCR_020337) with a 60x oil-immersion objective (NA 1.42). Images acquired using 488 nm, 546 nm and 647 laser excitations were scanned sequentially to prevent nonspecific bleed-through signal. Care was taken to avoid sensor saturation in all images, thus allowing quantification of signal intensity. A background correction was first applied at the same level for every image analyzed before any quantification. This background was determined from the level of signals in striosomes, regions that contain very low levels of DA neuron markers. An ImageJ (1.53 v, National Institutes of Health, Fiji RRID:SCR_002285) macro developed in-house was used to perform all quantifications (10.5281/zenodo.8007760). For the immunohistochemical characterization of brain tissue obtained from Syt1 cKO$^{DA}$ mice or virally induced Syt1 KO mice, grayscale images were converted to binary (a.k.a. halftone or black & white) by defining a grayscale cut-off point. Grayscale values below the cut-off became black and those above became white. The signal contrast/intensity threshold of each image within the same IHC was adjusted and the relative results (% of WT) were reported to account for signal variability due to pooling results from different sets of immunostainings. Then the area covered by the signal and the intensity for each signal were measured with ImageJ in a series of five different striatal sections ranging from bregma +0.98 to bregma −1.06 mm, with a total of 22 different spots for each hemisphere (Supplementary Fig. S4A).

## Stereology

One out of every 6th cryostat section was used for TH immunostaining stereological counting of DA neurons. After a PBS wash, the tissue was incubated for 10 min with 0.9% $H_2O_2$ solution, then washed with PBS again and incubated for 48 h with a rabbit anti-TH antibody (1:1000, Millipore Cat# AB152, RRID:AB_390204) at 4 °C, 12 h with goat anti-rabbit biotin-SP-AffiniPure secondary antibody (1:100, Jackson ImmunoResearch Labs Cat# 111-065-003, RRID:AB_2337959) at 4 °C and 3 h with horseradish peroxidase streptavidin (1:200, Jackson ImmunoResearch Labs Cat# 016-030-084, RRID:AB_2337238). The diaminobenzidine (DAB) reaction was carried out for 45 s, then stopped by incubation with 0.1 M acetate buffer. Slices were mounted on charged microscope slides (Superfrost/Plus, Fisher Scientific, Canada) and left to dry for 96 h after which they were stained with cresyl violet and went through incubations with increasing concentrations of alcohol. After short isopropanol and xylene baths, slides were sealed with Permount mounting medium (SP15-100, Fisher, USA) using glass coverslips. TH-immunoreactive neurons were counted using a ×100 oil-immersion objective on a Leica microscope equipped with a motorized stage. A $60 \times 60\,\mu m^2$ counting frame was used in the Stereo Investigator (MBF Bioscience, RRID:SCR_002526) sampling software with a 12 μm optical dissector (2 μm guard zones) and counting site intervals of 150 μm after a random start (100 μm intervals for unilateral lesion). Mesencephalic dopaminergic nuclei, including the VTA, SNc and RRF were examined. Stereological estimates of the total number of TH-immunoreactive neurons within each nucleus were obtained. The number of TH-negative neurons was also estimated similarly in each region based on cresyl violet staining.

## Primary neuronal co-cultures

For all experiments, postnatal day 0–3 (P0-P3) mice were cryoanesthetized, decapited, and used for co-cultures according to a previously described protocol[57]. Primary DA neurons from VTA or SNc were prepared from Syt1$^{-/-}$ or Syt1$^{+/+}$ pups and co-cultured with ventral striatum and dorsal striatum neurons from Syt1$^{-/-}$ or Syt1$^{+/+}$ pups, respectively. Neurons were seeded (60 000 cells/mL) on a monolayer of cortical astrocytes grown on collagen/poly-ʟ-lysine-coated glass coverslips. All cultures were incubated at 37 °C in 5% $CO_2$ and maintained in 2/3 of Neurobasal, enriched with 1% penicillin/streptomycin, 1% Glutamax, 2% B-27 supplement and 5% fetal bovine serum (Invitrogen, Canada) plus 1/3 of minimum essential medium enriched with 1% penicillin/streptomycin, 1% Glutamax, 20 mM glucose, 1 mM sodium pyruvate and 100 μl of MITO+ serum extender. All primary neuronal co-cultures were used at 14 days in vitro (DIV).

## Immunocytochemistry on cell cultures

Cultures were fixed at 14-DIV with 4% paraformaldehyde (PFA; in PBS, pH 7.4), permeabilized with 0.1% triton X-100 during 20 min, and nonspecific binding sites were blocked with 10% bovine serum albumin during 10 min. Primary antibodies were: mouse anti-TH (1:2000, Sigma-Aldrich Cat# MAB318, RRID:AB_2201528), rabbit anti-TH (1:2000, Millipore Cat# AB152, RRID:AB_390204), rabbit anti-Syt1 (1:1000, Synaptic Systems Cat# 105 103, RRID:AB_11042457), rabbit anti-VMAT2 (1:2000, gift of Dr. Gary Miller, Colombia University) and mouse anti-MAP2 (1:2000, Millipore Cat# MAB3418, RRID:AB_94856). These were subsequently detected using goat anti-mouse or rabbit Alexa Fluor-488-conjugated (A-11001 RRID:AB_2534069, A-11008, RRID:AB_143165), Alexa Fluor-546-conjugated (A-11003 RRID:AB_2534071, A-11010 RRID:AB_2534077), Alexa Fluor-568-conjugated (A-11031 RRID:AB_144696, A-21124 RRID:AB_2535766) and Alexa Fluor-647-conjugated (A-21235 RRID:AB_2535804, A-21244 RRID:AB_2535812) secondary antibodies (1:500, Invitrogen, Canada).

## RT-qPCR

We used RT-qPCR to quantify the amount of mRNA encoding Syt1, 4, 7 and 11, Doc2b, TH, DAT and VMAT2 in brain tissue from P70 Syt1$^{+/+}$ and Syt1$^{-/-}$ mice. The brains were quickly harvested, and the ventral mesencephalon containing the SN/VTA structures were microdissected and homogenized in 500 μL of trizol. Next, RNA extraction was performed using RNAeasy Mini Kit (Qiagen, Canada) according to the manufacturer's instructions. RNA integrity was validated using a Bioanalyzer 2100 (Agilent, RRID:SCR_019715). Total RNA was treated with DNase and reverse transcribed using the Maxima First Strand cDNA synthesis kit with ds DNase (Thermo Fisher Scientific). Gene expression was determined using assays designed with the Universal Probe Library from Roche (www.universalprobelibrary.com). For each qPCR assay, a standard curve was performed to ensure that the efficiency of the assay was between 90 and 110%. Assay information is presented in Supplementary Table 1. The QuantStudio qPCR instrument (Thermo Fisher Scientific, RRID:SCR_023003) was used to detect the amplification level. Relative expression comparison (RQ = $2^{-\Delta\Delta CT}$) was calculated using the Expression Suite software (Thermo Fisher Scientific), using GAPDH as an endogenous control. Efficiency correction was performed to mathematically compensate for differences in amplification efficiency of the targets and endogenous controls when calculating relative quantities.

## Western blot

Striatum samples microdissected from Syl$^{+/+}$, Syt1$^{+/-}$, Syt1$^{-/-}$ adult mice were lysed in RIPA buffer (Fisher Scientific, PI89900) containing a protease inhibitor cocktail (Sigma). Homogenized tissue samples were centrifuged at 12,000×$g$ for 30 min at 4 °C. Supernatant was collected and protein quantification was done with BCA reagent (Thermo Scientific Pierce BCA Protein Assay Kit, PI23227). 20 μg of each sample was separated on 8% SDS-PAGE followed by transfer onto a nitrocellulose membrane. Membrane blocking was done with 10% skimmed milk for 90 min at RT with gentle shaking. The membranes were incubated overnight at 4 °C with gentle shaking with rat anti-DAT (1:1000, Millipore Cat# MAB369, RRID:AB_2190413), rabbit anti-TH (1:1000, Millipore Cat# AB152, RRID:AB_390204), rabbit anti-VMAT2 (1:5000, kindly provided by Dr. G.W. Miller, University, USA[93]) and mouse anti-β-actin-HRP (1:5000, Sigma-Aldrich Cat# A3854, RRID:AB_262011) primary antibodies. Membranes were washed 5 times with TBST buffer for 5 min each time. After this, appropriate secondary antibodies (1:5000) were added and the incubation was done at RT for 90 min with gentle shaking. Membranes were washed again with TBST buffer for five times × 5 min and developed using Clarity Western ECL substrate (Bio-Rad, 1004384863). Images were captured on a Luminescent Image Analyzer (GE Healthcare) using Image quant LAS 4000 software (RRID:SCR_018374). Membranes were stripped and re-probed for β-actin as a loading control. The unprocessed scans of the blots are provided in the Source data file.

## Microdialysis guide cannula implantation

In all, 10–12-weeks-old male and female Syt1 cKO$^{DA}$ mice were anesthetized with sodium isoflurane (2.5% isoflurane at 0.5 L/min oxygen flow), coupled with infiltration analgesia of 1.5 mg/kg lidocaine/bupivacaine 10 min prior to incision of the skull and stereotaxic implantation with a microdialysis guide cannula (CMA 7, Harvard Apparatus) into the left dorsal subdivision of the striatum (coordinates: 1.0 mm anterior of bregma, 2.0 mm lateral of bregma, and −3.3 mm below pia) and the right SNc and VTA (coordinates: −3.3 mm anterior of bregma, −0.8 mm lateral of bregma, and −3.4 mm below pia) following Paxinos coordinates[92]. These guide cannulas were then used to insert the microdialysis probes (6 kDa MW cut-off, CMA 7, Harvard Apparatus) into the target sites. Guide cannulas were secured with acrylic dental cement and an anchor screw was threaded into the cranium. Buprenorphine (0.05 mg/kg, subcutaneously) was used for postoperative

analgesia (once daily for 2 days). Animals were allowed 1-week (housed one per cage) to recover from cannula implantation before dialysis measurements of extracellular DA, 3,4-dihydroxyphenylacetic acid (DOPAC), norepinephrine (NE), and serotonin (5-HT) levels by microdialysis. A removeable obturator was inserted into the cannula to prevent cerebrospinal fluid seepage and infection.

### Microdialysis

Microdialysis probes were calibrated in aCSF containing (in mM): NaCl (126), KCl (3), NaHCO$_3$ (26), NaH$_2$PO$_4$ (3), MgCl$_2$ (1.3), CaCl$_2$ (2.3), and L-ascorbic acid (0.2). In vitro probe recovery ranged from 4 to 30% at a flow rate of 1 μl/min. A computer-controlled microinfusion pump (CMA) was used to deliver perfusate to the probes, and the dialysate was collected from the outlet line. Probes were inserted into the indwelling guide cannulas of the anesthetized animals and perfused with aCSF (flow rate set at 1 μl/min). To minimize the influence of needle trauma on experimental outcomes, dialysates samples were collected during a typical 60 min equilibration period but were discarded and not analyzed. Three samples were then taken at 20 min intervals for 60 min. Each 10 μl dialysate sample was collected in a fraction vial preloaded with 1 μl of 0.25 mol/l perchloric acid to prevent analyte degradation and immediately stored at 4 °C for subsequent analysis. Following microdialysis, mice received an intraperitoneal injection of ketamine/xylazine (120 mg/10 mg/kg) and were intracardially injected with cold saline, decapitated and the brains were harvested. After the microdialysis experiments, the striatum of each mouse was microdissected, and flash frozen in isopentane cooled to −35 °C on dry ice and stored at −80 °C prior to HPLC quantification.

### High-performance liquid chromatography

Extracellular DA, DOPAC, 5-HT and NE concentrations in the left dorsal subdivision of the striatum and the right SNc and ventral VTA were determined with high-performance liquid chromatography (HPLC) using a Dionex pump (Ultimate 3000) coupled with electrochemical detection (EC)[94]. Samples were run through a Luna C18 (2) 75 mm × 4.6 mm 3 μm analytical column at a flow rate of 1.5 ml/min and the electrochemical detector (ESA CoulArray, model #5600 A) was set at a potential of −250 mV and +300 mV. The mobile phase consisted of 6% methanol, 0.341 mM 1-octanesulfoic acid sodium salt, 168.2 mM sodium acetate, 66.6 mM citric acid monohydrate, 0.025 mM ethylenediamine-tetra-acetic acid disodium and 0.71 mM triethylamine adjusted to pH 4.0–4.1 with acetic acid. Using ESA's CoulArray v3.0 software, the position of the peaks for each metabolite was compared to an external standard solution containing 25 ng/ml DA, DOPAC, 5-HT, NE, and 50 mM acetic acid prepared fresh daily from stock solutions and loaded with samples into a refrigerated (10 °C) Dionex RS autosampler (Ultimate 3000). Under these conditions, the retention time for DOPAC, NE, DA, and 5-HT was approximately 2.5 min, 1.1 min, 2.8 min, and 3.76 min, respectively, with a total run time of 22 min/sample. Chromatographic peak analysis was accomplished by identification of unknown peaks in a sample matched according to retention times from known standards using Chromeleon Chromatography Data System (CDS) Software (RRID:SCR_016874). The mean standard ($n = 5$) was used to quantify unknown peaks and to have the same retention time on each run. The analytical curve, as determined with duplicate standard solutions of DOPAC, NE, DA and 5-HT in nanopure water, was in the range of 7–500 ng/ml ($R^2 = 0.9963$, 0.9971, 0.9899, or 0.9775, respectively). The analytical curve was constructed by plotting the area under the curve. For tissular measures, microdissected striatum were homogenized in a solution that contained 45 μl of 0.25 M perchlorate and 15 μl of 2,3-dihydroxybenzoic acid (100 mg/ml) which served as an internal standard. Following centrifugation at 10,000 rpm for 15 min at 4 °C, the supernatant was isolated to detect DA, DOPAC, NE and 5-HT using HPLC-EC as

previously described. In parallel, pellets were reconstituted in 50 μl of 0.1 N NaOH and kept for protein quantification using a BCA protein assay kit (Thermo Scientific Pierce BCA Protein Assay Kit, PI23227). Dialysate concentrations are expressed as ng/ml and tissular concentrations as ng/mg of total proteins.

### Statistical analysis

Statistical analyses were performed with Prism 9 software (GraphPad Prism, V9, RRID:SCR_002798, *$P < 0.05$, **$P < 0.01$, ***$P < 0.001$, ****$P < 0.0001$). Parametric statistical tests were used because samples contained data with normal distributions. Data are presented as mean ± SEM. The level of statistical significance was established at $P < 0.05$ in one or two-way ANOVAs with Brown–Forsythe or Geiser–Greenhouse correction when needed or two-tailed $t$ tests with Welch's correction when needed. A ROUT outlier analysis was performed when required ($Q = 1\%$). Post hoc tests were only performed when the global ANOVA was significant. The Tukey post hoc test was used when all the means were compared to each other and the Sidak post hoc test was used when only subsets of means were compared.

### Reporting summary

Further information on research design is available in the Nature Portfolio Reporting Summary linked to this article.

## Data availability

All key data are included in the manuscript and in the source data file. Access to original raw images and electrochemistry recording traces is available on request from the corresponding author. Source data are provided with this paper.

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

## Acknowledgements

We would like to thank Dr. R. Schneggenburger for kindly providing the Syt1-floxed mouse, Dr. J. Surmeier for kindly providing TH-cre and control viruses, Dr. G. Miller for kindly providing VMAT2 antibody, Dr. A. Gratton for interpretation of HPLC data and M.-E. Delignat–Lavaud for illustrations. This work was funded by the National Sciences and Engineering Research Council of Canada (NSERC, grant RGPIN-2020-05279) to L.-E.T. and by the Krembil Foundation (to L.-E.T.). This research was also funded in whole or in part by Aligning Science Across Parkinson's [ASAP 000525] (to L.-E.T.) through the Michael J. Fox Foundation for Parkinson's Research (MJFF). For the purpose of open access, the author has applied a CC BY public copyright license to all author-accepted manuscripts arising from this submission. B.D.-L. received a graduate student award from Parkinson Canada.

## Author contributions

B.D.-L.: conceptualization and design, acquisition, analysis, validation, and interpretation of the data (voltammetry, behavior, autoradiography, IHC, viral infections and RT-qPCR), drafting or revising the manuscript. J.K.: acquisition and analysis of data (IHC). C.D. and R.D.: acquisition and analysis of data (ICC). I.M.: acquisition and analysis of the data (microdialysis). S.M.: acquisition and analysis of data (western blot). N.G.: acquisition and analysis of the data (stereology and confocal microscopy), revising the manuscript. A.T. acquisition and analysis of the data (stereology), L.M.: acquisition of the data (HPLC). C.L.: acquisition of the data (autoradiography). S.B.: acquisition of the data (ICC). M.-J.B.: acquisition of data (cell culture). P.R.-N.: review and editing. D.L.: review, editing and analysis of data (autoradiography). L.D.B.: review and editing. L.-E.T.: conceptualization and design, resources, funding acquisition, writing— review and editing.

## Competing interests

The authors declare no competing interests.
