## [Peer Review File · Nature Communications]

Synaptotagmin-1-dependent phasic axonal dopamine release
is dispensable for basic motor behaviors in miceREVIEWER COMMENTS

Reviewer #1 (Remarks to the Author):

The calcium sensor synaptotagmin-1 is critical for phasic axonal dopamine release in the striatum and mesencephalon, but is dispensable for basic motor behaviors in mice

In this well-written and well-presented study, Delignat-Lavaut and colleagues have extensively analysed the effect of a conditional KO of Synaptotagmin-1 (Syt1) in dopaminergic midbrain neurons. The authors report an extensive reduction of phasic dopamine release in Syt1 ^{-/-} mice, induced by single electrical pulses, in both the dorsal and the ventral striatum (~95/90%). They conclude that Syt1 acts as the main calcium sensor for fast activity-dependent DA release in the striatum, similar as recently reported (Banerjee et al, 2021, eLife: Synaptotagmin-1 is the Ca²⁺ sensor for fast striatal dopamine release).

Somaotodendritic (STD) dopamine release in response to repetitive stimuli was also reduced in Syt1 ^{-/-} mice (~65%) in the VTA as well as in the SNc. However, the Syt1^{-/-} mice displayed no “parkinsonian” motor phenotype; in contrast in some tests, they performed even better than the +/+ mice. Pharmacological experiments indicated sensitized D2-autoreceptors and DAT-function, in line with an observed increased D2 receptor binding, particularly in the ventro-lateral striatum. Moreover, striatal TH and VMAT2 protein appeared to be elevated, while those of DAT and 5-HT were rather reduced in Syt1 ^{-/-} mice. Midbrain neuron numbers, as well as basal dopamine levels in the dorsal striatum and in the midbrain were apparently not altered in Syt1^{-/-} mice. And in vitro cell culture analyses indicate that the basic development of dopaminergic neurons was also not impacted by Syt1 deletion. Importantly, AAV viral-based DAT-CRE knock down of Syt1 in midbrain neurons from adult mice also resulted in reduced dopamine release in dorsal and ventral striatum, as well as in the VTA, while striatal protein levels of TH and DAT were not changed. The authors conclude that the pool of releasable dopamine is not decreased by loss of Syt1 and that the ^{-/-} mice can uphold sufficient dopamine levels to maintain a basal tone of D1 and D2 receptor activation.

Specific comments/questions.

1. For stimulation with 10 Hz pulse trains I could only find the results for the ^{-/-}, but not the comparison with +/+ mice (Fig. S1D). Was the pulse-train stimulation evoked dopamine-overflow altered in dorsal or ventral striatum in ^{-/-} mice compared to +/+ mice?

2. In Figure 1, only the STD dopamine release into the VTA is shown (Fig. 1D), the SNc data are only in the supplement, although also significantly reduced to a similar extent as in the VTA (Figure S1C). I would suggest moving Figure S1C into the main Figure 1. Similarly, in Fig.7, besides dopamine release in dorsal and ventral striatum, only data for evoked VTA dopamine release are given, the respective data for the SNc are not shown and need to be added.

3. In the combined FSCV with selective optogenetic stimulation experiments (Fig. 2), the dopamine release results are shown for the striatum (not separated into dorsal / ventral), and for the VTA only. Surprisingly, only in the VTA, but not the striatum, a significant reduction in dopamine release was observed in ^{-/-} (ChR2-Kv). However, as this difference (and almost all dopamine release at all) was absent in the TTX-groups, the authors conclude that STD dopamine release in the VTA is independent of Syt1, and they suggest that the observed reduction in ^{-/-} mice is caused by differences in axonal release. Respective Syt1 ^{-/-} (ChR2-Kv) data for the SNc are not given. I would strongly suggest to add these data for the SNc, given that no axonal release is described here, and the authors observed a reduced dopamine release here as well in the Syt1 ^{-/-} mice in Fig S1C.

Could a separate analysis for dorsal and for ventral striatum be performed as well? Given the limited striatal expression, was there still any dopamine release evoked in the dorsal striatum? How did the authors control that the observed differences in dopamine release were not caused by differences in the viral-infection/expression?

4. In Figure 4, for the quinpirole experiments, a single stimulus was given (similar as in Fig 1B/C), for the sulpiride and nomifensine experiments, repetitive stimuli were given to evoke striatal dopamine release. Have the authors tested both paradigms for all three drugs? If so, I would ask to add the full respective data-set, if not, I would suggest to add the rationale behind each chosen paradigm.
5. Did quinpirole, sulpiride, or nomifensine had a differential effect on STD dopamine release in VTA and/or SNc of Syt1^{-/-} compared to ^{+/+} mice, and/or was the D2-expression or maximal binding capacity altered? These data would be very helpful for the reader and should be included, as they would either support or weaken the authors data interpretation, that Syt1 is not important for STD dopamine release.
6. Was the expression or binding capacity of the DAT altered in striatum (or VTA/SNc), in line with the functional changes, shown in Fig 4E/F? As Fig. S4 and Fig. 6 point to reduced DAT and VMAT2 levels - how do the authors explain that the response to cocaine was not altered but only the response to amphetamine (Fig. 3G)?
7. Figure S4: the individual values should be added as dots into the bargraphs, as already done for all other figures.
8. How exactly were the Immunosignals quantified? I guess it is a relative quantification, but I could not find the details in the methods. I would suggest describing this method in more detail.
9. For stereology, only every 6th of each 40µm section was counted. However, the loss of dopamine neurons might not be homogeneous, and particularly small differences in cell numbers are less likely to be detected with such a large sampling interval. Hence, I would recommend counting at least every 2nd section.
10. For RT-qPCR, relative expression comparison ($RQ = 2^{-\Delta\Delta CT}$) was used that does not take into account the individual assay performances. I would recommend to recalculate the values, by taking the standard-curve parameters into account, in particular the individual slopes. Alternatively, these data could be omitted, as even Syt-1 mRNA was only ~25% lower in midbrain-tissue from Syt1^{-/-} mice, hence the results are difficult to interpret, as the authors state themselves.
11. In the very important viral Syt1 KO experiments, striatal and VTA dopamine release was reduced but striatal TH and DAT levels were not altered. Were VMAT2 and/or D2-levels also not altered (protein or mRNA)? Was STD dopamine release in the SNc altered? Were TH, DAT and/or VMAT2, D2 levels in the SNc or the VTA altered?
12. The authors conclude for the general DAT-specific Syt1 ^{-/-}, that the absence of a motor-phenotype in adult mice is due to compensation, but this compensation is not present in the viral-based adult Syt1^{-/-} mice. To proof this hypothesis, results of the respective behavioural experiments of the viral Syt1 ^{-/-} mice should be given as well. Did the authors observe a motor phenotype in this cohort, as expected if their theory is correct?
13. As the n-numbers in general are quite low in these kind of experiments, I would suggest using non-parametric tests over parametric ones, when possible. The authors use parametric tests only. Are the described significances also present when analysed with non-parametrical tests?
14. In some figures, the labelling/lettering should be enlarged for better visibility (e.g Fig. 2C/D upper).

Reviewer #2 (Remarks to the Author):

The authors present a well-thought-through study on the effect of dopaminergic (DA) neurodegeneration on dopamine-dependent motor function. They selectively deleted synaptotagmin-1 (syt1) in DA neurons (KO mice), keeping the somatodendritic dopamine release intact but near-obliterating the axonal dopamine release. I take this opportunity to critically review the autoradiography (AR) study in the present study. The authors use AR to quantify the affinity (K_d) and total density (B_{max}) of D1 and D2 receptors using [3H]SCH23390 and [3H]raclopride in the syt1 KO mice and compare that to normal mice.

In striatal sections, they find no change in the affinity of both the radioligands. They also find no change in D1 receptor density. However, D2 receptor density is significantly increased in the ventrolateral striatum. An increase in D2 receptor density can be seen in the whole striatum, although not statistically significant. They have used gold-standard methods for AR, and I would like to acknowledge their efforts. Although, I have some thoughts and comments.

1) Authors perform saturation assay using an increasing ligand concentration, but only four sections/dilutions (data points) (Line 736) to fit the model. This may not be enough to calculate K_d values accurately. Further, with [3H]raclopride, they only use concentrations between 0 and 5 nM, but if you look at K_d values (Supplementary figure 3), they range between 5 and 15 nM; this is a considerable variation that is seen. The variation is probably due to poor resolution or because higher concentrations (at saturation concentration) are not used. The curves are most probably not at the saturation phase, making the calculation of B_{max} noisy, which further adds more noise to the K_d data. I believe the high variation in K_d with [3H]SCH23390 is also because of the reasons mentioned above. Please justify why these concentrations were chosen. It would be interesting to look at some reference AR images and reference saturation curves with the B_{max} , K_d , and confidence intervals. Minor comment: Please add the exact model used (one site binding or two sites binding). For the purpose of this review, I assume that one site binding was used (because of the model equation shown).

2) Although evident in behavioral data, the increase in D2 B_{max} in the striatum and ventrolateral striatum was driven purely because of two data points, while two data points seem to be similar to control (and may have overlapping standard deviations). Assuming more frozen animal brains are available, I believe more data points will be required to concretely say that there is an increase in D2 receptor density. I agree that performing saturation assays is much more work. Nevertheless, B_{max} assays can be performed on more animals. This means that one could use concentration at 4 to 5 times K_d (average K_d between all animals) and reasonably assume that you are at saturation. At 4-5x K_d , you should be very close to B_{max} , hence only using one concentration and 1 section to get the same results. I doubt if the dopamine receptor affinities will change in the syt1 KO mice. Further addition of 4-5 animals in each group will help make the results more concrete. I firmly believe this will make the results more robust. Nevertheless, I must acknowledge them for performing the assays in the current method.

3) Both the D1 and D2 receptors exist in high- and low-affinity states. If higher concentrations were used for the saturation assay, the K_d and B_{max} for the high- and low-affinity states could be calculated. This would have shown if the increase in receptor density was due to the low-affinity or high-affinity receptors, which are highly relevant. Comment regarding this in the discussion would be interesting.

Minor comments:

1) In Figure 5 as well as Supplementary Figure 3, the bottom right graph does not have the same x-axis values. Please correct that.

2) The unit of "nCi/mg of tissue" is incorrect. Tritium standards have the unit nCi/mg Tissue equivalent (TE), which is not the same as the unit used. I would change it to nCi/mg TE.

3) Following the nomenclature from Innis et al. 2007 "Consensus nomenclature for in vivo imaging of reversibly binding radioligands," the definition of B_{max} should be "density of receptor" or "total density of receptor" (Line 266).

Reviewer #3 (Remarks to the Author):

This report shows not only that synaptotagmin 1 (Syt1) is a key regulator of dopamine (DA) release, at least from striatal DA axons, but that motor behaviors previously shown to require DA signaling are largely intact (and in some cases enhanced) in mice with conditional deletion of Syt1 in DA neurons and their axonal processes (Syt1 cKO mice). There are many strengths in the report, including the range of approaches used to study DA release and DA-related molecules in these mice, and the parallels noted between the maintenance of motor behavior in Syt1 cKO mice and in early Parkinson's disease (PD) in which DA release is also diminished. However, there are three general weaknesses. The first is that a basic role for Syt1 in axonal DA release was reported in 2020 by Banerjee et al. Although the present study demonstrates that a low level of DA release persists in the striatum of Syt1 cKO mice, rather than a complete loss as reported previously, this is an incremental advance. The second weakness is that the paper is long on results, but short on data interpretation. For example, there are interesting observations that are not internally consistent, including an apparent increase in D2 receptor sensitivity seen in behavioral experiments that was not supported by convincing evidence for increased sensitivity in binding studies. Other observations were noted but not interpreted, like improvement seen in rotorod behavior in Syt1 cKO mice that were contrary to expectation for conditions of decreased DA release. Lastly, although the report notes parallels between maintenance of motor behavior in cKO mice and in human patients in early-stage PD, the data presented are observational rather than mechanistic, and so shed limited new light on compensatory mechanisms in PD. Overall, this is a solid and generally thorough report, but one needs better data interpretation (and possibly more experiments) to advance understanding to the extent possible with the tools that the authors have at their disposal. Specific comments and suggestions are given below.

1) The title is misleading as written. A role for Syt1 in DA release is indeed shown, however, the rest of the title implies that Syt1 is not required for motor behavior, whereas it clearly is (e.g., for glutamate release). The point is that Syt1 in DA neurons is not required for motor behavior.

2) How optically evoked DA release is seen in the striatum of ChR2-Kv expressing mice is not clear, given the stated absence of Kv2.1 expression in axons and terminals (113-115), and the absence of eYFP in dorsal striatum (123-126) and limited signal in ventral striatum. One would expect no release in dorsal striatum, and yet release levels for dorsal and ventral striatum remain at 30% of those evoked by conventional ChR expression.

3) 144. Previous studies have shown that somatodendritic DA release is TTX sensitive, so the rationale for using TTX to selectively eliminate axonal contributions to DA release with ChR1-Kv stimulation is not clear. In fact, the observation that release persisted when Na⁺ channels were blocked suggests that suggests that the release observed is non-physiological, possibly from Ca²⁺ entry via ChR2.

4) The proper experiment to test DA receptor sensitivity would be to do a concentration response of the influence on DA release, rather than applying maximal concentration, e.g., for the D2 agonist, quinpirole.

5) The same concern arises for the use of nomifensine to inhibit the DAT. No references are given to indicate where on the concentration response curves the drugs tested fall in slice experiments.

6) 160-161. Not only is there an absence of motor deficits in Syt1 cKO mice, the authors actually observed motor improvement in the rotorod test with a longer latency to fall. This requires further discussion, in that motor improvement is different than simple maintenance of behaviors from compensatory upregulation. Presumably, this improvement is DA-independent, but this was not tested.

7) The autoradiography data do not strongly support the authors' assertion that D2 but not D1 sensitivity is increased. Although a slight increase in the number of D2 binding sites was seen in

ventrolateral striatum, there was no change in the K_d for binding, which is an indicator of sensitivity (Fig. S3). There is also little consideration of what a difference between D2 and D1 upregulation would mean functionally. If D2 receptor sensitivity really is increased in Syt1 cKOs, but D1 sensitivity is not, this would imply a greater influence of DA in opposing movement in the indirect pathway than in motor facilitation via the direct pathway. Of course, there are D2 autoreceptors on DA axons, as well – although increased sensitivity of these would be expected to curtail DA release. Reconciliation of the conflicting data and amplification of what the data mean would increase mechanistic insight of the studies.

8) 217. Stimulus methods should be indicated in the text.

9) 221-244. If there was an abolition of detectable DA release with quinpirole, what do the concentrations reported mean? Also, why is release greater after treatment than before?

10) 255-257. The rationale for suggesting that nomifensine may rescue part of the impaired activity-dependent DA release in Syt1^{-/-} mice is not clear. Indeed, this speculation seems off-base given that DAT-mediated uptake might be expected to be decreased as part of the compensatory mechanism involved in maintenance of DA-dependent motor function in these mice.

11) 277. What is the evidence for maintained extracellular DA?

12) 285, 465, 474. The idea of changes in terminal markers and pre- and post-synaptic changes seems at odds with the authors' recently published work suggesting predominantly non-synaptic release sites in the striatum (Ducrot et al. 2021).

13) 290. What "signal surface" means in the context of studies of biogenic amine markers needs to be explained in the text.

14) 306-307, 309-310. It would make sense to present tissue content data immediately after TH and VMAT2 levels. The increases seen in TH and VMAT imply that content should be increased – but it is not. This argues against the authors' suggestions that these changes contribute to the great effect of amphetamine than cocaine. Moreover, given that the action of amphetamine requires DAT, and this is decreased in the striatum of Syt1 KO mice, the greater effect of amphetamine remains unexplained.

15) 316. Given the suggestion that the changes in TH, DAT, etc might work to maintain extracellular DA, the microdialysis data in Fig. 8 would make sense to present earlier as well. Although it might be sufficient to cite previous literature that already reported (Banerjee et al. 2020).

16) 358-361. The rationale for the observed increase in immunostaining for TH, VMAT2 and DAT, but no change in protein levels for any of these is not convincing.

17) Most experiments indicate that males and females were used. Were any sex differences noted? Were the number of males and females balanced for each study. There seem to be a wide range of data points for all experiments

18) The methods for the behavioral studies should indicate whether a separate cohort was used for each behavioral test or if multiple tests were conducted on a the same animals.

Minor

Significant digits should be consistent throughout (e.g., 231: 1.34 μM \pm 0.1 μM pre-treatment should be either 1.3 \pm 0.1 μM or 1.34 \pm 0.10 μM).

659. Filled not filed.

Reviewer #4 (Remarks to the Author):

The present study developed a conditional knockout (in dopamine neurons) of the calcium sensor Syt1. They show unconditioned motor behaviours are unchanged in the knockout, despite significantly reduced evoked dopamine release (axonal) in the striatum. The authors also suggest their data support unchanged basal dopamine levels in the knockout. The paper is interesting and experimental work is substantial, however, there is a key aspect that reduces reviewer enthusiasm.

The paper does not provide sufficient detail regarding the results of statistical tests. This is particularly obvious in Figure 8 and the accompanying text where the authors suggest that there is no significant difference in basal dopamine (using two techniques) between wildtype and the knockout. However, a cursory calculation of statistical power would suggest these experiments are substantially under-powered to draw any conclusion of basal dopamine (sigma value of ~90 ng/mg from panel B, α of 0.05, 80% power). Without the key results of the statistical tests (F-value, degrees of freedom), the 'no difference in basal dopamine' finding, and 'basal tone is sufficient for basic movement' conclusion is questionable.

Given the significance of basal dopamine to the overall focus of the paper (Abstract line 8, Discussion line 555 & 584), a revision would need to either include sufficient detail of the statistical tests used (e.g. F-value, degrees of freedom) justifying the no change in basal dopamine finding, and/or conduct additional experiments to achieve sufficient statistical power, and/or remove the conclusion that basal dopamine is unchanged in the knockout.

We thank the four reviewers for their constructive comments that helped us improve the manuscript. For some of the requested additional data, this required the production of new mouse cohorts and new experiments, explaining the multiple months required prior to submission of the present revised manuscript. We were not able to perform all suggested additional experiments as this would have required more than one-year full-time effort, but we hope the reviewers will agree that we have significantly strengthened the manuscript and responded positively to most of their suggestions. In addition to all of these suggestions, we have also performed a completely new set of behavioral experiments implicating an operant nose-poking task for food. Considering that Syt1 conditional KO mice have mostly intact basic motor behaviors, we examined a more complex learning task that many would have predicted to depend on phasic dopamine release. We report that performance in such a task is not impaired in the KO mice, thus further making the point that many dopamine-dependent behaviors probably can still be performed with the support of basal tonic levels of dopamine. We believe that this further strengthens the manuscript and will be of even more interest for the field. **Our responses to the comments of the reviewers are in red.**

Responses to the suggestions of the first reviewer:

“In this well-written and well-presented study, Delignat-Lavaut and colleagues have extensively analysed the effect of a conditional KO of Synaptotagmin-1 (Syt1) in dopaminergic midbrain neurons. The authors report an extensive reduction of phasic dopamine release in Syt1 $-/-$ mice, induced by single electrical pulses, in both the dorsal and the ventral striatum (~95/90%). They conclude that Syt1 acts as the main calcium sensor for fast activity-dependent DA release in the striatum, similar as recently reported (Banerjee et al, 2021, eLife: Synaptotagmin-1 is the Ca²⁺ sensor for fast striatal dopamine release).

Somaotodenritic (STD) dopamine release in response to repetitive stimuli was also reduced in Syt1 $-/-$ mice (~65%) in the VTA as well as in the SNc. However, the Syt1 $-/-$ mice displayed no “parkinsonian” motor phenotype; in contrast in some tests, they performed even better than the $+/+$ mice. Pharmacological experiments indicated sensitized D2-autoreceptors and DAT-function, in line with an observed increased D2 receptor binding, particularly in the ventro-lateral striatum. Moreover, striatal TH and VMAT2 protein appeared to be elevated, while those of DAT and 5-HT were rather reduced in Syt1 $-/-$ mice. Midbrain neuron numbers, as well as basal dopamine levels in the dorsal striatum and in the midbrain were apparently not altered in Syt1 $-/-$ mice. And in vitro cell culture analyses indicate that the basic development of dopaminergic neurons was also not impacted by Syt1 deletion. Importantly, AAV viral-based DAT-CRE knock down of Syt1 in midbrain neurons from adult mice also resulted in reduced dopamine release in dorsal and ventral striatum, as well as in the VTA, while striatal protein levels of TH and DAT were not changed. The authors conclude that the pool of releasable dopamine is not decreased by loss of Syt1 and that the $-/-$ mice can uphold sufficient dopamine levels to maintain a basal tone of D1 and D2 receptor activation.”

1. “For stimulation with 10 Hz pulse trains I could only find the results for the $-/-$, but not the comparison with $+/+$ mice (Fig. S1D). Was the pulse-train stimulation evoked dopamine-overflow altered in dorsal or ventral striatum in $-/-$ mice compared to $+/+$ mice?”
 - a. **We revised figure S1 to provide this. The results show that in the cKO mice, train stimulation caused only a modest increase in peak evoked DA overflow, and only in the ventral striatum.**
2. « In Figure 1, only the STD dopamine release into the VTA is shown (Fig. 1D), the SNc data are only in the supplement, although also significantly reduced to a similar extent as in the VTA (Figure S1C). I would suggest moving Figure S1C into the main Figure 1. Similarly, in Fig.7,

besides dopamine release in dorsal and ventral striatum, only data for evoked VTA dopamine release are given, the respective data for the SNc are not shown and need to be added.”

- a. This is now included in the revised figure 1E. We unfortunately could not perform new experiments to also add SNc data to figure 7 for the adult KO experiments because of our inability to obtain a new batch of the same TH-Cre virus used. But we hope that the reviewer will agree that since Fig. 1 shows equivalent effects in the VTA and SNc for the DAT-Cre-based KO, it is to be expected that the same would be seen with the TH-Cre virus approach.
3. « In the combined FSCV with selective optogenetic stimulation experiments (Fig. 2), the dopamine release results are shown for the striatum (not separated into dorsal / ventral), and for the VTA only. Surprisingly, only in the VTA, but not the striatum, a significant reduction in dopamine release was observed in -/- (ChR2-Kv). However, as this difference (and almost all dopamine release at all) was absent in the TTX-groups, the authors conclude that STD dopamine release in the VTA is independent of Syt1, and they suggest that the observed reduction in -/- mice is caused by differences in axonal release. Respective Syt1 -/- (ChR2-Kv) data for the SNc are not given. I would strongly suggest to add these data for the SNc, given that no axonal release is described here, and the authors observed a reduced dopamine release here as well in the Syt1 -/- mice in Fig S1C. Could a separate analysis for dorsal and for ventral striatum be performed as well? Given the limited striatal expression, was there still any dopamine release evoked in the dorsal striatum? How did the authors control that the observed differences in dopamine release were not caused by differences in the viral-infection/expression?”
- a. We understand the reviewer’s interest in seeing data obtained in the SNc in light of beliefs in the field that contrarily to the VTA, the SNc has little if any DA neuron axon collaterals. We considered performing these additional experiments but opted not to, in part because this would have required even more time prior to resubmitting the manuscript, but also and most importantly because we have since been exploring more extensively the axonal domain of both VTA and SNc DA neurons and we see in fact just as many axonal-like collaterals in both regions (the image below shows some dopaminergic axonal varicosities (green) in the SNc of a mouse in which a conditional YFP virus was expressed in a small number of SNc DA neurons; red is Syt1 and blue is MAP2). These data will be included in a separate manuscript examining this question in detail and so, we hope that the reviewer will understand why we chose not to add this here. The reviewer also asked if we could separate data in this part of the figure showing the striatum into ventral and dorsal sectors. We have now done this in the revised figure 2 (panels C and D). As can be seen in panel A, there was very limited expression of the ChR2-Kv construct in the dorsal striatum and this translated into little if any evoked DA release in the dorsal striatum in these slices (revised panel 2C). There was more release detectable in the ventral striatum and this allowed demonstrating the expected decrease in DA release in the Syt1 KO slices in the absence of TTX. Finally, the reviewer also asked if we controlled for possible difference in viral expression across different animals. We did not do this in a formal way, but each mouse was injected at the exact same coordinates with the exact same batch of virus. For each recording, a visual inspection was performed to look at the YFP level and we did not observe any notable difference between the mice.

4. “In Figure 4, for the quinpirole experiments, a single stimulus was given (similar as in Fig 1B/C), for the sulpiride and nomifensine experiments, repetitive stimuli were given to evoke striatal dopamine release. Have the authors tested both paradigms for all three drugs? If so, I would ask to add the full respective data-set, if not, I would suggest to add the rationale behind each chosen paradigm.”
 - a. The reviewer asks why the effect of quinpirole in Fig. 4 (present Fig. 6) was tested with single-pulse stimulation, while the effects of sulpiride and nomifensine were evaluated with train stimulation. The reason is that D2 agonistic effects are easily seen on responses evoked with single pulses. However, it is well established that to see the effect of D2 antagonism, prior activation of the D2 autoreceptor is required, something that is attained during train stimulation. Same applies for DAT blockade, whose functional effects are easier to see with trains, especially for the Syt1 KO condition. This is now explained in the text (page 14).
5. “Did quinpirole, sulpiride, or nomifensine had a differential effect on STD dopamine release in VTA and/or SNc of Syt1^{-/-} compared to ^{+/+} mice, and/or was the D2-expression or maximal binding capacity altered? These data would be very helpful for the reader and should be included, as they would either support or weaken the authors data interpretation, that Syt1 is not important for STD dopamine release”.
 - a. This was not possible to examine because, as stated in the text, all recordings of STD DA release are performed in the presence of sulpiride and nomifensine. We did try to examine the binding of D2Rs in midbrain sections, but the specific signal was much lower compared to the striatum and it proved impossible to obtain a satisfactory signal to noise ratio.
6. “Was the expression or binding capacity of the DAT altered in striatum (or VTA/SNc), in line with the functional changes, shown in Fig 4E/F? As Fig. S4 and Fig. 6 point to reduced DAT and VMAT2 levels - how do the authors explain that the response to cocaine was not altered but only the response to amphetamine (Fig. 3G)?”
 - a. This is a point that we cover in the discussion section of the manuscript (pages 24 and 25). As discussed, we did quantify DAT using western blot and qPCR and failed to see significant changes in the Syt1^{-/-} mice (except for a small increase in DAT mRNA in the Syt1^{+/-} mice). Because we see decreased DAT immunolabelling in immunohistochemistry and an increased proportional effect of DAT blockade with nomifensine on evoked DA overflow, we can only speculate that the increase in amphetamine-induced locomotion could be due to increased VMAT2-dependent DA stocks and a more efficient DAT reverse transport in the Syt1 KO mice. We now allude to this in the revised discussion section (page 24).
7. “Figure S4: the individual values should be added as dots into the bargraphs, as already done for all other figures. “
 - a. We modified figure S4 to add individual data points in the bar graphs, as requested.
8. “How exactly were the Immunosignals quantified? I guess it is a relative quantification, but I could not find the details in the methods. I would suggest describing this method in more detail.”

- a. As requested, we added more information in the revised methods section to better explain how immunohistochemistry material was quantified (page 39).
9. “For stereology, only every 6th of each 40µm section was counted. However, the loss of dopamine neurons might not be homogeneous, and particularly small differences in cell numbers are less likely to be detected with such a large sampling interval. Hence, I would recommend counting at least every 2nd section. »
- a. The procedure that we used, including the sampling interval of 1/6 slice is the present gold standard for unbiased stereological counting. With this sampling interval and the detailed method used, we can confidently estimate the whole population of SNc and VTA DA neurons.
10. “For RT-qPCR, relative expression comparison ($RQ = 2^{-\Delta\Delta CT}$) was used that does not take into account the individual assay performances. I would recommend to recalculate the values, by taking the standard-curve parameters into account, in particular the individual slopes. Alternatively, these data could be omitted, as even *Syt1* mRNA was only ~25% lower in midbrain-tissue from *Syt1*^{-/-} mice, hence the results are difficult to interpret, as the authors state themselves. »
- a. We have now done this and updated the results and the methods section. This led to two minor changes to our conclusions: (1) the increase in *Syt7* mRNA that we had previously seen in the KO mice is now not significant, and (2) the increase in TH mRNA previously seen in both the HET and KO mice is now only significant in the KO mice. The reviewer also pointed out that we only see a ~25% decrease in *Syt1* mRNA in the *Syt1* KO tissue. In fact, this was the decrease observed in the het tissue (now recalculated to 17%). In the KO, the decrease was more than 50% (now recalculated to 49%). This is to be expected considering that a lot of *Syt1* mRNA in such tissue will come from non-DA neurons, for which there was no KO.
11. “In the very important viral *Syt1* KO experiments, striatal and VTA dopamine release was reduced but striatal TH and DAT levels were not altered. Were VMAT2 and/or D2-levels also not altered (protein or mRNA)? Was STD dopamine release in the SNc altered? Were TH, DAT and/or VMAT2, D2 levels in the SNc or the VTA altered? “
- a. We have now added this, and the results are presented in the new supplementary 8 figure. Neither TH, DAT nor VMAT2 levels were significantly changed. Due to time and material limitations, we were however not able to produce new mice to perform again the voltammetry recordings in the SNc, to complement the VTA recordings. But since the decrease in DA release in the dorsal striatum was robust, we hope the reviewer will agree that it is to be expected that DA release in the SNc would also be reduced, like it is in the VTA.
12. “The authors conclude for the general DAT-specific *Syt1*^{-/-}, that the absence of a motor-phenotype in adult mice is due to compensation, but this compensation is not present in the viral-based adult *Syt1*^{-/-} mice. To proof this hypothesis, results of the respective behavioural experiments of the viral *Syt1*^{-/-} mice should be given as well. Did the authors observe a motor phenotype in this cohort, as expected if their theory is correct? »
- a. We would have really liked to be able to do this. Unfortunately, because the TH-Cre virus has a partial off-target expression, this is expected to lead to KO of *Syt1* in a small subset of non-DA neurons, which would obviously greatly complicate the interpretation of any behavioral phenotype. Also, just to be clear, in the paper, we did not conclude in the discussion that the lack of motor phenotype is due to compensations. Rather, we concluded that this was likely to be due to the possibility that many DA-dependent motor tasks depend on tonic, basal levels of DA.

13. “As the n-numbers in general are quite low in these kind of experiments, I would suggest using non-parametric tests over parametric ones, when possible. The authors use parametric tests only. Are the described significances also present when analysed with non-parametrical tests?”
 - a. For each experiment, we did in fact verify that the use of parametric tests was valid. Our “n” values were in the range of 6-10 for most experiments.
14. “In some figures, the labelling/lettering should be enlarged for better visibility (e.g Fig. 2C/D upper).”
 - a. As requested, we increased the size of the lettering in some of the figures to increase legibility.

Responses to the suggestions of the second reviewer:

“The authors present a well-thought-through study on the effect of dopaminergic (DA) neurodegeneration on dopamine-dependent motor function. They selectively deleted synaptotagmin-1 (syt1) in DA neurons (KO mice), keeping the somatodendritic dopamine release intact but near obliterating the axonal dopamine release. I take this opportunity to critically review the autoradiography (AR) study in the present study. The authors use AR to quantify the affinity (Kd) and total density (Bmax) of D1 and D2 receptors using [3H]SCH23390 and [3H]raclopride in the syt1 KO mice and compare that to normal mice.

In striatal sections, they find no change in the affinity of both the radioligands. They also find no change in D1 receptor density. However, D2 receptors density is significantly increased in the ventrolateral striatum. An increase in D2 receptor density can be seen in the whole striatum, although not statistically significant. They have used gold-standard methods for AR, and I would like to acknowledge their efforts. Although, I have some thoughts and comments. »

We thank the reviewer for his/her suggestions to improve the analysis of the autoradiography experiment. We have carried out a new series of binding experiments.

1. “Authors perform saturation assay using an increasing ligand concentration, but only four sections/dilutions (data points) (Line 736) to fit the model. This may not be enough to calculate Kd values accurately. Further, with [3H]raclopride, they only use concentrations between 0 and 5 nM, but if you look at Kd values (Supplementary figure 3), they range between 5 and 15 nM; this is a considerable variation that is seen. The variation is probably due to poor resolution or because higher concentrations (at saturation concentration) are not used. The curves are most probably not at the saturation phase, making the calculation of Bmax noisy, which further adds more noise to the Kd data. I believe the high variation in Kd with [3H]SCH23390 is also because of the reasons mentioned above. Please justify why these concentrations were chosen. It would be interesting to look at some reference AR images and reference saturation curves with the Bmax, Kd, and confidence intervals. Minor comment: Please add the exact model used (one site binding or two sites binding). For the purpose of this review, I assume that one site binding was used (because of the model equation shown). »
 - a. Saturation curves were obtained using 7 concentrations of respective radiolabeled ligands (see a representative example below). We used a one-site binding model to fit our curves. These precisions were added in the revised version of the manuscript.

2. “Although evident in behavioral data, the increase in D2 Bmax in the striatum and ventrolateral striatum was driven purely because of two data points, while two data points seem to be similar to control (and may have overlapping standard deviations). Assuming more frozen animals brains are available, I believe more data points will be required to concretely say that there is an increase in D2 receptor density. I agree that performing saturation assays is much more work. Nevertheless, Bmax assays can be performed on more animals. This means that one could use concentration at 4 to 5 times Kd (average Kd between all animals) and reasonably assume that you are at saturation. At 4-5x Kd, you should be very close to Bmax, hence only using one concentration and 1 section to get the same results. I doubt if the dopamine receptor affinities will change in the syt1 KO mice. Further addition of 4-5 animals in each group will help make the results more concrete. I firmly believe this will make the results more robust. Nevertheless, I must acknowledge them for performing the assays in the current method. »
 - a. We agree with the reviewer, our saturation curves were far from saturation, especially in the Syt1^{-/-} mice group that showed higher Kd values (less affinity). So, we repeated the experiment on a new set of animals and added to the saturation curves a point at 10 nM concentration to be closer to 4-5 times the Kd. This greatly increased the accuracy and reduced the variability of the data of D2 receptor binding parameters. Since we changed the experimental design, we cannot directly pool the raw data (Bmax and Kd values) of the two experiments. We present only the data of the second experiment, which represents more accurate values of binding, since our saturation curves were closer to binding saturation. Note however, that pooling the data of both experiments after transforming the values in percent of changes compared to wild type animals, we obtained similar results (data not presented). We chose to present the data in absolute values (nCi/mg for Bmax and nM for Kd).
3. “Both the D1 and D2 receptors exist in high- and low-affinity states. If higher concentrations were used for the saturation assay, the Kd and Bmax for the high- and low-affinity states could be calculated. This would have shown if the increase in receptor density was due to the low-affinity or high-affinity receptors, which are highly relevant. Comment regarding this in the discussion would be interesting.”
 - a. The generation of saturation curves and estimated Kd values using a two-site binding model gave similar results (as one binding site analysis). Minor points have also been addressed (Bmax are presented in nCi/mg of tissue equivalent (TE)).
4. Minor comments were take care of.

Responses to the suggestions of the third reviewer:

“This report shows not only that synaptotagmin 1 (Syt1) is a key regulator of dopamine (DA) release, at least from striatal DA axons), but that motor behaviors previously shown to require DA signaling are largely intact (and in some cases enhanced) in mice with conditional deletion of Syt1 in DA neurons and their axonal processes (Syt1 cKO mice). There are many strengths in the report, including the range of approaches used to study DA release and DA-related molecules in these mice, and the parallels noted between the maintenance of motor behavior in Syt1 cKO mice and in early Parkinson’s disease (PD) in which DA release is also diminished. However, there are three general weaknesses. The first is that a basic role for Syt1 in axonal DA release was reported in 2020 by Banerjee et al. Although the present study demonstrates that a low level of DA release persists in the striatum of Syt1 cKO mice, rather than a complete loss as reported previously, this is an incremental advance. The second weakness is that the paper is long on results, but short on data interpretation. For example, there are interesting observations that are not internally consistent, including an apparent increase in D2 receptor sensitivity seen in behavioral experiments that was not supported by convincing evidence for increased sensitivity in binding studies. Other observations were noted but not interpreted, like improvement seen in rotorod behavior in Syt1 cKO mice that were contrary to expectation for conditions of decreased DA release. Lastly, although the report notes parallels between maintenance of motor behavior in cKO mice and in human patients in early-stage PD, the data presented are observational rather than mechanistic, and so shed limited new light on compensatory mechanisms in PD. Overall, this is a solid and generally thorough report, but one needs better data interpretation (and possibly more experiments) to advance understanding to the extent possible with the tools that the authors have at their disposal. Specific comments and suggestions are given below.”

1. “The title is misleading as written. A role for Syt1 in DA release is indeed shown, however, the rest of the title implies that Syt1 is not required for motor behavior, whereas it clearly is (e.g., for glutamate release). The point is that Syt1 in DA neurons is not required for motor behavior.”
 - a. We have modified the title, as requested by the reviewer to clarify that we are referring to Syt1 in dopamine neurons only. The new title is: “The calcium sensor synaptotagmin-1 in dopamine neurons is critical for phasic axonal dopamine release in the striatum and mesencephalon but is dispensable for basic motor behaviors in mice.”
2. “How optically evoked DA release is seen in the striatum of ChR2-Kv expressing mice is not clear, given the stated absence of Kv2.1 expression in axons and terminals (113-115), and the absence of eYFP in dorsal striatum (123-126) and limited signal in ventral striatum. One would expect no release in dorsal striatum, and yet release levels for dorsal and ventral striatum remain at 30% of those evoked by conventional ChR expression.”
 - a. We have now revised figure 2 to show separately the release detected in the dorsal and ventral striatum. As the data show, barely no release is detected in dorsal striatum compared to the ventral striatum, where small amounts of the construct can be seen. It seems clear that in a small subset of VTA DA neurons, the ChR2-Kv construct can be delivered to some of the axonal domain.
3. “Previous studies have shown that somatodendritic DA release is TTX sensitive, so the rationale for using TTX to selectively eliminate axonal contributions to DA release with ChR1-Kv stimulation is not clear. In fact, the observation that release persisted when Na⁺ channels were blocked suggests that suggests that the release observed is non-physiological, possibly from Ca²⁺ entry via ChR2.”

- a. We would like to clarify that although some previous work has suggested that a component of STD DA release may occur independently from TTX-dependent firing (Serre et al., 2004, Yee et al., 2019), multiple other authors reported a strong TTX-sensitivity (Cragg and Greenfield, 1997; Beckstead et al., 2004). STD DA release is clearly calcium-dependent, with the calcium arising either from voltage-gated calcium channels (Llinás et al., 1984) or from mobilization of intracellular calcium (Patel et al., 2009). Here, considering the possibility of local DA neuron axon collaterals in the VTA and SNc, our strategy was to block axonal firing with TTX. This approach was successful as can be seen by the fact that the small amount of release detected in the ventral striatum was completely blocked by TTX. The fact that we detected DA release in the VTA even in the presence of TTX is likely due to the fact that the membrane depolarization induced by Chr2 is sufficient to cause the opening of some voltage-gated calcium channels. We agree that we cannot exclude that some calcium could perhaps enter through Chr2 itself. The fact that we failed to detect any release in the ventral striatum in the presence of TTX argues against this possibility. But even if such a thing happened, it still allows us to test our hypothesis that STD DA release triggered in this way is Syt1 independent.
4. “The proper experiment to test DA receptor sensitivity would be to do a concentration response of the influence on DA release, rather than applying maximal concentration, e.g., for the D2 agonist, quinpirole.”
 - a. We agree. However, because this is a rather peripheral point in the paper, we were not able to devote more time to expand this part of the study. But nonetheless, our results showing an increase in the extent of D2 release inhibition by 1 μ M quinpirole are in line with our other results also arguing for adaptations of the D2 receptor in these mice, including an increase in BMax for D2R binding in the autoradiography experiments and our observation of increased effects of quinpirole and sulpiride on locomotor activity in the cKO mice.
5. “The same concern arises for the use of nomifensine to inhibit the DAT. No references are given to indicate where on the concentration response curves the drugs tested fall in slice experiments.”
 - a. We now mention in the revised text that a near saturating concentration of nomifensine was used to examine the net contribution of DAT activity on phasic DA release. Our intent was not to estimate DAT affinity for nomifensine. Importantly, testing multiple doses of nomifensine, quinpirole or supiride would have required close to 100 additional mice and a large amount of work. We hope the reviewer will understand.
6. “160-161. Not only is there an absence of motor deficits in Syt1 cKO mice, the authors actually observed motor improvement in the rotorod test with a longer latency to fall. This requires further discussion, in that motor improvement is different than simple maintenance of behaviors from compensatory upregulation. Presumably, this improvement is DA-independent, but this was not tested. »”
 - a. We agree with the reviewer that the improvement in performance of the Syt1 cKO mice in the rotarod task is intriguing. Analyzing the underlying molecular and physiological mechanisms underlying this increase in performance is however clearly beyond the scope of the present work. We have now added a comment on this in the revised discussion (page 26).
7. “The autoradiography data do not strongly support the authors’ assertion that D2 but not D1 sensitivity is increased. Although a slight increase in the number of D2 binding sites was seen

in ventrolateral striatum, there was no change in the K_d for binding, which is an indicator of sensitivity (Fig. S3). There is also little consideration of what a difference between D2 and D1 upregulation would mean functionally. If D2 receptor sensitivity really is increased in Syt1 cKO, but D1 sensitivity is not, this would imply a greater influence of DA in opposing movement in the indirect pathway than in motor facilitation via the direct pathway. Of course, there are D2 autoreceptors on DA axons, as well – although increased sensitivity of these would be expected to curtail DA release. Reconciliation of the conflicting data and amplification of what the data mean would increase mechanistic insight of the studies. «

- a. The reviewer rightly points out that we need to be careful in the interpretation of the autoradiography results. As described in our response to the second reviewer, we have now performed a new set of acquisitions with the D2 ligand. The new results point toward not only an increase in B_{Max} , but also a reduced K_d . The interpretation is certainly not simple. We have now modified the description of these results and added a point on the interpretation of these results in the revised discussion section (page 24).
8. “217. Stimulus methods should be indicated in the text. »
 - a. As requested, we now specify in the text that electrical stimulation was used for the cyclic voltammetry experiments of figure 6 (top of page 14).
9. “221-244. If there was an abolition of detectable DA release with quinpirole, what do the concentrations reported mean? Also, why is release greater after treatment than before? »
 - a. We would like to point out that in cyclic voltammetry, which is a subtractive technique, a value will always be calculated, even if the signal is close to zero. Such values essentially represent background levels. We have revised the text to clarify and to correct the fact that we had not provided the pre- and post-treatment values for both striatal regions. This also clarifies the fact that DA levels after quinpirole are substantially lower in both regions.
10. “255-257. The rationale for suggesting that nomifensine may rescue part of the impaired activity-dependent DA release in Syt1^{-/-} mice is not clear. Indeed, this speculation seems off-base given that DAT-mediated uptake might be expected to be decreased as part of the compensatory mechanism involved in maintenance of DA-dependent motor function in these mice. «
 - a. We agree that this is very speculative and have now removed this sentence.
11. « 277. What is the evidence for maintained extracellular DA?”
 - a. The reviewer points out that at line 277 of the original manuscript, our mention of maintained extracellular DA levels appeared as unclear. This is certainly because we only presented the microdialysis data later in the manuscript. In the present revised manuscript, we moved the microdialysis section earlier in the sequence of figures (as requested by the present reviewer). This problem therefore does not exist anymore.
12. “285, 465, 474. The idea of changes in terminal markers and pre- and post-synaptic changes seems at odds with the authors’ recently published work suggesting predominantly non-synaptic release sites in the striatum (Ducrot et al. 2021). “
 - a. We agree that is not idea considering the non-synaptic nature of DA neuron connectivity. We have now corrected this throughout the manuscript and refer instead to “terminal” or “striatal” sites of change.
13. “290. What “signal surface” means in the context of studies of biogenic amine markers needs to be explained in the text.”

- a. As requested, we now explain more clearly in the revised methods section what we mean and how we quantified the signal surface of TH, VMAT2 or DAT immunoreactivity (page 39).
14. -16. “306-307, 309-310. It would make sense to present tissue content data immediately after TH and VMAT2 levels. The increases seen in TH and VMAT imply that content should be increased – but it is not. This argues against the authors’ suggestions that these changes contribute to the great effect of amphetamine than cocaine. Moreover, given that the action of amphetamine requires DAT, and this is decreased in the striatum of Syt1 KO mice, the greater effect of amphetamine remains unexplained. » “316. Given the suggestion that the changes in TH, DAT, etc might work to maintain extracellular DA, the microdialysis data in Fig. 8 would make sense to present earlier as well. Although it might be sufficient to cite previous literature that already reported (Banerjee et al. 2020).” “358-361. The rationale for the observed increase in immunostaining for TH, VMAT2 and DAT, but no change in protein levels for any of these is not convincing.”
 - a. As requested, we have moved up earlier in the manuscript the tissue dopamine and microdialysis results. This is now the revised figure 5. With regards to the fact that increased levels of TH or VMAT2 immunoreactivity should correspond to increased tissue levels of dopamine (which we do not see), we agree with the reviewer that this is puzzling. But a possible interpretation is that this change in TH immunoreactivity may correspond not to an increase in the total level of this protein, but rather to an increase in the axonal arbor size of these neurons, secondary to reduced autoreceptor activation (see our 2018 paper in PLOS Genetics). Because we do not see more TH protein in western blot (supplementary figure 7), this could lead to a situation where there are more DA neuron terminals, but each with less relative TH. With regards to VMAT2, we see a corresponding change in immunoreactivity and protein level, with a tendency for increased levels of mRNA as well. We have added a comment on this on page 25 of the revised discussion. In the case of DAT, we have also revised the section raising different interpretations of our findings (page 25).
17. “Most experiments indicate that males and females were used. Were any sex differences noted? Were the number of males and females balanced for each study. There seem to be a wide range of data points for all experiments »
 - a. We did include both male and females and kept an eye open for any clear sex differences. However, we did not see any and the sample sizes were unfortunately underpowered to detect sex differences.
18. « The methods for the behavioral studies should indicate whether a separate cohort was used for each behavioral test or if multiple tests were conducted on a the same animals.”
 - a. As requested, we have clarified how different cohorts of mice were used for the different sets of behavioral experiments (page 36).
19. We have taken care of the two smaller issues raised.

Responses to the suggestions of the fourth reviewer:

“The present study developed a conditional knockout (in dopamine neurons) of the calcium sensor Syt1. They show unconditioned motor behaviours are unchanged in the knockout, despite significantly reduced evoked dopamine release (axonal) in the striatum. The authors also suggest their data support unchanged basal dopamine levels in the knockout. The paper is interesting and experimental work is substantial, however, there is a key aspect that reduces reviewer enthusiasm. »

1. “The paper does not provide sufficient detail regarding the results of statistical tests. This is particularly obvious in Figure 8 and the accompanying text where the authors suggest that there is no significant difference in basal dopamine (using two techniques) between wildtype and the knockout. However, a cursory calculation of statistical power would suggest these experiments are substantially under-powered to draw any conclusion of basal dopamine (sigma value of ~90 ng/mg from panel B, α of 0.05, 80% power). Without the key results of the statistical tests (F-value, degrees of freedom), the ‘no difference in basal dopamine’ finding, and ‘basal tone is sufficient for basic movement’ conclusion is questionable. Given the significance of basal dopamine to the overall focus of the paper (Abstract line 8, Discussion line 555 & 584), a revision would need to either include sufficient detail of the statistical tests used (e.g. F-value, degrees of freedom) justifying the no change in basal dopamine finding, and/or conduct additional experiments to achieve sufficient statistical power, and/or remove the conclusion that basal dopamine is unchanged in the knockout. »
 - a. As requested, we now added the F-values and the degrees of freedom, throughout the manuscript. Otherwise, we agree that the microdialysis results show a lot of variability. But our results reach exactly the same conclusion as a previous study, also showing no change in basal extracellular DA levels in the Syt1 cKO mice (Banerjee et al., 2020).

REVIEWER COMMENTS

Reviewer #2 (Remarks to the Author):

With the addition of the new results and findings, the authors have properly addressed every question surrounding the autodiography investigations. These findings significantly help in strengthening the manuscript.

Reviewer #3 (Remarks to the Author):

This revised paper has been strengthened by addressing several of the reviewers' concerns, albeit arguing against others. A particular strength is the addition of experiments to examine reward behavior in mice with conditional deletion of synaptotagmin 1 in dopamine neurons (Syt1^{-/-}) and in controls (Syt1^{+/-}). As reviewers noted previously, the basic finding of decreased axonal dopamine release in conditional Syt1^{-/-} mice was reported several years ago. The present study extends that by examining behavior in Syt1^{-/-} mice, with the unanticipated finding that motor and reward behaviors tested are largely unimpacted by loss of Syt1 in dopamine neurons. This leads to the very nice conclusions discussed in lines 633-645, including that the minimal behavioral effects of a profound attenuation of phasic dopamine release helps explain the maintenance of motor behavior in Parkinson's patients who have marked loss of dopamine. The findings further suggest that a dopamine tone is more important than a burst of dopamine cell activity and dopamine release for both motor behavior and reward learning. These findings are robust, and therefore allow for reliable interpretation. Unfortunately, these potentially impactful findings and conclusions are lost in a very long and speculative discussion about compensatory changes in dopamine receptor and transporter sensitivity that is based on incomplete, and in some cases inconsistent data from pharmacological and immunohistochemical studies in the report. Additionally, two recent publications report a role for Syt1 in somatodendritic dopamine release from midbrain dopamine neurons. These findings need to be incorporated into the present rationale and discussion, as both argue against the authors' claim that there is no evidence for Syt1 in dopamine neurons. Despite this evidence in the literature, the decrease in dopamine release reported here is discounted, and argued away based on unconvincing optical stimulation experiments. Those data need to be aligned with other recent findings in the literature. These recent papers do not take away from the present work – as long as the present work emphasizes the most solid aspects of the vast array of experimental results reported, as noted above.

1) The authors now use "axon terminals" to indicate dopamine release sites. As this group and others have demonstrated, however, axonal release can occur from release sites all along the axon, so that the idea of a "terminal" per se has little meaning.

2) Based on their previously published results (Ref 11), the authors state that there is no evidence for Syt1 in the somatodendritic compartment of SNc dopamine neuron (p. 6, li. 102-103 and elsewhere). However, recently published functional and anatomical evidence demonstrate the presence of functional SYT1 in this compartment (Hikima et al. 2022; Lebowitz et al. 2023). The background for the experiments reported in Fig. 1 should be updated in light of these published findings. These results along with the data in Fig. 1 D,E also argue against the validity of evidence for the absence of Syt 1 in dopamine neurons (Ref 11), which should be discussed.

3) Interpretation of voltammetric detection of FSCV signals evoked using local electrical stimulation in mouse SNc is complicated by the presence of 5-HT release concurrently with dopamine (John et al. 2006). This is not considered as a confounding factor, but should be.

4) The rationale and results from the optical stimulation experiments using ChR2-Kv in midbrain dopamine neurons remain hard to understand. The authors try to explain that somatodendritic dopamine release can be TTX-insensitive; however, recent studies using modern methods find that stimulated release is action-potential dependent – although spontaneous release under specific conditions is not. Thus, the TTX-insensitive release reported here (Fig. 2E,G,H) is very likely non-

physiological, possibly from depolarization-dependent opening of Ca²⁺ channels by ChR2. Thus, the TTX-insensitive release from the VTA provides little information about the role of Syt1 in the process of somatodendritic release.

5) p. 8, 138-139. Whether the use of this construct sheds light on the role of Syt1 in dorsal striatum is also not obvious. Although release data were TTX-sensitive in dorsal and ventral striatum (pooled in Fig 2F, though separated in 2C and D), in contrast to midbrain, it remains unclear why there was any evoked dopamine release at all, given the authors' rationale that ChR2-Kv should be only in the somatodendritic compartment. The authors go on to say that the small level of presumed dopamine release evoked with in these mice in dorsal striatum did not differ between Syt1^{-/-} mice and WT, "suggesting Syt1 is not required for this form of release in this region." But it is not clear what "this form of release" is.

6) 155. Given that the authors found that somatodendritic dopamine release was lower in Syt1^{-/-} mice than in WT in electrical stimulation experiments, the conclusion that Syt1 is not involved in "selectively triggered" somatodendritic dopamine release seems contrived, as well as not obviously physiologically relevant as noted above.

7) 163-165. Do the authors think the site of action of impaired dopamine release is axonal or somatodendritic or both? This should be expanded.

8) The findings with D1 and D2 receptor drugs seem contradictory at best (Fig. 3). In Parkinson's disease, D1 and D2 agonists are administered to improve mobility, yet here, D1 and D2 agonism cause a marked decrease in locomotion in all mouse genotypes tested. Equally perplexing is that a D2 antagonist (raclopride) also caused a decrease in locomotion. A contributing factor is that injection of saline alone (Fig. 3D) also caused a sharp decrease in locomotor activity in Syt^{+/+} and Syt1^{-/-} mice, which would be a confounding factor in the similar drug studies.

9) Rather than address the bigger picture issues raised by these internally inconsistent data with D1 and D2 drugs, the authors instead identify some subtle differences among the genotypes in the first 5 min after drug administration. They then base conclusions about receptor sensitivity on these small transient differences. For example, data 5 min after administration of quinpirole, a D2 receptors agonist, shows greater motor suppression in Syt1^{-/-} than in Syt1^{+/+} mice. However, this single time point result is not convincing, given the near-complete suppression of movement 5 min later in all 3 genotypes with the single, saturating concentration tested. Better would have to have tested lower drug concentrations that might have revealed a more robust difference, and better yet would have been to have done a dose response curve. Similarly, rather than attempt to explain how both raclopride and quinpirole suppress movement, they again pick the 5 min time point to show a lesser initial effect of D2 antagonism with raclopride in Syt1^{-/-} than in Syt1^{+/+} mice, after which movement is largely absent. This difference is framed as a "significantly increased response in the Syt1^{-/-} mice compared to Syt1^{+/+} mice" – however, this is misleading, as a lesser decrease is not an increase (Fig. 3L).

10) The lack of difference in basal dopamine concentration with microdialysis (Fig. 5) is a key finding in the report. However, baseline determinations are most accurate when conducted using the no-net flux method, as this takes into account changes in uptake activity, etc. The limitations of the method used should be discussed.

11) In the microdialysis studies, [5-HT] was 2-fold higher in Syt1^{-/-} than Syt1^{+/+} mice in both striatum and midbrain. These were not significant, which could reflect under-sampling, given the variability of the data. However, it is not clear why ANOVA was used here. If unpaired t-tests were used to compare [5-HT] in striatum and separately in midbrain, was a difference seen? In any case, a role for 5-HT should be discussed as a possible contributing factor in the lack of effect of Syt1 deletion in dopamine neurons on the various behaviors reported here.

12) The data in Fig. 6 support detection of dopamine in slices from Syt1^{-/-} mice, and that D2 receptor

and DAT function are maintained. But these are the only reliable conclusions that can be made. As noted for the in vivo experiments with a single dose of dopamine receptor drugs, examination of a single, saturating concentration of drug cannot provide information about receptor or transporter sensitivity – only that they are still functional in the KOs. Indeed, the comparable % increase in peak dopamine with sulpiride in KO and Syt1+/+ argue for unchanged D2 expression. In contrast, the greater increase in peak dopamine seen with DAT inhibition in Syt1-/- vs. Syt1+/+ mice would be consistent with enhanced DAT expression or function in the KOs. This result is the opposite of the predicted decrease in DAT activity when dopamine release is low – and opposite of what the authors actually found in their immunohistochemical evaluation of DAT expression (Fig. 8). This paradoxical result and the inconsistency with the immunohistochemistry should be discussed. Overall, the overstated conclusions about receptor and DAT sensitivity from these data should be removed from the text and the legend for Fig. 6.

13) 268-271. The lack of difference in the dopamine concentration of microdialysis samples from Syt1+/+ vs Syt1-/- reported here differs from the significantly lower dopamine levels reported previously for similar Syt1+/+ mice (Banerjee et al. 2018). The authors should discuss factors that might contribute to this difference, given that some of their conclusions about behaviors rely on this.

14) 384-385. The lack of change in dopamine tissue content (Fig. 5B), seems inconsistent with observed increases in TH and VMAT2, and argues against the conclusion that "the total density of dopaminergic axonal processes and possibly the total vesicular pool/stock of DA are increased."

15) 485-488. Related to this, evoked dopamine release from the dorsal striatum in the present study is ~5% of Syt1-/-, which is comparable to the release in KO mice illustrated in the previous report by Banerjee et al. (Fig 1G in that report), even if the previous authors used the term "abolished" to describe it. This difference is not as great as the present authors imply.

16) 633-645. This is a key concluding statement for the meaning of the present findings and should be emphasized, and much of the rest of the speculative discussion decreased by 50% (including speculative discussion of the role other synaptotagmins, and the overinterpreted compensation data based on single dose experiments).

Minor

Data in the text should be presented consistently with respect to the position of units and the use of significant figures. For example, on p 9, data are presented as 124s ± 12 for Syt1+/+ mice vs 197s ± 23s.... This should be 124 ± 12 s for Syt1+/+ mice vs 197 ± 23 s... with the units after the SEM only. Similarly, the time to descend in the pole test is given as 8s ± 0.7s vs 11s ± 175 0.5s – which be either 8.0 ± 0.7 s vs 11.0 ± 0.5 s or 8 ± 1 s vs 11 ± 1 s, depending on the actual precision of the timing measurement. The same issues occur throughout.

For Fig. 4, it can be assumed that the black lines are Syt1-/- and the green Syt1+/+, but this is not indicated in a key in the figure or specified in the legend. Both would be helpful.

71-76. Three genotypes were studied but only two sets of data are presented. Data for Syt1+/+ is labeled, but the other data is not. Presumably, this is for Syt1-/-, but this section needs to be edited for clarity.

Reviewer #4 (Remarks to the Author):

The authors have largely addressed my concerns over statistical rigor particularly relating to measures of basal dopamine (Fig6 and line 265). The additional statistical information is essential and has now been provided. However, there now remains what I believe to be a typographical error: an unpaired t-test should report a 't value', not an 'F-value' as is the case on line 265-267. The correct presentation of the test should be $t(20)$ (not $F(9,11)$) - this should be (easily) corrected before publication.

Having addressed mine and the other reviewers comments, I believe this work is ready and suitable for publication here.

We thank the reviewers for their additional suggestions to improve our manuscript. We have now further revised the manuscript and addressed all remaining concerns. We hope that the manuscript will now be considered suitable for publication in *Nature Communications*. Our point-by-point response to the remaining concerns can be found below in red.

Reviewer 3 comments:

We have addressed reviewer 3's concerns (section below). We have also shortened the part of the discussion discussing possible compensatory mechanisms. The sentences that were removed are indicated in red highlighting.

This revised paper has been strengthened by addressing several of the reviewers' concerns, albeit arguing against others. A particular strength is the addition of experiments to examine reward behavior in mice with conditional deletion of synaptotagmin 1 in dopamine neurons (Syt1^{-/-}) and in controls (Syt1^{+/-}). As reviewers noted previously, the basic finding of decreased axonal dopamine release in conditional Syt1^{-/-} mice was reported several years ago. The present study extends that by examining behavior in Syt1^{-/-} mice, with the unanticipated finding that motor and reward behaviors tested are largely unimpacted by loss of Syt1 in dopamine neurons. This leads to the very nice conclusions discussed in lines 633-645, including that the minimal behavioral effects of a profound attenuation of phasic dopamine release helps explain the maintenance of motor behavior in Parkinson's patients who have marked loss of dopamine. The findings further suggest that a dopamine tone is more important than a burst of dopamine cell activity and dopamine release for both motor behavior and reward learning. These findings are robust, and therefore allow for reliable

interpretation. Unfortunately, these potentially impactful findings and conclusions are lost in a very long and speculative discussion about compensatory changes in dopamine receptor and transporter sensitivity that is based on incomplete, and in some cases inconsistent data from pharmacological and immunohistochemical studies in the report. Additionally, two recent publications report a role for Syt1 in somatodendritic dopamine release from midbrain dopamine neurons. These findings need to be incorporated into the present rationale and discussion, as both argue against the authors' claim that there is no evidence for Syt1 in dopamine neurons. Despite this evidence in the literature, the decrease in dopamine release reported here is discounted, and argued away based on unconvincing optical stimulation experiments. Those data need to be aligned with other recent findings in the literature. These recent papers do not take away from the present work – as long as the present work emphasizes the most solid aspects of the vast array of experimental results reported, as noted above.

We thank the reviewer for her/his positive comments on our work. We agree that the implications of this work for the maintenance of motor functions in Parkinson's disease are important and we have tried in this further revised manuscript to highlight this better by decreasing the space allocated to discussing possible compensatory mechanisms, as suggested (see pages 28-29 for the sentences that were removed).

1) The authors now use "axon terminals" to indicate dopamine release sites. As this group and others have demonstrated, however, axonal release can occur from release sites all along the axon, so that the idea of a "terminal" per se has little meaning.

We agree and have replaced all occurrences of "axon terminals" by "release sites".

2) Based on their previously published results (Ref 11), the authors state that there is no evidence for Syt1 in the somatodendritic compartment of SNc dopamine neuron (p. 6, li. 102-103 and elsewhere). However, recently published functional and anatomical evidence demonstrate the presence of functional SYT1 in this compartment (Hikima et al. 2022; Lebowitz et al. 2023). The background for the experiments reported in Fig. 1 should be updated in light of these published findings. These results along with the data in Fig. 1 D,E also argue against the validity of evidence for the absence of Syt 1 in dopamine neurons (Ref 11), which should be discussed.

We agree that there is some functional evidence now supporting the possibility that Syt1 is also present in the somatodendritic compartment of dopamine neurons. We now acknowledge this in the revised introduction and cite the two papers mentioned by the reviewer (page 4). However, considering that neither of these two papers provides quantitative anatomical evidence directly showing the presence of Syt1 in the somatodendritic domain of dopamine neurons, we also refer in the revised discussion to the need for more quantitative anatomical work (page 30).

3) Interpretation of voltammetric detection of FSCV signals evoked using local electrical stimulation in mouse SNc is complicated by the presence of 5-HT release concurrently with dopamine (John et al. 2006). This is not considered as a confounding factor, but should be.

We agree and we now refer to this in the methods section (page 37). We state that: "For recordings in the VTA and SNc, it is theoretically possible that FSCV signals were partly contaminated by the presence of 5-HT, as previous studies showed that this biogenic amine is present in these regions and can be detected together with DA (PMID: 9375669, PMID: 7813678). However, the shape of our cyclic voltamograms suggest that such contamination was minimal or nonexistent.

4) The rationale and results from the optical stimulation experiments using ChR2-Kv in midbrain dopamine neurons remain hard to understand. The authors try to explain that somatodendritic dopamine release can be TTX-insensitive; however, recent studies using modern methods find that stimulated release is action-potential dependent – although spontaneous release under specific conditions is not. Thus, the TTX-insensitive release reported here (Fig. 2E,G,H) is very likely non-physiological, possibly from depolarization-dependent opening of Ca²⁺ channels by ChR2. Thus, the TTX-insensitive release from the VTA provides little information about the role of Syt1 in the process of somatodendritic release.

We would like to clarify that we do not believe that somatodendritic DA release is partially TTX-insensitive. We did not state this in the manuscript. We purposely decided to develop the ChR2-Kv technique to directly depolarize the somatodendritic domain of DA neurons, independently from action potentials, and this was specifically to attempt to limit the implication of axonal mechanisms in the DA release induced in the VTA and SNc. We wrote on page 7: “we devised a new strategy to trigger STD DA release more selectively”. We have now clarified this in the revised manuscript (page 8).

5) p. 8, 138-139. Whether the use of this construct sheds light on the role of Syt1 in dorsal striatum is also not obvious. Although release data were TTX-sensitive in dorsal and ventral striatum (pooled in Fig 2F, though separated in 2C and D), in contrast to midbrain, it remains unclear why there was any evoked dopamine release at all, given the authors’ rationale that ChR2-Kv should be only in the somatodendritic compartment. The authors go on to say that the small level of presumed dopamine release evoked with in these mice in dorsal striatum did not differ between Syt1^{-/-} mice and WT, “suggesting Syt1 is not required for this form of release in this region.” But it is not clear what “this form of release” is.

We understand the reviewer’s concerns. It is possible that ChR2-Kv found in the axonal domain after overexpression represents aberrantly targeted channels. But, as the results show, this represented only a small part of the release normally evoked by standard ChR2 or electrical stimulation. We removed the part of the sentence found unclear by the reviewer.

6) 155. Given that the authors found that somatodendritic dopamine release was lower in Syt1^{-/-} mice than in WT in electrical stimulation experiments, the conclusion that Syt1 is not involved in “selectively triggered” somatodendritic dopamine release seems contrived, as well as not obviously physiologically relevant as noted above.

We hope that the explanations given above for point 4 will have clarified the situation and what we mean by “selectively-triggered” STD DA release. To clarify things further, we modified the sentence in the following manner: “Together these observations suggest that when STD DA release is triggered using an approach that minimizes any axonal activation, the signal that is detected is unaltered by loss of Syt1.” (pages 8-9).

7) 163-165. Do the authors think the site of action of impaired dopamine release is axonal or somatodendritic or both? This should be expanded.

We were referring mainly to axonal release. This is now clarified in the text (page 9).

8) The findings with D1 and D2 receptor drugs seem contradictory at best (Fig. 3). In Parkinson’s disease, D1 and D2 agonists are administered to improve mobility, yet here, D1 and D2 agonism cause a marked decrease in locomotion in all mouse genotypes tested. Equally perplexing is that a D2 antagonist (raclopride) also caused a decrease in locomotion. A contributing factor is that injection of

saline alone (Fig. 3D) also caused a sharp decrease in locomotor activity in Syt+/+ and Syt1-/- mice, which would be a confounding factor in the similar drug studies.

The reviewer is right that D2 agonists can be used to treat PD. However, D1 agonists are not (although some work had explored their use in the past). There is a very broad and multiple-decade old literature showing that D2 agonists at doses that activate D2 autoreceptors (such as the dose we used) induce a large decrease in locomotion in rodents. This is thought to be due to D2 autoreceptors shutting down dopamine release. As such, our results showing a decrease in locomotion induced by quinpirole are not surprising. Similarly, D2 receptor antagonists (including all antipsychotics such as raclopride) are well-known to reduce locomotion. In this case, this is due to the blockade of postsynaptic receptors in the striatum. We hope this clarifies the situation.

9) Rather than address the bigger picture issues raised by these internally inconsistent data with D1 and D2 drugs, the authors instead identify some subtle differences among the genotypes in the first 5 min after drug administration. They then base conclusions about receptor sensitivity on these small transient differences. For example, data 5 min after administration of quinpirole, a D2 receptors agonist, shows greater motor suppression in Syt1-/- than in Syt1+/+ mice. However, this single time point result is not convincing, given the near-complete suppression of movement 5 min later in all 3 genotypes with the single, saturating concentration tested. Better would have to have tested lower drug concentrations that might have revealed a more robust difference, and better yet would have been to have done a dose response curve. Similarly, rather than attempt to explain how both raclopride and quinpirole suppress movement, they again pick the 5 min time point to show a lesser initial effect of D2 antagonism with raclopride in Syt1-/- than in Syt1+/+ mice, after which movement is largely absent. This difference is framed as a “significantly increased response in the Syt1-/- mice compared to Syt1+/+ mice” – however, this is misleading, as a lesser decrease is not an increase (Fig. 3L).

We hope that the explanations given in point 8 above clarify why our behavioral results are not inconsistent. With regards to the reason to focus on the first 5 min, this is due to pharmacokinetic considerations. Considering that we worked with single saturating doses of the ligands, the results measured during the first few minutes will necessarily correspond to partial occupancy of the receptors. We now state this (page 11). This being said, we agree with the reviewer that performing dose-response curves would have been ideal (now mentioned in the revised discussion (page 26). However, this would correspond to months of additional work and generation of multiple cohorts of additional mice. Considering that this limitation concerns a very peripheral point of our manuscript, we hope the reviewer will understand why we decided against going forward with this. We have also corrected the sentence mentioned by the reviewer to refer instead to “significantly increased inhibition of locomotion” (page 11).

10) The lack of difference in basal dopamine concentration with microdialysis (Fig. 5) is a key finding in the report. However, baseline determinations are most accurate when conducted using the no-net flux method, as this takes into account changes in uptake activity, etc. The limitations of the method used should be discussed.

We agree with the reviewer. We now state that additional experiments using the no-net flux approach would be useful to validate and extend our findings. This mentioned in the revised discussion section (page 26).

11) In the microdialysis studies, [5-HT] was 2-fold higher in *Syt1*^{-/-} than *Syt1*^{+/+} mice in both striatum and midbrain. These were not significant, which could reflect under-sampling, given the variability of the data. However, it is not clear why ANOVA was used here. If unpaired t-tests were used to compare [5-HT] in striatum and separately in midbrain, was a difference seen? In any case, a role for 5-HT should be discussed as a possible contributing factor in the lack of effect of *Syt1* deletion in dopamine neurons on the various behaviors reported here.

We thank the reviewer for bringing this issue to our attention. In the present case, we agree that using unpaired t-tests instead of ANOVAs to analyze dialysate levels is more appropriate because we do not compare concentrations between structures. Therefore, we revised this analysis and report the new statistical results in the revised manuscript. With this new analysis, we can now conclude that there was a significant increase in extracellular 5-HT levels in the striatum of the KO mice (page 15). We also discuss the possible implication of adaptations in the 5-HT system in the KO mice (page 28).

Despite a reasonably high number of animals used for the microdialysis experiments (n= 10-12), we acknowledge that the variability of the data is a limitation. However, our findings regarding dopamine levels in *Syt1* cKO^{DA} mice are not strikingly different compared to the previous results reported by Banerjee and al.

12) The data in Fig. 6 support detection of dopamine in slices from *Syt1*^{-/-} mice, and that D2 receptor and DAT function are maintained. But these are the only reliable conclusions that can be made. As noted for the in vivo experiments with a single dose of dopamine receptor drugs, examination of a single, saturating concentration of drug cannot provide information about receptor or transporter sensitivity – only that they are still functional in the KOs. Indeed, the comparable % increase in peak dopamine with sulpiride in KO and *Syt1*^{+/+} argue for unchanged D2 expression. In contrast, the greater increase in peak dopamine seen with DAT inhibition in *Syt1*^{-/-} vs. *Syt1*^{+/+} mice would be consistent with enhanced DAT expression or function in the KOs. This result is the opposite of the predicted decrease in DAT activity when dopamine release is low – and opposite of what the authors actually found in their immunohistochemical evaluation of DAT expression (Fig. 8). This paradoxical result and the inconsistency with the immunohistochemistry should be discussed. Overall, the overstated conclusions about receptor and DAT sensitivity from these data should be removed from the text and the legend for Fig. 6.

As this is the second round of revision, we are surprised by these new comments. We agree that using single doses has limitations and we have now acknowledged this in the revised discussion (page 27). The reviewer mentions that our FSCV results with sulpiride argue for unchanged D2 expression. In the dorsal striatum, it is true that the ability of sulpiride to increase DA release was proportionally not different in the cKO mice. However, sulpiride caused a proportionally higher increase in DA release in the ventral striatum (Fig. 6D). And quinpirole caused a proportionally higher inhibition of DA release in both the dorsal and ventral striatum (Fig. 6A, 6B). This fits with our proposition that there are more D2 receptors in the ventral striatum in the cKO mice, a conclusion that is also supported by our autoradiography results and behavioral results. Such a conclusion is also keeping with a very broad literature showing that when DA levels are chronically reduced in the brain (here limited to blocked phasic DA release), the number or the function of D2 receptors are homeostatically upregulated. With regards to conclusions concerning the DAT and the fact that there is a mismatch between the immunohistochemistry results (suggesting reduced DAT) and the western blot and qPCR results (suggesting no change), we agree this is worth acknowledging and discussion. This is something that we already do (pages 27-28, lines 623 to 635). Considering the request by reviewer 3 to shorten the section of the discussion that concerns compensations, we opted not to develop this section further

and hope the reviewer will understand. Finally, as requested, we removed the reference to changes in DAT function from the title of figure 6.

13) 268-271. The lack of difference in the dopamine concentration of microdialysis samples from Syt1+/+ vs Syt1-/- reported here differs from the significantly lower dopamine levels reported previously for similar Syt1+/+ mice (Banerjee et al. 2018). The authors should discuss factors that might contribute to this difference, given that some of their conclusions about behaviors rely on this.

In the Banerjee paper, basal levels of extracellular dopamine measured by microdialysis and HPLC were only modestly decreased in the cKO mice (figure 3G). Three time points were measured and at only one of these was a significant decrease observed. As such, our findings are not strikingly different compared to these previous results. We now mention this in the revised discussion (page 25).

14) 384-385. The lack of change in dopamine tissue content (Fig. 5B), seems inconsistent with observed increases in TH and VMAT2, and argues against the conclusion that "the total density of dopaminergic axonal processes and possibly the total vesicular pool/stock of DA are increased."

We agree. We have shortened this sentence to write instead "the total density of dopaminergic axonal processes is increased" (page 20).

15) 485-488. Related to this, evoked dopamine release from the dorsal striatum in the present study is ~5% of Syt1-/-, which is comparable to the release in KO mice illustrated in the previous report by Banerjee et al. (Fig 1G in that report), even if the previous authors used the term "abolished" to describe it. This difference is not as great as the present authors imply.

This is true. In that paper, the actual percent decrease is not provided. So, it is difficult to compare with the number we obtained. In the revised manuscript, we replaced the word "completely" by "mostly" and emphasized instead the difference between dorsal and ventral striatum (page 24), something that was not examined in the previous report.

16) 633-645. This is a key concluding statement for the meaning of the present findings and should be emphasized, and much of the rest of the speculative discussion decreased by 50% (including speculative discussion of the role other synaptotagmins, and the overinterpreted compensation data based on single dose experiments).

We have reduced the part of the discussion addressing compensations (sections in red on pages 28-29). We agree that this change will leave more prominence to the other conclusions related to Parkinson's disease. However, we have not increased further this section so as not to make the discussion too long.

Minor

Data in the text should be presented consistently with respect to the position of units and the use of significant figures. For example, on p 9, data are presented as 124s ± 12 for Syt1+/+ mice vs 197s ± 23s.... This should be 124 ± 12 s for Syt1+/+ mice vs 197 ± 23 s... with the units after the SEM only. Similarly, the time to descend in the pole test is given as 8s ± 0.7s vs 11s ± 175 0.5s – which be either 8.0 ± 0.7 s vs 11.0 ± 0.5 s or 8 ± 1 s vs 11 ± 1 s, depending on the actual precision of the timing measurement. The same issues occur throughout.

We would like to thank the reviewer for his/her attention to details. We went throughout the text to modify the position of units each time this was relevant.

For Fig. 4, it can be assumed that the black lines are $Syt1^{-/-}$ and the green $Syt1^{+/+}$, but this is not indicated in a key in the figure or specified in the legend. Both would be helpful.

The key was at the bottom of the graph. But we have now moved it to the top for better visibility. We also modified the legend to indicate the color code.

71-76. Three genotypes were studied but only two sets of data are presented. Data for $Syt1^{+/+}$ is labeled, but the other data is not. Presumably, this is for $Syt1^{-/-}$, but this section needs to be edited for clarity.

We modified the result section to include the values for heterozygous mice.

REVIEWERS' COMMENTS

Reviewer #3 (Remarks to the Author):

The authors have addressed all of the minor points in the previous review, and now highlight the major conclusions by deleting some of the speculative discussion about compensation in the discussion. Overall, the behavioral studies remain a strength, and the conclusions that can be drawn from them are the centerpiece of the manuscript. Other points will be resolved in time.